# Sample-Efficient Private Learning of Mixtures of Gaussians

**Hassan Ashtiani**
McMaster University
zokaeiam@mcmaster.ca

**Mahbod Majid**
MIT
mahbod@mit.edu

**Shyam Narayanan**
Citadel Securities*
shyam.s.narayanan@gmail.com

## Abstract

We study the problem of learning mixtures of Gaussians with approximate differential privacy. We prove that roughly $kd^2 + k^{1.5}d^{1.75} + k^2d$ samples suffice to learn a mixture of $k$ *arbitrary* $d$-dimensional Gaussians up to low total variation distance, with differential privacy. Our work improves over the previous best result [AAL24b] (which required roughly $k^2d^4$ samples) and is provably optimal when $d$ is much larger than $k^2$. Moreover, we give the first optimal bound for privately learning mixtures of $k$ *univariate* (i.e., 1-dimensional) Gaussians. Importantly, we show that the sample complexity for learning mixtures of univariate Gaussians is linear in the number of components $k$, whereas the previous best sample complexity [AAL21] was quadratic in $k$. Our algorithms utilize various techniques, including the *inverse sensitivity mechanism* [AD20b, AD20a, HKMN23], *sample compression for distributions* [ABDH+20], and methods for bounding volumes of sumsets.

## 1 Introduction

Learning **Gaussian Mixture Models** (GMMs) is one of the most fundamental problems in algorithmic statistics. Gaussianity is a common data assumption, and the setting of Gaussian mixture models is motivated by heterogeneous data that can be split into numerous clusters, where each cluster follows a Gaussian distribution. Learning mixture models is among the most important problems in machine learning [Bis06], and is at the heart of several unsupervised and semi-supervised machine learning models. The study of Gaussian mixture models has had numerous scientific applications dating back to the 1890s [Pea94], and is a crucial tool in modern data analysis techniques in a variety of fields, including bioinformatics [LKWB22], anomaly detection [ZSM+18], and handwriting analysis [Bis06].

In this work, we study the problem of learning a GMM from samples. We focus on the *density estimation* setting, where the goal is to learn the overall mixture distribution up to low total variation distance. Unlike the *parameter estimation* setting for GMMs, density estimation can be done even without any boundedness or separation assumptions on the parameters of the components. In fact, it is known that mixtures of $k$ Gaussians in $d$-dimensions can be learned up to total variation distance $\alpha$ using $\widetilde{O}(kd^2/\alpha^2)$ samples [ABH+18].

Ensuring data privacy has emerged as an increasingly important challenge in modern data analysis and statistics. *Differential privacy* (DP) [DMNS06] is a rigorous way of defining privacy, and is considered to be the gold standard both in theory and practice, with deployments by Apple [Tea17], Google [EPK14], Microsoft [DKY17], and the US Census Bureau [DLS+17]. As is the case for many data analysis tasks, standard algorithms for learning GMMs leak potentially sensitive information about the individuals who contributed data. This raises the question of whether we can do density estimation for GMMs under the constraint of differential privacy.

---

*Work done as a student at MIT

38th Conference on Neural Information Processing Systems (NeurIPS 2024).

Private density estimation for GMMs with unrestricted Gaussian components is a challenging task. In fact, privately learning a single unrestricted Gaussian has been the subject of multiple recent studies [AAK21, KMS$^+$22b, AL22, KMV22, AKT$^+$23, HKMN23]. Private learning of GMMs is significantly more challenging, because even without privacy constraints, parameter estimation for GMMs requires exponentially many samples in terms of the number of components [MV10]. Therefore, it is not clear how to use the typical recipe of "adding noise" to the estimated parameters or "privately choosing" from the finite-dimensional space of parameters. Consequently, the only known sample complexity bounds for privately learning unrestricted GMMs are loose [AAL24b, AAL21].

Let us first formally define the problem of learning GMMs. We represent a GMM $\mathcal{D} = \sum_{i=1}^{k} w_i \mathcal{N}(\mu_i, \Sigma_i)$ by its parameters, namely $\{(w_i, \mu_i, \Sigma_i)\}_{i=1}^{k}$, where $w_i \geq 0$, $\sum_i w_i = 1$, $\mu_i \in \mathbb{R}^d$, and $\Sigma_i$ is a positive definite matrix. In the following, a GMM learning algorithm $\mathcal{A}$ receives a set of data points in $\mathbb{R}^d$ and outputs a (representation of) a GMM. The total variation distance between two distributions is $d_{\mathrm{TV}}(\tilde{\mathcal{D}}, \mathcal{D}) = \frac{1}{2} \int_{\mathbb{R}^d} |\mathcal{D}(x) - \tilde{\mathcal{D}}(x)| dx$[2].

**Definition 1.1** (Learning GMMs). For $\alpha, \beta \in (0, 1)$, we say $\mathcal{A}$ learns GMMs with $n$ samples up to accuracy $\alpha$ and failure probability $\beta$ if for every GMM $\mathcal{D}$, given samples $X_1, \ldots, X_n \overset{i.i.d.}{\sim} \mathcal{D}$, it outputs (a representation of) a GMM $\tilde{\mathcal{D}}$ such that $d_{\mathrm{TV}}(\tilde{\mathcal{D}}, \mathcal{D}) \leq \alpha$ with probability at least $1 - \beta$.

$\alpha$ and $\beta$ are called the accuracy and failure probability, respectively. For clarity of presentation, we will typically fix the value of $\beta$ (e.g., $\beta = 1/3$). The above definition does not enforce the constraint of differential privacy. The following definitions formalizes (approximate) differential privacy.

**Definition 1.2** (Differential Privacy (DP) [DMNS06, DKM$^+$06]). Let $\varepsilon, \delta \geq 0$. A randomized algorithm $\mathcal{A} : \mathcal{X}^n \to \mathcal{O}$ is said to be $(\varepsilon, \delta)$-differentially private $((\varepsilon, \delta)$-DP) if for any two neighboring datasets $\mathbf{X}, \mathbf{X}' \in \mathcal{X}^n$ and any measurable subset $O \subset \mathcal{O}$,

$$\mathbb{P}[\mathcal{A}(\mathbf{X}') \in O] \leq e^{\varepsilon} \cdot \mathbb{P}[\mathcal{A}(\mathbf{X}) \in O] + \delta.$$

If the GMM learner $\mathcal{A}$ of Definition 1.1 is $(\varepsilon, \delta)$-DP, we say that $\mathcal{A}$ privately learns GMMs. Formally, we have the following definition.

**Definition 1.3** (Privately learning GMMs). Fix the number of samples $n$, dimension $d$, and number of mixture components $k$. For $\alpha, \beta \in (0, 1)$ and $\varepsilon, \delta \geq 0$, a randomized algorithm $\mathcal{A}$, that takes as input $X_1, \ldots, X_n \in \mathbb{R}^d$, $(\varepsilon, \delta)$-privately learns GMMs up to accuracy $\alpha$ and failure probability $\beta$, if:

1. For any GMM $\mathcal{D}$ that is a mixture of up to $k$ Gaussians in $d$ dimensions, if $\mathbf{X} = \{X_1, \ldots, X_n\} \overset{i.i.d.}{\sim} \mathcal{D}$, $\mathcal{A}(X_1, \ldots, X_n)$ outputs a GMM $\tilde{\mathcal{D}}$ such that $d_{\mathrm{TV}}(\tilde{\mathcal{D}}, \mathcal{D}) \leq \alpha$ with probability at least $1 - \beta$ (over the randomness of the data $\mathbf{X}$ and the algorithm $\mathcal{A}$).

2. For *any* neighboring datasets $\mathbf{X}, \mathbf{X}' \in (\mathbb{R}^d)^n$ (not necessarily drawn from any GMM) and any measurable subset $O \subset \mathcal{O}$, $\mathbb{P}[\mathcal{A}(\mathbf{X}') \in O] \leq e^{\varepsilon} \cdot \mathbb{P}[\mathcal{A}(\mathbf{X}) \in O] + \delta$.

Finally, we assume a default value for $\beta$ of $1/3$, meaning that if not stated, the failure probability $\beta$ is assumed to equal $1/3$.

Our main goal in this paper is to understand the number of samples (as a function of the dimension $d$, the number of mixture components $k$, the accuracy $\alpha$, and the privacy parameters $\varepsilon, \delta$) that are needed to privately and accurately learn the GMM up to low total variation distance.

## 1.1 Results

In this work, we provide improved sample complexity bounds for privately learning mixtures of arbitrary Gaussians, improving over previous work of [AAL21, AAL24b]. Moreover, our sample complexity bounds are optimal in certain regimes, when the dimension is either 1 or a sufficiently large polynomial in $k$ and $\log \frac{1}{\delta}$. For general dimension $d$, we prove the following theorem.

**Theorem 1.4.** *For any $\alpha, \varepsilon, \delta \in (0, 1), k, d \in \mathbb{N}$, there exists an inefficient $(\varepsilon, \delta)$-DP algorithm that can learn a mixture of $k$ arbitrary full-dimensional Gaussians in $d$ dimensions up to accuracy $\alpha$, using the following number of samples:*

$$n = \widetilde{O} \left( \frac{kd^2}{\alpha^2} + \frac{kd^2 + d^{1.75}k^{1.5} \log^{0.5}(1/\delta) + k^{1.5} \log^{1.5}(1/\delta)}{\alpha\varepsilon} + \frac{k^2 d}{\alpha} \right).$$

---

[2]We are slightly abusing the notation and using $\mathcal{D}(x)$ as the pdf of $\mathcal{D}$ at points $x$.

Notably, the mixing weights and the means can be arbitrary and the covariances of the Gaussians can be arbitrarily poorly conditioned, as long as the covariances are non-singular[3].

We remark that we omit the dependence on $\beta$ (and assume by default a failure probability of $1/3$). However, it is well-known that one can obtain failure probability $\beta$ with only a multiplicative $O(\log 1/\beta)$ blowup in sample complexity, in a black-box fashion[4]. In fact, our analysis can yield even better dependencies on $\beta$ in some regimes, though to avoid too much complication, we do not analyze this.

For reasonably large dimension, i.e., $d \geq k^2 \log^2(1/\delta)$, this can be simplified to $\tilde{O}\left(\frac{kd^2}{\alpha^2} + \frac{kd^2}{\alpha\varepsilon}\right)$, which is in fact optimal (see Theorem 1.6). Hence, we obtain the *optimal* sample complexity for sufficiently large dimension. Theorem 1.4 also improves over the previous best sample complexity upper bound of [AAL24b], which uses

$$\widetilde{O}\left(\frac{k^2d^4 + kd^2\log(1/\delta)}{\alpha^2\varepsilon} + \frac{kd\log(1/\delta)}{\alpha^3\varepsilon} + \frac{k^2d^2}{\alpha^4\varepsilon}\right)$$

samples. Our results provide a polynomial improvement in all parameters, but to simplify the comparison, if we ignore dependencies in the error parameter $\alpha$ and privacy parameters $\varepsilon, \delta$, we improve the sample complexity from $k^2d^4$ to $kd^2 + k^2d + k^{1.5}d^{1.75}$: note that our result is quadratic in the dimension whereas [AAL24b] is quartic.

When the dimension is $d = 1$, we can provide an improved result, which is *optimal* for learning mixtures of univariate Gaussians (see Theorem 1.6 for a matching lower bound).

**Theorem 1.5.** *For any $\alpha, \varepsilon, \delta \in (0,1), k \in \mathbb{N}$, there exists an inefficient $(\varepsilon, \delta)$-DP algorithm that can learn a mixture of $k$ arbitrary univariate Gaussians (of nonzero variance) up to accuracy $\alpha$, using the following number of samples:*

$$n = \widetilde{O}\left(\frac{k}{\alpha^2} + \frac{k\log(1/\delta)}{\alpha\varepsilon}\right).$$

For privately learning mixtures of univariate Gaussians, the previous best-known result for arbitrary Gaussians required $\widetilde{O}\left(\frac{k^2\log^{3/2}(1/\delta)}{\alpha^2\varepsilon}\right)$ samples [AAL21]. Importantly, we are the first paper to show that the sample complexity can be *linear* in the number of components.

Our work purely focuses on sample complexity, and as noted in Theorems 1.4 and 1.5, they do not have polynomial time algorithms. We note that the previous works of [AAL21, AAL24b] also do not run in polynomial time. Indeed, there is reason to believe that even non-privately, it is impossible to learn GMMs in polynomial time (in terms of the optimal sample complexity) [DKS17, BRST21, GVV22].

Finally, we prove the following lower bound for learning GMMs in any fixed dimension $d$.

**Theorem 1.6.** *Fix any dimension $d \geq 1$ number of components $k \geq 2$, any $\alpha, \varepsilon$ at most a sufficiently small constant $c^*$, and $\delta \leq (\alpha\varepsilon/d)^{O(1)}$. Then, any $(\varepsilon, \delta)$-DP algorithm that can learn a mixture of $k$ arbitrary full-dimensional Gaussians in $d$ dimensions up to total variation distance $\alpha$, with probability at least $2/3$, requires at least the following number of samples:*

$$\tilde{\Omega}\left(\frac{kd^2}{\alpha^2} + \frac{kd^2}{\alpha\varepsilon} + \frac{k\log(1/\delta)}{\alpha\varepsilon}\right).$$

Note that for $d = 1$, this matches the upper bound of Theorem 1.5, thus showing that our univariate result is near-optimal in all parameters $\alpha, \varepsilon, \delta$. Moreover, our lower bound refutes the conjecture of [AAL21], which conjectures that only $\Theta\left(\frac{k}{\alpha^2} + \frac{k}{\alpha\varepsilon} + \frac{\log(1/\delta)}{\varepsilon}\right)$ samples are needed in the univariate case and $\Theta\left(\frac{kd^2}{\alpha^2} + \frac{kd^2}{\alpha\varepsilon} + \frac{\log(1/\delta)}{\varepsilon}\right)$ samples are needed in the $d$-dimensional case. However, we note that our lower bound asymptotically differs from the conjectured bound in [AAL21] only when $\delta$ is extremely small.

---

[3]For clarity of presentation, we assume the covariance matrices are not singular. However, extending our results to degenerate matrices is straightforward.

[4]To obtain success probability $\beta$ with $O(n \cdot \log 1/\beta)$ samples, we repeat the procedure $T = O(\log 1/\beta)$ times on independent groups of $n$ samples each, to get $T$ estimates $\tilde{\mathcal{D}}_1, \ldots, \tilde{\mathcal{D}}_T$, and by a Chernoff bound, at least $51\%$ of the estimates are within total variation distance $\alpha$ of the true mixture $\mathcal{D}$. So, by choosing an estimate that is within $2\alpha$ of at least $51\%$ of the estimates, it is still within $3\alpha$ total variation distance of $\mathcal{D}$.

## 1.2 Related work

In the non-private setting, the sample complexity of learning unrestricted GMMs with respect to total variation distance (a.k.a. density estimation) is known to be $\widetilde{\Theta}(kd^2/\alpha^2)$ [ABM18, ABH$^+$18], where the upper bound is obtained by the so-called distributional compression schemes.

In the private setting, the only known sample complexity upper bound for unrestricted GMMs [AAL24b] is roughly $k^2 d^4 \log(1/\delta)/(\alpha^4 \varepsilon)$, which exhibits sub-optimal dependence on various parameters[5]. This bound is achieved by running multiple non-private list-decoders and then privately aggregating the results. For the special case of axis-aligned GMMs, an upper bound of $k^2 d \log(1/\delta)^{3/2}/(\alpha^2 \varepsilon)$ is known [AAL21]. These are the only known results even for privately learning (unbounded) univariate GMMs. In other words, the best known upper bound for sample complexity of privately learning univariate GMMs has quadratic dependence on $k$.

In the related public-private setting [BKS22, BBC$^+$23], it is assumed that the learner has access to some public data. In this setting, [BBC$^+$23] show that unrestricted GMMs can be learned with a moderate amount of public and private data.

Assuming the parameters of the Gaussian components (and the condition numbers of the covariance matrices) are bounded, one can create a cover for GMMs and use private hypothesis selection [BSKW19] or the private minimum distance estimator [AAK21] to learn the GMM. On the flip side, [ASZ21] prove a lower bound on the sample complexity of learning GMMs, though their lower bound is weaker than ours and is only against pure-DP algorithms.

The focus of our work is on density estimation. A related problem is learning the parameters a GMM, which has received extensive attention in the (non-private) literature (e.g., [Das99, MV10, BS10, LM21, BDJ$^+$22, LL22] among many other papers). To avoid identifiability issues, one has to assume that the Gaussian components are sufficiently separated and have large-enough weights. In the private setting, the early work of [NRS07] demonstrated a privatized version of [VW04] for learning GMMs with fixed (known) covariance matrices. The strong separation assumption (of $\Omega(k^{1/4})$) between the Gaussian components in [NRS07] was later relaxed to a weaker separation assumption [CCAd$^+$23]. A substantially more general result for privately learning GMMs with unknown covariance matrices was established in [KSSU19], based on a privatized version of [AM05]. Yet, this approach also requires a polynomial separation (in terms of $k$) between the components, as well as a bound on the spectrum of the covariance matrices. [CKM$^+$21] weakened the separation assumption of [KSSU19] and improved over their sample complexity. This result is based on a generic method that learns a GMM using a private learner for Gaussians and a non-private clustering method for GMMs. Finally, [AAL23] designed an efficient reduction from private learning of GMMs to its non-private counterpart, removing the boundedness assumptions on the parameters and achieving minimal separation (e.g., by reducing to [MV10]). Nevertheless, unlike density estimation, parameter estimation for unrestricted GMMs requires exponentially many samples in terms of $k$ [MV10].

A final important question is that of *efficient* algorithms for learning GMMs. Much of the work on learning GMM parameters focuses on computational efficiency (e.g,. [MV10, BS10, LM21, BDJ$^+$22, LL22]), as does some work on density estimation (e.g., [CDSS14, ADLS17]). However, under some standard hardness assumptions, it is known that even non-privately learning mixtures of $k$ $d$-dimensional Gaussians with respect to total variation distance cannot be done in polynomial time as a function of $k$ and $d$ [DKS17, BRST21, GVV22].

**Addendum.** In a concurrent submission, [AAL24a] extended the result of [AAL24b] for learning unrestricted GMMs to the agnostic (i.e., robust) setting. In contrast, our algorithm works only in the realizable (non-robust) setting. Moreover, [AAL24a] slightly improved the sample complexity result of [AAL24b] from $\widetilde{O}(\log(1/\delta)k^2 d^4/(\varepsilon\alpha^4))$ to $\widetilde{O}(\log(1/\delta)k^2 d^4/(\varepsilon\alpha^2))$. The sample complexity of our approach is still significantly better than [AAL24a] in terms of all parameters—similar to the way it improved over [AAL24b].

---

[5]More precisely, the upper bound is $\widetilde{O}\left(\frac{k^2 d^4}{\alpha^2 \varepsilon} + \frac{kd^2 \log(1/\delta)}{\alpha^2 \varepsilon} + \frac{kd \log(1/\delta)}{\alpha^3 \varepsilon} + \frac{k^2 d^2}{\alpha^4 \varepsilon}\right)$

## 2 Technical overview and roadmap

We highlight some of our conceptual and technical contributions. We mainly focus on the high-dimensional upper bound, and discuss the univariate upper bound at the end.

### 2.1 Reducing to crude approximation

Suppose we are promised a bound on the means and covariances of the Gaussian components, i.e., $\frac{1}{R} \cdot I \preccurlyeq \Sigma_i \preccurlyeq R \cdot I$ and $\|\mu_i\|_2 \leq R$ for all $i \in [k]$. In this case, there is in fact a known algorithm, using private hypothesis selection [BSKW19, AAK21], that can privately learn the distribution using only $O\left(\frac{kd^2 \log R}{\alpha^2} + \frac{kd^2 \log R}{\alpha \varepsilon}\right)$ samples. Moreover, with a more careful application of the hypothesis selection results (see Appendix D), we can prove that result holds even if $(\mu_i, \Sigma_i)$ are possibly unbounded, but we have some very crude approximation. By this, we mean that if for each $i \in [k]$ we know some $\hat{\Sigma}_i$ such that $\frac{1}{R} \cdot \Sigma_i \preccurlyeq \hat{\Sigma}_i \preccurlyeq R \cdot \Sigma_i$, then it suffices to have $n = O\left(\frac{kd^2 \log R}{\alpha^2} + \frac{kd^2 \log R}{\alpha \varepsilon}\right)$ samples to learn the full GMM in total variation distance.

Our main goal will be to learn every covariance $\Sigma_i$ with such an approximation, for $R = \mathrm{poly}\left(k, d, \frac{1}{\alpha}, \frac{1}{\varepsilon}\right)$, so that $\log R$ can be hidden in the $\tilde{O}$ factor. To explain why this goal is sufficient, suppose we can crudely learn every covariance $\Sigma_i$ with approximation ratio $R$, using $n'$ samples. We then need $O\left(\frac{kd^2 \log R}{\alpha^2} + \frac{kd^2 \log R}{\alpha \varepsilon}\right) = \tilde{O}\left(\frac{kd^2}{\alpha^2} + \frac{kd^2}{\alpha \varepsilon}\right)$ additional samples to learn the full distribution using hypothesis selection, so the total sample complexity is $\tilde{O}\left(n' + \frac{kd^2 \log R}{\alpha^2} + \frac{kd^2 \log R}{\alpha \varepsilon}\right)$. Hence, we will aim for this easier goal of crudely learning each covariance, for both Theorem 1.4 and Theorem 1.5, using as few samples $n'$ as possible. We will also need to approximate each mean $\mu_i$, though for simplicity we will just focus on covariances in this section.

### 2.2 Overview of Theorem 1.5 for univariate GMMs

The main goal will be to simply provide a rough approximation to the set of standard deviations $\sigma_i = \sqrt{\Sigma_i}$, as we can finish the procedure with hypothesis selection, as discussed above.

**Bird's eye view:** Say we are given a dataset $\mathbf{X} = \{X_1, \ldots, X_n\}$: note that every $X_j \in \mathbb{R}$ since we are dealing with univariate Gaussians. The main insight is to sort the data in increasing order (i.e., reorder so that $X_1 \leq X_2 \leq \cdots \leq X_n$) and consider the unordered multiset of successive differences $\{X_2 - X_1, X_3 - X_2, \ldots, X_n - X_{n-1}\}$. One can show that if a single datapoint $X_j$ is changed, then the set of consecutive differences (up to permutation) does not change in more than 3 locations (see Lemma F.5 for a formal proof).

We then apply a standard private histogram approach. Namely, for each integer $a \in \mathbb{Z}$, we create a corresponding bucket $B_a$, and map each $X_{j+1} - X_j$ into $B_a$ if $2^a \leq X_{j+1} - X_j < 2^{a+1}$. If some mixture component $i$ had variance $\Sigma_i = \sigma_i^2$, we should expect a significant number of $X_{j+1} - X_j$ to at least be crudely close to $\sigma_i$, such as for the $X_j$ drawn from the $i^{\text{th}}$ mixture component. So, some corresponding bucket should be reasonably large. Finally, by adding noise to the count of each bucket and taking the largest noisy counts, we will successfully find an approximation to all variances.

**In more detail:** For each (possibly negative) integer $a$ let $c_a$ be the number of indices $i$ such that $2^a \leq X_{j+1} - X_j < 2^{a+1}$. We will prove that, if the weight of the $i^{\text{th}}$ component in the mixture is $w_i$ and there are $n$ points, then we should expect at least $\Omega(w_i \cdot n)$ indices $j$ to be in a bucket $a$ with $2^a$ within a $\mathrm{poly}(n)$ multiplicative factor of the standard deviation $\sigma_i$ (see Lemma F.3). The point of this observation is that there are at most $O(\log n)$ buckets $B_a$ with $2^a$ between $\frac{\sigma_i}{\mathrm{poly}(n)}$ and $O(\sigma_i)$, but there are at least $\Omega(w_i \cdot n)$ indices mapping to one of these buckets. So by the Pigeonhole principle, one of these buckets has at least $\Omega\left(\frac{w_i \cdot n}{\log n}\right)$ indices, i.e., $c_a \geq \Omega\left(\frac{w_i \cdot n}{\log n}\right)$ for some $a$ with $\frac{\sigma_i}{\mathrm{poly}(n)} \leq 2^a \leq O(\sigma_i)$.

Moreover, we know that if we change a single data point $X_j$, the set of consecutive differences $\{X_{j+1} - X_j\}$ after sorting changes by at most 3. So, if we change a single $X_j$, at most 3 of the counts $c_a$ can change, each by at most 3.

Now, for every integer $a$, draw a noise value from the *Truncated Laplace* distribution (see Definition A.2 and Lemma A.3), and add it to $c_a$ to get a noisy count $\tilde{c}_a$. The details of the noise distribution are not important, but the idea is that this distribution is *always* bounded by $O\left(\frac{1}{\varepsilon}\log\frac{1}{\delta}\right)$. Moreover, the Truncated Laplace distribution preserves $(\varepsilon,\delta)$-DP. This means that the counts $\{\tilde{c}_a\}_{a\in\mathbb{Z}}$ will have $(O(\varepsilon),O(\delta))$-DP, because the true counts $c_a$ only change minimally across adjacent datasets.

Our crude approximation to the set of standard deviations will be the set of $2^a$ such that $\tilde{c}_a$ exceeds some threshold which is a large multiple of $\frac{1}{\varepsilon}\log\frac{1}{\delta}$. If $n \geq \tilde{O}\left(\frac{k\log(1/\delta)}{\alpha\varepsilon}\right)$ and the weight $w_i \geq \alpha/k$, it is not hard to verify that $\frac{w_i \cdot n}{\log n}$ exceeds a large multiple of $\frac{1}{\varepsilon}\log\frac{1}{\delta}$. So, for each $i \leq k$ with weight at least $\alpha/k$, some corresponding $a$ with $\frac{\sigma_i}{\text{poly}(n)} \leq 2^a \leq O(\sigma_i)$ will have count $c_a$ significantly exceeding the threshold, and thus noisy count $\tilde{c}_a$ exceeding the threshold. This will be enough to crudely approximate the values $\sigma_i$ coming from large enough weight. We can ignore any component with weight less than $\alpha/k$, as even if all but one of components have such small weight, together they only contribute $\alpha$ weight. So, we can ignore them and it will only cost us $\alpha$ in total variation distance, which we can afford.

**Putting everything together:** In summary, we needed $\tilde{O}\left(\frac{k\log(1/\delta)}{\alpha\varepsilon}\right)$ samples to approximate each covariance (of sufficient weight) up to a $\text{poly}(n)$ multiplicative factor. By setting $R = \text{poly}(n)$ and using the reduction described in Section 2.1, we then need an additional $O\left(\frac{k\log n}{\alpha^2} + \frac{k\log n}{\alpha\varepsilon}\right)$ samples. If we set $n = \tilde{O}\left(\frac{k}{\alpha^2} + \frac{k\log(1/\delta)}{\alpha\varepsilon}\right)$, we will obtain that $n \geq \tilde{O}\left(\frac{k\log(1/\delta)}{\alpha\varepsilon}\right) + O\left(\frac{k\log n}{\alpha^2} + \frac{k\log n}{\alpha\varepsilon}\right)$, which is sufficient to solve our desired problem in the univariate setting.

Note that this proof relies heavily on the use of private histograms and the order of the data points in the real line. Therefore, it cannot be extended to the high-dimensional setting. We will use a completely different approach to prove Theorem 1.4.

## 2.3 Overview of Theorem 1.4 for general GMMs

As in the univariate case, we only need rough approximations of the covariances. We will learn the covariances one at a time: in each iteration, we privately identify a single covariance $\hat{\Sigma}_i$ which crudely approximates some covariance $\Sigma_i$ in the mixture, with $(\varepsilon/\sqrt{k\log(1/\delta)}, \delta/k)$-DP. Using the well-known *advanced composition* theorem (see Theorem A.1), we will get an overall privacy guarantee of $(\varepsilon,\delta)$-DP. However, to keep things simple, we will usually aim for $(\varepsilon,\delta)$-DP when learning each covariance, and we can replace $\varepsilon$ with $\varepsilon/\sqrt{k\log(1/\delta)}$ and $\delta$ with $\delta/k$ at the end.

A natural approach for parameter estimation, rather than learn $\Sigma_i$ one at a time, is to learn all of the covariances together. However, we believe that this approach would cause the sample complexity to multiply by a factor of $k$, compared to learning a single covariance. The advantage of learning the covariances one at a time is that we can apply advanced composition: this will cause the sample complexity to multiply by roughly $\sqrt{k\log(1/\delta)}$ instead.

In the rest of this subsection, our main focus is to identify a single crude approximation $\hat{\Sigma}_i$.

**Applying robustness-to-privacy:** The first main insight is to use the robustness-to-privacy conversion of Hopkins et al. [HKMN23] (see also [AUZ23]). Hopkins et al. prove a black-box (but not computationally efficient) approach that can convert robust algorithms into differentially private algorithms, using the exponential mechanism and a well-designed score function. This reduction only works for finite dimensional parameter estimation and therefore cannot be applied directly to density estimation. On the other hand, parameter estimation for arbitrary GMMs requires exponentially many samples in the number of components [MV10]. However, we will demonstrate that this lower bound does not hold when we only need a very crude estimation of the parameters.

The idea is the following. For a dataset $\mathbf{X} = \{X_1, \ldots, X_n\}$, let $\mathcal{S} = \mathcal{S}(\widetilde{\Sigma}, \mathbf{X})$ be a *score* function, which takes in a dataset $\mathbf{X}$ of size $n$ and some candidate covariance $\widetilde{\Sigma}$, and outputs some non-negative integer. At a high level, the score function $\mathcal{S}(\widetilde{\Sigma}, \mathbf{X})$ will be the smallest number of data points $t$ that we should change in $\mathbf{X}$ to get to some new data set $\mathbf{X}'$ with a specific desired property: $\mathbf{X}'$ should "look like" a sample generated from a mixture distribution with $\widetilde{\Sigma}$ being the covariance of

one of its components – namely, a component with a significant mixing weight. One can define "looks like" in different ways, and we will adjust the precise definition later. We remark that this notion of score roughly characterizes robustness, because if the samples in $\mathbf{X}$ were truly drawn from a Gaussian mixture model with covariances $\{\Sigma_i\}_{i=1}^k$, we should expect $\mathcal{S}(\Sigma_i, \mathbf{X})$ to be 0 (since $\mathbf{X}$ should already satisfy the desired property), but if we altered $k$ data points, the score should be at most $k$. The high-level choice of score function is somewhat inspired by a version of the exponential mechanism called the *inverse sensitivity mechanism* [AD20b, AD20a], though the precise way of defining the score function requires significant care and is an important contribution of this paper.

The robustness-to-privacy framework of [HKMN23], tailored to learning a single covariance, implies the following general result, which holds for any score function $\mathcal{S}$ following the blueprint above. For now, we state an informal (and slightly incorrect) version.

**Theorem 2.1** (informal - see Theorem C.3 for the formal statement). *For any $\eta \in [0, 1)$, and any dataset $\mathbf{X}$ of size $n$, define the value $V_\eta(\mathbf{X})$ to be the* volume *(i.e., Lebesgue measure) of the set of covariance matrices $\widetilde{\Sigma}$ (where the covariance can be viewed as a vector by flattening), such that $\mathcal{S}(\widetilde{\Sigma}, \mathbf{X}) \leq \eta \cdot n$.*

*Fix a parameter $\eta < 0.1$, and suppose that for* any *dataset $\mathbf{X}$ of size $n$ such that $V_{\eta/2}(\mathbf{X})$ is strictly positive, the ratio of volumes $V_\eta(\mathbf{X})/V_{\eta/2}(\mathbf{X})$ is at most some $K$ (which doesn't depend on $\mathbf{X}$). Then, if $n \geq \frac{\log K}{\varepsilon \cdot \eta}$, there is a differentially private algorithm that can find a covariance $\widetilde{\Sigma}$ of low score (i.e., where $\mathcal{S}(\widetilde{\Sigma}, \mathbf{X}) \leq \eta \cdot n$) using $n$ samples.*

Note that Theorem 2.1 does not seem to say anything about whether $\mathbf{X}$ comes from a mixture of Gaussians. However, we aim to instantiate this theorem with a score function that is carefully designed for GMMs. Recall that we want $\mathcal{S}(\widetilde{\Sigma}, \mathbf{X})$ to capture the number of points in $\mathbf{X}$ that need to be altered to make it look like a data set that was generated from a mixture, with $\widetilde{\Sigma}$ being the covariance of one of the Gaussian components. In other words, $\mathcal{S}(\widetilde{\Sigma}, \mathbf{X})$ should be small if (and hopefully only if) a "mildly corrupted" version of $\mathbf{X}$ includes a subset of points that are generated from a Gaussian with covariance $\widetilde{\Sigma}$. At the heart of designing such a score function, one needs to come up with a form of "robust Gaussianity tester" that tells whether a given set of data points are generated from a Gaussian distribution. Aside from this challenge, the volume ratio associated with the chosen score function needs to be small for every dataset $\mathbf{X}$ otherwise the above theorem would require a large $n$ (i.e., number of samples). These two challenges are, however, related. If the robust Gaussianity tester has high specificity—i.e., rejects most of the sets that are not generated from a (corrupted) Gaussian—then the volume ratio is likely to be small (i.e., a smaller number of candidates $\widetilde{\Sigma}$ would receive a good/low score).

**First Attempt:** We first try an approach which resembles that of [HKMN23] for privately learning a single Gaussian. We "define" $\mathcal{S}(\widetilde{\Sigma}, \mathbf{X})$ as the smallest integer $t$ satisfying the following property: there exists a subset $\mathbf{Y} \subset \mathbf{X}$ of size $n/k$, such that we can change $t$ data points from $\mathbf{Y}$ to get to $\mathbf{Y}'$, where $\mathbf{Y}'$ "looks like" i.i.d. samples from a Gaussian with some covariance $\Sigma$ that is "similar to" $\widetilde{\Sigma}$. The choice of $\mathbf{Y}$ having size $n/k$ is motivated by the fact that each mixture component, on average, has $n/k$ data points in $\mathbf{X}$.

The notions of "looks like" and "similar to" of course need to be formally defined. We say $\Sigma$ is "similar to" $\widetilde{\Sigma}$ (or $\Sigma \approx \widetilde{\Sigma}$) if they are spectrally close, i.e., $0.5\Sigma \preccurlyeq \widetilde{\Sigma} \preccurlyeq 2\Sigma$. We say that $\mathbf{Y}'$ "looks like" samples from a Gaussian with covariance $\Sigma$ if some covariance estimation algorithm predicts that $\mathbf{Y}'$ came from a Gaussian with covariance $\Sigma$. The choice of covariance estimation algorithm will be quite nontrivial and ends up being a key ingredient in proving Theorem 1.4.

To apply Theorem 2.1, we first set $\eta = c/k$ for some small constant $c$. We cannot set a larger value $\eta$, because if we change $t \approx n/k$ data points, we could in fact create a totally arbitrary new Gaussian component with large weight. Since there is no bound on the eigenvalues of the covariance matrix, this could cause the volume $V_\eta(\mathbf{X})$ to be infinite. The main question we must answer is how to bound the volume ratio $V_\eta(\mathbf{X})/V_{\eta/2}(\mathbf{X})$. To answer this question, we first need to understand what it means for $\mathcal{S}(\widetilde{\Sigma}, \mathbf{X}) \leq \eta \cdot n$. If $\mathcal{S}(\widetilde{\Sigma}, \mathbf{X}) \leq \eta \cdot n$, then there exists a corresponding set $\mathbf{Y} \subset \mathbf{X}$ of size $n/k$, and we can change $\eta \cdot n = c \cdot |\mathbf{Y}|$ points from $\mathbf{Y}$ to get to some $\mathbf{Y}'$ which looks like samples from a Gaussian with covariance $\Sigma \approx \widetilde{\Sigma}$. Thus, $\mathbf{Y}$ looks like $c$-corrupted samples from such a Gaussian (i.e.,

a $c$ fraction of the data is corrupted). This motivates using a *robust* covariance estimation algorithm: indeed, robust algorithms can still approximately learn $\Sigma$ even if a small constant fraction of data is corrupted, so for any $\mathbf{Y} \subset \mathbf{X}$, we expect that no matter how we change a $c$ fraction of $\mathbf{Y}$ to obtain $\mathbf{Y}'$, the robust algorithm's covariance estimate should not change much. So, for any fixed $\mathbf{Y}$, the set of possible $\Sigma$, and thus the set of possible $\widetilde{\Sigma}$, should not be that large.

In summary, to bound $V_\eta(\mathbf{X})$ versus $V_{\eta/2}(\mathbf{X})$, there are at most $\binom{n}{n/k}$ choices for $\mathbf{Y} \subset \mathbf{X}$ in the former case, and at least 1 choice in the latter case (since we assume $V_{\eta/2}(\mathbf{X}) > 0$ in Theorem 2.1). Moreover, for any such $\mathbf{Y}$, the volume of corresponding $\widetilde{\Sigma}$ should be exponential in $d^2$ (either for $V_\eta(\mathbf{X})$ or $V_{\eta/2}(\mathbf{X})$), since the dimensionality of the covariance is roughly $d^2$. So, this suggests that the overall volume ratio is at most $\binom{n}{n/k} \cdot e^{O(d^2)}$. Since $\log \binom{n}{n/k} \approx (n/k) \cdot \log k$, if we plug into Theorem 2.1 it suffices to have $n \geq \frac{(n/k)\log k + d^2}{\varepsilon \cdot (c/k)}$. Unfortunately this is impossible unless $\varepsilon \geq \log k$.

These ideas will serve as a good starting point, though we need to improve the volume ratio analysis. To do so, we also modify the robust algorithm, by strengthening the assumptions on what it means for samples to "look like" they came from a Gaussian.

**Improvement via Sample Compression:**  To improve the volume ratio, we draw inspiration from a technique called *sample compression*, which has been used in previous work on non-private density estimation for mixtures of Gaussians [ABH+18, ABDH+20]. The idea behind sample compression is that one does not need the full set $\mathbf{Y}$ to do robust covariance estimation; instead, we look at a smaller set of samples. For instance, if $\mathbf{Y} \subset \mathbf{X}$ looks like $c$-corrupted samples from a Gaussian of covariance $\Sigma \approx \widetilde{\Sigma}$, we expect that a random subset $\mathbf{Z}$ of $\mathbf{Y}$ also looks like $c$-corrupted samples from such a Gaussian. Moreover, as long as one uses $m \geq O(d)$ corrupted samples from a Gaussian, we can still (inefficiently) approximate the covariance. This motivates us to modify the robust algorithm as follows: rather than just checking whether $\mathbf{Y}$ looks like $c$-corrupted samples from a Gaussian of covariance roughly $\widetilde{\Sigma}$, we also test whether an average subset $\mathbf{Z} \subset \mathbf{Y}$ of size $m$ does as well. Therefore, if $\widetilde{\Sigma}$ has low score, there exists a corresponding set $\mathbf{Z} \subset \mathbf{X}$ of size $m$, and there are only $\binom{n}{m} \leq e^{m \log n}$ choices for $\mathbf{Z}$. So, now it suffices to have $n \geq \frac{m \log n + d^2}{\varepsilon \cdot (c/k)}$, which will give us a bound of $\tilde{O}(d^2 k/\varepsilon)$, as long as $m \leq O(d^2)$. Importantly, we still check the robust algorithm on $\mathbf{Y}$ of size roughly $n/k$, which allows us to keep the robustness threshold $\eta$ at roughly $c/k$.

There is one important caveat that for each subset $\mathbf{Z}$, there is a distinct corresponding covariance $\Sigma$, and the volume of $\widetilde{\Sigma} \approx \Sigma$ can change drastically as $\Sigma$ changes. (For instance, the volume of $\widetilde{\Sigma} \approx T \cdot \Sigma$ is $T^{\Theta(d^2)}$ times as large as the volume of $\widetilde{\Sigma} \approx \Sigma$. Since we have no bounds on the possible covariances, $T$ could be unbounded.) For our volume ratio to actually be bounded by about $\binom{n}{m} \cdot e^{O(d^2)}$, we want the volume of $\widetilde{\Sigma} \approx \Sigma$ to stay invariant with respect to $\Sigma$. This can be done by defining a "normalized volume" where the normalization is inversely proportional to the determinant (see Appendix C.1 for more details). The robustness-to-privacy conversion (Theorem 2.1) will still hold.

While the bound of $\tilde{O}(d^2 k/\varepsilon)$ seems good, we recall that this bound is merely the sample complexity for $(\varepsilon, \delta)$-DP crude approximation of a *single* Gaussian component. As discussed at the beginning of this subsection, to learn all $k$ Gaussians, we actually need $(\varepsilon/\sqrt{k \log(1/\delta)}, \delta/k)$-DP, rather than $(\varepsilon, \delta)$-DP, when crudely approximating a single component. This will still end up leading to a significant improvement over previous work [AAL24b], but we can improve the volume ratio even further and thus obtain even better sample complexity.

**Improving Dimension Dependence:**  Previously, we used the fact that the volume of candidate $\widetilde{\Sigma}$ (corresponding to a fixed $\mathbf{Z}$) was roughly exponential in $d^2$ for either $V_{\eta/2}(\mathbf{X})$ or $V_\eta(\mathbf{X})$, so the ratio should also be roughly exponential in $d^2$. Here, we improve this ratio, which will improve the overall volume ratio.

First, we return to understanding the guarantees of the robust algorithm. It is known that, given $m \geq O(d)$ samples from a Gaussian of covariance $\Sigma$, we can provide an estimate $\hat{\Sigma}$ such that $(1 - O(\sqrt{d/m}))\Sigma \preccurlyeq \hat{\Sigma} \preccurlyeq (1 - O(\sqrt{d/m}))\Sigma$. As above, we need to solve this even if a $c$ fraction of

samples are corrupted. While this can cause the relative spectral error to increase from $1 \pm O(\sqrt{d/m})$ to $1 \pm O(c + \sqrt{d/m})$, for now let us ignore the additional $c$ factor.

If $V_{\eta/2}(\mathbf{X}) > 0$, then there is some covariance $\Sigma$ and some set $\mathbf{Y}$ of size $n/k$, where the robust algorithm thinks $\mathbf{Y}$ looks like (possibly corrupted) Gaussian samples of covariance $\Sigma$. So, every $\widetilde{\Sigma}$ such that $0.5\Sigma \preccurlyeq \widetilde{\Sigma} \preccurlyeq 2\Sigma$ has score of at most $\eta/2 \cdot n$. This gives us a lower bound on $V_{\eta/2}(\mathbf{X})$. We now want to upper bound $V_\eta(\mathbf{X})$. If $\mathcal{S}(\widetilde{\Sigma}, \mathbf{X}) < \eta n$, we still have that the robust algorithm thinks some $\mathbf{Y}$ looks like Gaussian samples of covariance $\Sigma$, where $0.5\Sigma \preccurlyeq \widetilde{\Sigma} \preccurlyeq 2\Sigma$. But now, we use the additional fact that for some $\mathbf{Z} \subset \mathbf{Y}$ of size $m$, the robust algorithm on $\mathbf{Z}$ finds a covariance $\hat{\Sigma}$. By the accuracy of the robust algorithm, $\Sigma$ and $\hat{\Sigma}$ should be similar, i.e., $(1 - O(\sqrt{d/m}))\Sigma \preccurlyeq \hat{\Sigma} \preccurlyeq (1 - O(\sqrt{d/m}))\Sigma$ (where we ignored the $c$ factor). Thus, there exists some $\mathbf{Z} \subset \mathbf{X}$ of size $m$ and a $\hat{\Sigma}$ corresponding to $\mathbf{Z}$, such that $0.5(1 - O(\sqrt{d/m}))\hat{\Sigma} \preccurlyeq \widetilde{\Sigma} \preccurlyeq 2(1 + O(\sqrt{d/m}))\hat{\Sigma}$.

Therefore, from $\eta/2$ to $\eta$, we have dilated the candidate set of $\widetilde{\Sigma}$ by a factor of $1 + O(\sqrt{d/m})$ in the worst case, and we have at least 1 choice in the $\eta/2$ case but at most $\binom{n}{m}$ choices in the $\eta$ case. Thus, the overall volume ratio is at most $\binom{n}{m} \cdot (1 + O(\sqrt{d/m}))^{d^2} = e^{O(m \log n + d^2 \cdot \sqrt{d/m})}$, since the dimension of $\widetilde{\Sigma}$ is roughly $d^2$. Consequently, it now suffices to have $n \geq \frac{m \log n + d^2 \cdot \sqrt{d/m}}{\varepsilon \cdot (c/k)}$ : setting $m = d^{5/3}$ gives us an improved bound of $\tilde{O}(d^{5/3}k/\varepsilon)$ for learning a single $\Sigma_i$.

There are some issues with this approach: most notably, we ignored the fact that the spectral error is really $c + \sqrt{d/m}$ rather than $\sqrt{d/m}$. However, the robust algorithm can do better than just estimating up to spectral error $c + \sqrt{d/m}$: it can also get an improved *Frobenius* error. While we will not formally state the guarantees on the robust algorithm here (see Theorem B.3 for the formal statement), the main high-level observation is that if the robust estimator $\hat{\Sigma}$ can be $1 \pm c$ times as large as the true covariance $\Sigma$ in only $O(1)$ directions then for an average direction the ratio of $\hat{\Sigma}$ to $\Sigma$ will be $1 \pm O(\sqrt{d/m})$. We can utilize this observation to bound the volume ratio, using some careful $\varepsilon$-net arguments (this is executed in Appendix C.3). Our dimension dependence of $d^{5/3}$ will increase to $d^{7/4}$, though this still improves over the previous $d^2$ bound.

We will formally define the score function $\mathcal{S}(\widetilde{\Sigma}, \mathbf{X})$ in Appendix E.1 and fully analyze the application of the robustness-to-privacy conversion, as outlined here, in Appendix E.2.

**Accuracy:** One important final step that we have so far neglected is ensuring that any $\widetilde{\Sigma}$ of low score must be crudely close to some $\Sigma_i$, if $\mathbf{X}$ is actually drawn from a GMM. We will just focus on the case where $\mathcal{S}(\widetilde{\Sigma}, \mathbf{X}) = 0$, so some $\mathbf{Y} \subset \mathbf{X}$ of size $n/k$ looks like a set of samples from a Gaussian with covariance $\widetilde{\Sigma}$. If the samples $\mathbf{Y}$ all came from the $i$-th component of the GMM, then it will not be difficult to show $\widetilde{\Sigma}$ is similar to $\Sigma_i$. The more difficult case is when data point in $\mathbf{Y}$ are generated from several components.

However, if $n \geq 20k^2d$, then $|\mathbf{Y}| \geq 20kd$, which means that by the Pigeonhole Principle, at least $20d$ points in $\mathbf{Y}$ come from the same mixture component $(\mu_i, \Sigma_i)$. We are able to prove that, with high probability over samples drawn from a single Gaussian component $\mathcal{N}(\mu_i, \Sigma_i)$, that the empirical covariance of any subset of size at least $20d$ is crudely close to $\Sigma_i$ (see Corollary B.5). As a result, when verifying that a subset $\mathbf{Y}$ "looks like" i.i.d. Gaussian samples with covariance $\Sigma$, we can ensure that the empirical covariance of every subset $\mathbf{Z} \subset \mathbf{Y}$ of size $20d$ is crudely close to $\Sigma$. Thus, if the score $\mathcal{S}(\widetilde{\Sigma}, \mathbf{X}) = 0$, $\widetilde{\Sigma}$ is close to $\Sigma$, which is crudely close to some $\Sigma_i$.

We also formally analyze the accuracy in Appendix E.2.

**Putting everything together:** To crudely learn a single Gaussian component with $(\varepsilon, \delta)$-DP, we will need $n \geq \tilde{O}(d^{7/4}k/\varepsilon)$ samples to find some covariance $\widetilde{\Sigma}$ with low score, and we also need $n \geq O(k^2d)$ so that a covariance $\widetilde{\Sigma}$ of low score is actually a crude approximation to one of the real mixture components. To crudely approximate all components, we learn each Gaussian component with $(\varepsilon/\sqrt{k \log(1/\delta)}, \delta/k)$-DP. The advanced composition theorem will imply that repeating this procedure $k$ times on the same data (to learn all $k$ components) will be $(\varepsilon, \delta)$-DP. Hence, by replacing

$\varepsilon$ with $\varepsilon/\sqrt{k\log(1/\delta)}$, we get that it suffices for $n \geq \tilde{O}\left(\frac{d^{7/4}k^{3/2}\sqrt{\log(1/\delta)}}{\varepsilon} + k^2 d\right)$, if we need to crudely learn all of the covariances. Finally, we can apply the private hypothesis selection technique (recall Section 2.1), which requires an additional $\tilde{O}\left(\frac{d^2 k}{\alpha^2} + \frac{d^2 k}{\alpha\varepsilon}\right)$. Combining these terms will give the final sample complexity.

We remark that the sample complexity obtained above is actually better than the complexity in Theorem 1.4. There are two reasons for this. The first is that we have been assuming each component has weight $1/k$, meaning it contributes about $n/k$ data points. In reality, the weights may be arbitrary and thus some components may have much fewer data points. However, it turns out that one can actually ignore any component with less than $\alpha/k$ weight, if we want to solve density estimation up to total variation distance $\alpha$. This will multiply the sample complexity terms $\tilde{O}\left(\frac{d^{7/4}k^{3/2}\sqrt{\log(1/\delta)}}{\varepsilon} + k^2 d\right)$, needed for crude approximation, by a factor of $1/\alpha$. Finally, the informal Theorem 2.1 is slightly inaccurate, and the accurate version of the theorem will end up adding the additional term of $\frac{k^{3/2}\log^{3/2}(1/\delta)}{\alpha\varepsilon}$. Along with the final term $\tilde{O}\left(\frac{d^2 k}{\alpha^2} + \frac{d^2 k}{\alpha\varepsilon}\right)$ from the private hypothesis selection, these terms will exactly match Theorem 1.4.

### 2.4 Roadmap

In Appendix A, we note some general preliminary results. In Appendix B, we note some additional preliminaries on robust learning of a single Gaussian. In Appendix C, we discuss the robustness-to-privacy conversion and prove some volume arguments needed for Theorem 1.4. In Appendix D, we explain how to reduce to the crude approximation setting, using private hypothesis selection. In Appendix E, we design and analyze the algorithm for multivariate Gaussians, and prove Theorem 1.4. In Appendix F, we design and analyze the algorithm for univariate Gaussians, and prove Theorem 1.5. In Appendix G, we prove Theorem 1.6. Finally, Appendix H proves some auxiliary results that we state in Appendices B and C.

## Limitations

Our results are on theoretical guarantees on the sample complexity of privately learning Mixtures of Gaussians. We do not provide any efficient or practical algorithms, and focus on statistical guarantees. We also do not discuss how to set the parameters and accuracy guarantees for practical applications, this is a question best left to practitioners. Finally, we assume each sample is i.i.d. drawn from a Gaussian Mixture Model distribution, though we remark that we use a "robustness-to-privacy" framework that will automatically make our algorithm robust to a roughly $\alpha/k$ fraction of corruptions.

**Acknowledgements.** Hassan Ashtiani was supported by an NSERC Discovery grant. Shyam Narayanan was supported by an NSF Graduate Fellowship and a Google Fellowship.

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

# A   Preliminaries

## A.1   Differential Privacy

We state the *advanced composition* theorem of differential privacy.

**Theorem A.1** (Advanced Composition Theorem [DR14, Theorem 3.20]). *Let $\varepsilon, \delta, \delta' \geq 0$ be arbitrary parameters. Let $\mathcal{A}_1, \ldots, \mathcal{A}_k$ be algorithms on a dataset $\mathbf{X}$, where each $\mathcal{A}_i$ is $(\varepsilon, \delta)$-DP. Then, the concatenation $\mathcal{A}(\mathbf{X}) = (\mathcal{A}_1(\mathbf{X}), \ldots, \mathcal{A}_k(\mathbf{X}))$ is $\left( \sqrt{2k \log \frac{1}{\delta'}} \cdot \varepsilon + k\varepsilon \cdot (e^\varepsilon - 1), k \cdot \delta + \delta' \right)$-DP.*

*Moreover, this holds even if the algorithms $\mathcal{A}_i$ are adaptive. By this, we mean that for all $i \geq 2$ the algorithm $\mathcal{A}_i$ is allowed to depend on $\mathcal{A}_1(\mathbf{X}), \ldots, \mathcal{A}_{i-1}(\mathbf{X})$. However, privacy must still hold for $\mathcal{A}_i$, conditioned on the previous outputs $\mathcal{A}_1(\mathbf{X}), \ldots, \mathcal{A}_{i-1}(\mathbf{X})$.*

Next, we note the properties of the *Truncated Laplace* distribution and mechanism.

**Definition A.2** (Truncated Laplace Distribution). For $\Delta, \varepsilon, \delta > 0$, the *Truncated Laplace Distribution* $\mathrm{TLap}(\Delta, \varepsilon, \delta)$ is the distribution with PDF proportional to $e^{-|x| \cdot \varepsilon / \Delta}$ on the region $[-A, A]$, where $A = \frac{\Delta}{\varepsilon} \cdot \log \left( 1 + \frac{e^\varepsilon - 1}{2\delta} \right)$, and PDF $0$ outside the region $[-A, A]$.

**Lemma A.3** (Truncated Laplace Mechanism [GDGK20, Theorem 1]). *Let $f : \mathcal{X}^n \to \mathbb{R}$ be a real-valued function, and let $\Delta > 0$, such that for any neighboring datasets $\mathbf{X}, \mathbf{X}', |f(\mathbf{X}) - f(\mathbf{X}')| \leq \Delta$. Then, the mechanism $\mathcal{A}$ that outputs $f(\mathbf{X}) + \mathrm{TLap}(\Delta, \varepsilon, \delta)$ is $(\varepsilon, \delta)$-DP.*

## A.2   Matrix and Concentration Bounds

In this section, we note some standard but useful lemmas. We first note the Courant-Fischer theorem.

**Theorem A.4** (Courant-Fischer). *Let $A \in \mathbb{R}^{d \times d}$ be a real symmetric matrix, with eigenvalues $\lambda_1 \geq \lambda_2 \geq \cdots \geq \lambda_d$. Then, for each $k$,*

$$\lambda_k = \max_{\substack{V \subset \mathbb{R}^d \\ \dim(V) = k}} \min_{\substack{x \in V \\ \|x\|_2 = 1}} x^\top A x = \min_{\substack{V \subset \mathbb{R}^d \\ \dim(V) = d-k+1}} \max_{\substack{x \in V \\ \|x\|_2 = 1}} x^\top A x,$$

*where $V$ refers to a linear subspace of $\mathbb{R}^d$.*

By setting $A = J^\top J$, we have the following corollary.

**Corollary A.5.** *Let $J \in \mathbb{R}^{d \times d}$ be a real (possibly asymmetric) matrix with singular values $\sigma_1 \geq \sigma_2 \geq \cdots \geq \sigma_d$. Then, for each $k$,*

$$\sigma_k = \max_{\substack{V \subset \mathbb{R}^d \\ \dim(V) = k}} \min_{\substack{x \in V \\ \|x\|_2 = 1}} \|Jx\|_2 = \min_{\substack{V \subset \mathbb{R}^d \\ \dim(V) = d-k+1}} \max_{\substack{x \in V \\ \|x\|_2 = 1}} \|Jx\|_2.$$

Next, we note a basic proposition.

**Proposition A.6.** *[HKMN23, Lemma 6.8] Let $M \in \mathbb{R}^{d \times d}$ be a real symmetric matrix, and $J \in \mathbb{R}^{d \times d}$ be any real-valued (but not necessarily symmetric) matrix such that $\|JJ^\top - I\|_{op} \leq \phi \leq 1$, for some parameter $\phi$. Then, $\|J^\top M J\|_F^2 \leq (1 + 3\phi) \cdot \|M\|_F^2$.*

We also note the Hanson-Wright inequality.

**Lemma A.7** (Hanson-Wright). *Given a $d \times d$ matrix $M \in \mathbb{R}^{d \times d}$ and a $d$-dimensional Gaussian vector $X \sim \mathcal{N}(0, I)$, for any $t \geq 0$,*

$$\mathbb{P}\left( \left| X^\top M X - \mathbb{E}[X^\top M X] \right| \geq t \right) \leq 2 \exp \left( -c \min \left( \frac{t^2}{\|M\|_F^2}, \frac{t}{\|M\|_{op}} \right) \right),$$

*for some universal constant $c > 0$.*

Finally, we note a folklore simple characterization of total variation distance (see also [AAL23, Theorem 1.8]).

**Lemma A.8.** *For any $\mu_1, \mu_2 \in \mathbb{R}^d$ and positive definite $\Sigma_1, \Sigma_2 \in \mathbb{R}^{d \times d}$,*

$$d_{\mathrm{TV}}(\mathcal{N}(\mu_1, \Sigma_1), \mathcal{N}(\mu_2, \Sigma_2)) \leq \frac{1}{\sqrt{2}} \cdot \max \left( \|\Sigma_1^{-1/2} \Sigma_2 \Sigma_1^{-1/2} - I\|_F, \|\Sigma_1^{-1/2}(\mu_1 - \mu_2)\|_2 \right).$$

# B Robust Estimation of a Single Gaussian

In this section, we establish what a robust algorithm can do for learning a single high-dimensional Gaussian. We will state the accuracy guarantees in terms of the following definition, which combines Mahalanobis distance, spectral distance, and Frobenius distance.

**Definition B.1.** Given mean-covariance pairs $(\mu, \Sigma)$ and $(\hat{\mu}, \hat{\Sigma})$, and given parameters $\gamma, \rho, \tau > 0$, we say that $(\hat{\mu}, \hat{\Sigma}) \approx_{\gamma, \rho, \tau} (\mu, \Sigma)$ if

- $\|\Sigma^{-1/2} \hat{\Sigma} \Sigma^{-1/2} - I\|_{op} \leq \gamma$. (Spectral distance)

- $\|\Sigma^{-1/2} \hat{\Sigma} \Sigma^{-1/2} - I\|_F \leq \rho$. (Frobenius distance)

- $\|\Sigma^{-1/2}(\hat{\mu} - \mu)\|_2 \leq \tau$. (Mahalanobis distance)

We will also use $\approx_{\gamma, \tau}$ as a shorthand for $\approx_{\gamma, \sqrt{d} \cdot \gamma, \tau}$, i.e., there is no additional Frobenius norm condition beyond what is already imposed by the operator norm condition.

Although $\approx_{\gamma, \rho, \tau}$ is neither symmetric nor transitive, symmetry and transitivity happen in an approximate sense. Namely, the following result holds: we defer the proof to Appendix H. We remark that similar results are known (e.g., [AL22, Lemma 3.2], [AAL23, Lemma 4.1]).

**Proposition B.2** (Approximate Symmetry and Transitivity). *Fix any $\gamma, \rho, \tau > 0$ such that $\gamma \leq 0.1$. Then, for any $(\mu_1, \Sigma_1) \approx_{\gamma, \rho, \tau} (\mu_2, \Sigma_2)$, we have that $(\mu_2, \Sigma_2) \approx_{2\gamma, 2\rho, 2\tau} (\mu_1, \Sigma_1)$, and for any $(\mu_1, \Sigma_1) \approx_{\gamma, \rho, \tau} (\mu_2, \Sigma_2)$ and $(\mu_2, \Sigma_2) \approx_{\gamma, \rho, \tau} (\mu_3, \Sigma_3)$, we have that $(\mu_1, \Sigma_1) \approx_{4\gamma, 4\rho, 4\tau} (\mu_3, \Sigma_3)$.*

We note the following theorem about the accuracy of robustly learning a single Gaussian. While this result will roughly follow from known results and techniques in the literature, we were unable to find a formal statement of the theorem, so we prove it in Appendix H.

**Theorem B.3.** *For some universal constants $c_0 \in (0, 0.01)$ and $C_0 > 1$, the following holds. Fix any $\eta \leq \gamma \leq c_0$, $\beta < 1$, and $\rho$ such that $\widetilde{O}(\eta) \leq \rho \leq c_0 \sqrt{d}$. Let $n \geq \widetilde{O}\left(\frac{d + \log(1/\beta)}{\gamma^2} + \frac{(d + \log(1/\beta))^2}{\rho^2}\right)$. Then, there exists an (inefficient, deterministic) algorithm $\mathcal{A}_0$ with the following property. For any $\mu, \Sigma$, with probability at least $1 - \beta$ over $\mathbf{X} = \{X_1, \ldots, X_n\} \sim \mathcal{N}(\mu, \Sigma)$, and for all datasets $\mathbf{X}'$ with $\mathrm{d}_H(\mathbf{X}, \mathbf{X}') \leq \eta \cdot n$, we have*

$$\mathcal{A}_0(\mathbf{X}') \approx_{C_0 \gamma, C_0 \rho, C_0 \gamma} (\mu, \Sigma).$$

*We remark that $\mathcal{A}_0$ may have knowledge of $\eta, \gamma, \rho, \beta$, but does not have knowledge of $\mu, \Sigma$, or the uncorrupted data $\mathbf{X}$.*

## B.1 Directional bounds

In this section, we note that, when given i.i.d. samples from a Gaussian, with high probability one cannot choose a reasonably large subset with an extremely different empirical mean or covariance.

First, we note the following proposition, for which the proof follows by an $\varepsilon$-net type argument.

**Proposition B.4.** *Let $n \geq n' \geq 20d$, and $L \geq C_1 n^2$ for a sufficiently large constant $C_1$. Suppose that some data points $\mathbf{X} = \{X_1, \ldots, X_n\}$ are i.i.d. sampled from $\mathcal{N}(\mathbf{0}, I)$. Then, with probability at least $1 - n^{-\Omega(n')}$, the following both hold.*

1. *For all $X_i \in \mathbf{X}$, $\|X_i\|_2 \leq L$.*

2. *For all subsets $\mathbf{Y} \subset \mathbf{X}$ of size $n'$, there does not exist any real number $r$ and any unit vector $v$ such that $|\langle X_i, v \rangle - r| \leq \frac{1}{L}$ for all $X_i \in \mathbf{Y}$.*

*Proof.* If $\|X_i\|_2 \geq L$, then $X^\top M X \geq L^2$ for $M$ the $d \times d$ identity matrix $I$. However, $\mathbb{E}[X^\top M X] = d$. So, by Lemma A.7, the probability of this event is at most $2e^{-c \cdot \min(n^4/d, n^2)} \leq 2e^{-c \cdot n^2} \leq n^{-\Omega(n')}$.

Next, we bound the probability that the first item holds but the second item doesn't hold. First, we can create a $\frac{1}{L^2}$-net of unit vectors $v'$, of size $(3L^2)^d \leq L^{3d}$, and a $\frac{1}{L}$-net of real numbers $r'$

from $[-L, L]$, of size $2L^2$. Now, suppose that the first item holds, but there is some unit vector $v$ and real number $r$ with $|\langle X_i, v \rangle - r| \leq \frac{1}{L}$ for all $X_i \in \mathbf{Y}$, for some $\mathbf{Y} \subset \mathbf{X}$ of size $n'$. In this case, $\langle X_i, v \rangle \leq \|X_i\|_2 \leq L$ for all $X_i \in \mathbf{X}$, so we can assume that $|r| \leq L$. Thus, if $v'$ is the closest unit vector to $v$ in the net and $r'$ is the closest real number to $r$ in the net, then $|\langle X_i, v' \rangle - r'| \leq |\langle X_i, v \rangle - r| + |\langle X_i, v' - v \rangle| + |r - r'| \leq \frac{1}{L} + \|X_i\|_2 \cdot \frac{1}{L^2} + \frac{1}{L} \leq \frac{3}{L}$.

In other words, if the first event holds and the second does not, there exists $v', r'$ in the net and $\mathbf{Y} \subset \mathbf{X}$ of size $n'$, such that $|\langle X_i, v' \rangle - r'| \leq \frac{3}{L}$ for all $X_i \in \mathbf{Y}$. We can now bound this via a union bound. To perform the union bound, we first bound the probability of this event holding for some fixed $v', r', \mathbf{Y}$.

For any fixed $v', r', \mathbf{Y}$, note that $\langle X_i, v' \rangle_{X_i \in \mathbf{Y}}$ is just $n'$ i.i.d. copies of $\mathcal{N}(0,1)$. Let's call these values $z_1, \ldots, z_{n'}$. If there is some $r'$ such that $|z_i - r'| \leq \frac{3}{L}$ for all $i$, then $|z_i - z_1| \leq \frac{6}{L}$ for all $i$. Since the PDF of a $\mathcal{N}(0,1)$ is uniformly bounded by at most $\frac{1}{2}$, for any fixed $z_1$, the probablity that any other $z_i$ is within $\frac{6}{L}$ of $z_1$ is at most $\frac{6}{L}$, so the overall probability is at most $(6/L)^{n'-1}$.

Now, the union bound is done over $\binom{n}{n'}$ choices of $\mathbf{Y}$, at most $L^{3d}$ choices of $v'$, and at most $2L^2$ choices of $\mathbf{Y}$. Thus, the overall probability is at most $(6/L)^{n'-1} \cdot n^{n'} \cdot L^{3d} \cdot 2L^2 \leq (6n)^{n'}/L^{n'-3-3d} \geq (6n/L^{0.7})^{n'}$. Thus, as long as $L \geq 100n^2$, this is much smaller than $n^{-\Omega(n')}$. $\qquad\square$

By shifting and scaling appropriately, we have the following corollary.

**Corollary B.5.** *Let $n \geq n' \geq 20d$, and $L \geq C_1 n^2$, for $C_1$ the same constant as in Proposition B.4. Suppose that some data points $\mathbf{X} = \{X_1, \ldots, X_n\}$ are drawn from $\mathcal{N}(\mu, \Sigma)$. Then, with probability at least $1 - n^{-\Omega(n')}$, the following both hold.*

1. *For all $X_i \in \mathbf{X}$, $\|\Sigma^{-1/2}(X_i - \mu)\|_2 \leq L$.*

2. *For all subsets $\mathbf{Y} \subset \mathbf{X}$ of size $n'$, there does not exist any real number $r$ and any nonzero vector $v$ such that $|\langle X_i, v \rangle - r| \leq \frac{1}{L} \|\Sigma^{1/2} v\|_2$ for all $X_i \in \mathbf{Y}$.*

## B.2 Modified robust algorithm

We can modify the robust algorithm of Theorem B.3 to have the following guarantees.

**Lemma B.6.** *Let $c_0, C_0$ be as in Theorem B.3, and $C_1$ be as in Proposition B.4. Let $L := C_1 \cdot n^2$. Also, fix any $\eta \leq \gamma \leq c_0$, any $e^{-d} \leq \beta < 1$, and and any $\rho$ such that $\widetilde{O}(\eta) \leq \rho \leq c_0 \sqrt{d}$. Finally, suppose $n \geq m \geq \widetilde{O}\left(\frac{d}{\gamma^2} + \frac{d^2}{\rho^2}\right)$.*

*Then, there exists an (inefficient, deterministic) algorithm $\mathcal{A}$ that, on a dataset $\mathbf{Y}$ of size $m$, outputs either some $(\hat{\mu}, \hat{\Sigma})$ or $\perp$, with the following properties.*

1. *For any datasets $\mathbf{Y}, \mathbf{Y}'$ with $d_{\mathrm{H}}(\mathbf{Y}, \mathbf{Y}') \leq \eta \cdot m$, if $\mathcal{A}(\mathbf{Y}) = (\hat{\mu}, \hat{\Sigma}) \neq \perp$ and $\mathcal{A}(\mathbf{Y}') = (\hat{\mu}', \hat{\Sigma}') \neq \perp$, then*

$$(\hat{\mu}', \hat{\Sigma}') \approx_{8C_0\gamma, 8C_0\rho, 8C_0\gamma} (\hat{\mu}, \hat{\Sigma}).$$

2. *For any fixed $\mu, \Sigma$, with probability at least $1 - O(\beta)$ over $Y_1, \ldots, Y_m \sim \mathcal{N}(\mu, \Sigma)$, $\mathcal{A}(\mathbf{Y}) \neq \perp$ and*

$$\mathcal{A}(\mathbf{Y}) \approx_{C_0\gamma, C_0\rho, C_0\gamma} (\mu, \Sigma).$$

3. *Fix an integer $k \geq 1$, and additionally suppose that $m \geq 40d \cdot k$. Suppose that $\mathbf{X} = \{X_1, \ldots, X_n\}$ are drawn i.i.d. from some mixture of $k$ Gaussians $(\mu_i, \Sigma_i)$. Then, with probability at least $1 - O(\beta)$ over $\mathbf{X}$, for every $\mathbf{Y}' \subset \mathbf{X}$ of size $m$ and every $\mathbf{Y}$ with $d_{\mathrm{H}}(\mathbf{Y}, \mathbf{Y}') \leq m/2$, either $\mathcal{A}(\mathbf{Y}) = \perp$, or $\mathcal{A}(\mathbf{Y}) = (\hat{\mu}, \hat{\Sigma})$, where there is some $i \in [k]$ such that $\frac{1}{9L^4} \cdot \Sigma_i \preccurlyeq \hat{\Sigma} \preccurlyeq 9L^4 \cdot \Sigma_i$ and $\|\Sigma_i^{-1/2}(\hat{\mu} - \mu_i)\|_2 \leq 10L^3$.*

*As in Theorem B.3, $\mathcal{A}$ has knowledge of $\eta, \gamma, \rho, \beta$, but not $\mu$ or $\Sigma$.*

*Proof.* The algorithm $\mathcal{A}$ works as follows. Start off by computing $\mathcal{A}_0(\mathbf{Y}) = (\hat{\mu}, \hat{\Sigma})$, where $\mathcal{A}_0$ is the algorithm of Theorem B.3. We can run the algorithm on any arbitrary dataset, even if it didn't come from Gaussian samples, and we can assume that the algorithm $\mathcal{A}_0$ always outputs some mean-covariance pair (for instance, by having it output $(\mathbf{0}, I)$ by default instead of $\perp$). Next, $\mathcal{A}$, on a dataset $\mathbf{Y}$, checks if any of the following three conditions hold.

    (a) There exists $\mathbf{Y}'$ such that $\mathrm{d_H}(\mathbf{Y}, \mathbf{Y}') \leq \eta \cdot m$, and $\mathcal{A}_0(\mathbf{Y}') \not\approx_{8C_0\gamma, 8C_0\rho, 8C_0\gamma} \mathcal{A}_0(\mathbf{Y})$.

    (b) There exists $Y_i \in \mathbf{Y}$ such that $\|\hat{\Sigma}^{-1/2}(Y_i - \hat{\mu})\|_2 > 3L$.

    (c) There exists a subset $\mathbf{Z} \subset \mathbf{Y}$ of size $20d$, and a unit vector $v$ and real number $r$, such that for all $Y_i \in \mathbf{Z}$, $|\langle Y_i, v \rangle - r| < \frac{1}{2L} \cdot \|\hat{\Sigma}^{1/2}v\|_2$.

If any of these conditions hold, the output is $\mathcal{A}(\mathbf{Y}) = \perp$. Otherwise, the output is $\mathcal{A}(\mathbf{Y}) = \mathcal{A}_0(\mathbf{Y})$.

We now verify that the algorithm satisfies the required properties. Property 1 clearly holds, because if there exist datasets $\mathbf{Y}, \mathbf{Y}'$ with $\mathrm{d_H}(\mathbf{Y}, \mathbf{Y}') \leq \eta \cdot m$, $\mathcal{A}(\mathbf{Y}) \neq \perp$ and $\mathcal{A}(\mathbf{Y}') \neq \perp$, and $\mathcal{A}(\mathbf{Y}) \not\approx_{8C_0\gamma, 8C_0\rho, 8C_0\gamma} \mathcal{A}(\mathbf{Y}')$, then we would have in fact set $\mathcal{A}(\mathbf{Y})$ to $\perp$.

For Property 2, note that with probability $1 - O(\beta)$, $\mathcal{A}_0(\mathbf{Y})$ satisfies the desired property by Theorem B.3, since $\beta \geq e^{-d}$ so $d + \log 1/\beta = O(d)$. So, we just need to make sure that we don't set $\mathcal{A}(\mathbf{Y})$ to be $\perp$. Since $Y_1, \ldots, Y_m \sim \mathcal{N}(\mu, \Sigma)$, then with probability $1 - O(\beta)$, both $\mathbf{Y}$ and any $\mathbf{Y}'$ with $\mathrm{d_H}(\mathbf{Y}, \mathbf{Y}') \leq \eta \cdot n$ satisfy the conditions of Theorem B.3. In other words, $\mathcal{A}_0(\mathbf{Y}) \approx_{C_0\gamma, C_0\rho, C_0\gamma}$ and $\mathcal{A}_0(\mathbf{Y}') \approx_{C_0\gamma, C_0\rho, C_0\gamma}$. So, by Proposition B.2, $\mathcal{A}_0(\mathbf{Y}') \approx_{8C_0\gamma, 8C_0\rho, 8C_0\gamma} \mathcal{A}_0(\mathbf{Y})$ for any $\mathbf{Y}'$ with $\mathrm{d_H}(\mathbf{Y}, \mathbf{Y}') \leq \eta \cdot n$. Thus, condition a) is not met.

Moreover, by Corollary B.5, with failure probability at most $m^{-\Omega(20d)} \leq \beta$, $\|\Sigma^{-1/2}(Y_i - \mu)\|_2 \leq L$ for all $Y_i \in \mathbf{Y}$, and for all $\mathbf{Z} \subset \mathbf{Y}$ of size $20d$, there does not exist a real $r$ and a nonzero vector $v$ such that $|\langle Y_i, v \rangle - r| \leq \frac{1}{L} \cdot \|\Sigma^{1/2}v\|_2$ for all $Y_i \in \mathbf{Y}$. Now, we know that by Theorem B.3, $(\hat{\mu}, \hat{\Sigma}) \approx_{0.5, 0.5} (\mu, \Sigma)$. So,

$$\|\hat{\Sigma}^{-1/2}(Y_i - \hat{\mu})\|_2 \leq 2\|\Sigma^{-1/2}(Y_i - \hat{\mu})\|_2 \leq 2(\|\Sigma^{-1/2}(Y_i - \mu)\| + 0.5) = 2L + 1 \leq 3L,$$

and if $|\langle Y_i, v \rangle - r| \leq \frac{1}{2L} \cdot \|\hat{\Sigma}^{1/2}v\|_2$, then $|\langle Y_i, v \rangle - r| \leq \frac{1}{L} \cdot \|\Sigma^{1/2}v\|_2$, for all $Y_i \in \mathbf{Y}$. Thus, conditions b) and c) are also not met, so Property 2 holds.

For Property 3, note that if $m \geq 40d \cdot k$, then $|\mathbf{Y} \cap \mathbf{X}| \geq 20d \cdot k$, i.e., $\mathbf{Y}$ contains at least $20dk$ uncorrupted points. So, by the Pigeonhole Principle, for every possible $\mathbf{Y}$, there is some index $i \in [k]$ such that at least $20d$ points in $\mathbf{Y} \cap \mathbf{X}$ come from the $i^{\text{th}}$ Gaussian component.

Now, let us condition on the event that Corollary B.5 holds for $\mathbf{X}$, where $n' = 20d$, which happens with at least $1 - n^{-\Omega(n')} \geq 1 - \beta$ probability. We claim that for every possible $\mathbf{Y}$, and for an $i \in [k]$ such that $20d$ points $\mathbf{Z} \subset \mathbf{Y} \cap \mathbf{X}$ came from the $i^{\text{th}}$ Gaussian component, then either $\mathcal{A}(\mathbf{Y}) = \perp$ or if $\mathcal{A}(\mathbf{Y}) = (\hat{\mu}, \hat{\Sigma})$ then $\frac{1}{9L^4} \cdot \Sigma_i \preccurlyeq \hat{\Sigma} \preccurlyeq 9L^4 \cdot \Sigma_i$ and $\|\Sigma_i^{-1/2}(\hat{\mu} - \mu_i)\|_2 \leq 10L^3$.

First, we verify that $\frac{1}{9L^4} \cdot \Sigma_i \preccurlyeq \hat{\Sigma} \preccurlyeq 9L^4 \cdot \Sigma_i$. Otherwise, there exists a unit vector $v$ such that either $v^\top \hat{\Sigma} v \geq 9L^4 \cdot v^\top \Sigma_i v$ or $v^\top \hat{\Sigma} v \leq \frac{1}{9L^4} \cdot v^\top \Sigma_i v$. In the former case, by Part 1 of Corollary B.5, for all $Y_i \in \mathbf{Z}$, $\|\Sigma_i^{-1/2}(Y_i - \mu_i)\|_2 \leq L$, so

$$|\langle v, Y_i - \mu_i \rangle| = |\langle \Sigma_i^{1/2}v, \Sigma_i^{-1/2}(Y_i - \mu_i)\rangle| \leq L \cdot \|\Sigma_i^{1/2}v\|_2 = L \cdot \sqrt{v^\top \Sigma_i v} \leq L \cdot \sqrt{\frac{v^\top \hat{\Sigma} v}{9L^4}} = \frac{1}{3L} \cdot \|\hat{\Sigma}^{1/2}v\|_2.$$

Thus, if we set $r = \langle v, \mu_i \rangle$, then $|\langle Y_i, v \rangle - r| < \frac{1}{2L} \cdot \|\hat{\Sigma}^{1/2}v\|_2$ for all $Y_i \in \mathbf{Z}$, so the algorithm $\mathcal{A}$ would have output $\perp$ due to condition c). Alternatively, if there exists a unit vector $v$ such that $v^\top \hat{\Sigma} v \leq \frac{1}{9L^4} \cdot v^\top \Sigma_i v$, then by condition b), $\|\hat{\Sigma}^{-1/2}(Y_i - \hat{\mu})\|_2 \leq 3L$ for all $Y_i \in \mathbf{Z}$, so

$$|\langle v, Y_i - \hat{\mu} \rangle| = |\langle \hat{\Sigma}^{1/2}v, \hat{\Sigma}^{-1/2}(Y_i - \mu)\rangle| \leq 3L \cdot \|\hat{\Sigma}^{1/2}v\|_2 \leq \frac{1}{L} \cdot \|\Sigma_i^{1/2}v\|_2.$$

So, there exists $r = \langle v, \hat{\mu} \rangle$ such that $|\langle v, Y_i \rangle - r| \leq \frac{1}{L} \cdot \sqrt{v^\top \Sigma_i v}$ which is impossible by Part 2 of Corollary B.5

Next, we verify that $\|\Sigma_i^{-1/2}(\hat{\mu} - \mu_i)\|_2 \le 10L^3$. By Triangle inequality, for every $Y_i \in \mathbf{Z}$, $\|\Sigma_i^{-1/2}(Y_i - \hat{\mu})\|_2 + \|\Sigma_i^{-1/2}(Y_i - \mu_i)\|_2 \ge \|\Sigma_i^{-1/2}(\mu_i - \hat{\mu})\|_2$. But, $\|\Sigma_i^{-1/2}(Y_i - \mu_i)\|_2 \le L$ by Part 1 of Corollary B.5, and $\|\Sigma_i^{-1/2}(Y_i - \hat{\mu})\|_2 \le 3L^2 \cdot \|\hat{\Sigma}^{-1/2}(Y_i - \hat{\mu})\|_2 \le 3L^2 \cdot 3L \le 9L^3$, because we just proved that $9L^4\Sigma_i \succcurlyeq \hat{\Sigma}$ and assuming we do not reject due to condition b). Thus, $\|\Sigma_i^{-1/2}(\mu_i - \hat{\mu})\|_2 \le 9L^3 + L \le 10L^3$. $\qquad\square$

## C  Volume and the Robustness-to-Privacy Conversion

In this section, we explain the robustness-to-privacy conversion [HKMN23] that we will utilize in proving Theorem 1.4. We will also need some relevant results about computing *volume*, which will be important in the robustness-to-privacy conversion.

### C.1  Normalized Volume

Given a pair $(\mu, \Sigma)$, where $\mu \in \mathbb{R}^d$ and $\Sigma \in \mathbb{R}^{d \times d}$ is positive definite, we will define $\mathrm{Proj}(\mu, \Sigma) \in \mathbb{R}^{d \cdot (d+3)/2}$ to represent the coordinates of $\mu$ along with the upper-diagonal coordinates of $\Sigma$. For any set $\Omega$ of mean-covariance pairs $(\mu, \Sigma)$, define $\mathrm{Proj}(\Omega) := \{\mathrm{Proj}(\mu, \Sigma) : (\mu, \Sigma) \in \Omega\}$. Because $\Sigma$ is symmetric, $\mathrm{Proj}(\mu, \Sigma)$ fully encodes the information about $\mu$ and $\Sigma$. We also define $\mathrm{vol}(\Omega)$ to be the Lebesgue measure of $\mathrm{Proj}(\Omega)$, i.e., the $\frac{d(d+3)}{2}$-dimensional measure of all points $\mathrm{Proj}(\mu, \Sigma)$ for $(\mu, \Sigma) \in \Omega$. Next, we will define the normalized volume

$$\mathrm{vol_n}(\Omega) := \int_{\theta \in \mathrm{Proj}(\Omega)} \frac{1}{(\det \Sigma)^{(d+2)/2}} d\theta, \tag{1}$$

where $\theta = \mathrm{Proj}(\mu, \Sigma)$, and we take a Lebesgue integral over $\theta \in \mathrm{Proj}(\Omega)$.

To motivate this choice of normalized volume, we will see that the volume is invariant under some basic transformations.

Given a function $f : \mathbb{R}^D \to \mathbb{R}^D$, for some integer $D \ge 1$ we recall that the Jacobian $J$ at some $x$ is the matrix with $J_{ij} = \frac{\partial f_i}{\partial x_j}(x)$. For a function that takes a symmetric matrix $\Sigma$ and outputs the symmetric matrix $A\Sigma A^\top$, we view $D = \frac{d(d+1)}{2}$ and the function $\Sigma \mapsto A\Sigma A^\top$ as a function $f$ from $\mathbb{R}^D \to \mathbb{R}^D$ by taking the upper triangular part of both $\Sigma$ and $A\Sigma A^\top$. Note that $f$ is a linear function (thus $f(x) = J \cdot x$ where $J \in \mathbb{R}^{D \times D}$, and moreover, the following fact is well-known.

**Lemma C.1.** *[MH08, Theorem 11.1.5] For $J$ as defined above,* $\det(J) = \det(A)^{d+1}$.

Now, for any fixed $\mu \in \mathbb{R}^d$ and positive definite $\Sigma \in \mathbb{R}^{d \times d}$, consider the transformation $(\hat{\mu}, \hat{\Sigma}) \mapsto (\Sigma^{1/2}\hat{\Sigma}\Sigma^{1/2}, \Sigma^{1/2}\hat{\mu} + \mu)$, viewed as a linear map $g$ from $\mathrm{Proj}(\hat{\mu}, \hat{\Sigma}) \to \mathrm{Proj}(\Sigma^{1/2}\hat{\mu} + \mu, \Sigma^{1/2}\hat{\Sigma}\Sigma^{1/2})$. By setting $A := \Sigma^{1/2}$, note that the map $g$ behaves like $f$ on the last $\frac{d(d+1)}{2}$ coordinates, and on the first $d$ coordinates, it is simply an affine map $\hat{\mu} \mapsto \Sigma^{1/2}\hat{\mu} + \mu$. Therefore, the overall linear map $g$ has determinant $\det(\Sigma^{1/2}) \cdot \det(\Sigma^{1/2})^{d+1} = (\det \Sigma)^{(d+2)/2}$.

From this, we can infer the following.

**Lemma C.2.** *Fix any $\mu \in \mathbb{R}^d$ and positive definite $\Sigma \in \mathbb{R}^{d \times d}$. Let $h$ be the map $(\hat{\mu}, \hat{\Sigma}) \mapsto (\Sigma^{1/2}\hat{\mu} + \mu, \Sigma^{1/2}\hat{\Sigma}\Sigma^{1/2})$. Then, for any set $S$ of mean-covariance pairs, the normalized volume of $S$ equals the normalized volume of $h(S)$.*

*Proof.* Let $g$ be the corresponding map from $\text{Proj}(\hat{\mu}, \hat{\Sigma})$ to $\text{Proj}(\Sigma^{1/2}\hat{\mu} + \mu, \Sigma^{1/2}\hat{\Sigma}\Sigma^{1/2})$. Using a simple integration by substitution, we have

$$
\begin{aligned}
\text{vol}_\text{n}(h(S)) &= \int_{\hat{\theta}=\text{Proj}(\hat{\mu},\hat{\Sigma})\in\text{Proj}(h(S))} \frac{1}{(\det\hat{\Sigma})^{(d+2)/2}} d\hat{\theta} \\
&= \int_{\hat{\theta}\in g(\text{Proj}(S))} \frac{1}{(\det\hat{\Sigma})^{(d+2)/2}} d\hat{\theta} \\
&= \int_{\hat{\theta}=\text{Proj}(\hat{\mu},\hat{\Sigma})\in\text{Proj}(S)} \frac{1}{(\det(\Sigma^{1/2}\hat{\Sigma}\Sigma^{1/2}))^{(d+2)/2}} \cdot (\det g) d\hat{\theta} \\
&= \int_{\hat{\theta}\in\text{Proj}(S)} \frac{1}{(\det\hat{\Sigma})^{(d+2)/2}} d\hat{\theta} = \text{vol}_\text{n}(S).
\end{aligned}
$$

$\square$

### C.2 Robustness to Privacy Conversion

We note the following restatement of (the inefficient, approx-DP version of) the main robustness-to-privacy conversion of [HKMN23].

In the following theorem, we will think of the parameter space as lying in $\mathbb{R}^D$, with some normalized volume $\text{vol}_\text{n}(\Omega) = \int_{\theta\in\Omega} p(\theta)d\theta$, where $p \geq 0$ is some nonnegative Lebesgue-measurable function and the integration is Lebesgue integration. In our application, we will think of $\theta = \text{Proj}(\mu, \Sigma)$, and $p(\theta) = (\det\Sigma)^{-(d+2)/2}$, to match with (1).

**Theorem C.3.** *Restatement of [HKMN23, Lemma 4.2] Let $0 < \eta < 0.1$ and $10\eta \leq \eta^* \leq 1$ be fixed parameters. Also, fix privacy parameters $\varepsilon_0, \delta_0 < 1$ and confidence parameter $\beta_0 < 1$. Let $\mathcal{S} = \mathcal{S}(\theta, \mathbf{X}) \in \mathbb{R}_{\geq 0}$ be a score function that takes as input a dataset $\mathbf{X} = \{X_1, \ldots, X_n\}$ and a parameter $\theta \in \Theta \subset \mathbb{R}^D$. For any dataset $\mathbf{X}$ and any $0 \leq \eta' \leq 1$, let $V_{\eta'}(\mathbf{X})$ be the $D$-dimensional normalized volume of points $\theta \in \Theta$ with score at most $\eta' \cdot n$, i.e., $V_{\eta'}(\mathbf{X}) = \text{vol}_\text{n}(\{\theta \in \Theta : \mathcal{S}(\theta, \mathbf{X}) \leq \eta' \cdot n\})$.*

*Suppose the following properties hold:*

- *(Bounded Sensitivity) For any two adjacent datasets $\mathbf{X}, \mathbf{X}'$ and any $\theta \in \Theta$, $|\mathcal{S}(\theta, \mathbf{X}) - \mathcal{S}(\theta, \mathbf{X}')| \leq 1$.*

- *(Volume) For some universal constant $C$, and for any $\mathbf{X}$ of size $n$ such that there exists $\theta$ with $\mathcal{S}(\theta, \mathbf{X}) \leq 0.7\eta^* n$, $n \geq C \cdot \frac{\log(V_{\eta^*}(\mathbf{X})/V_{0.8\eta^*}(\mathbf{X})) + \log(1/\delta_0)}{\varepsilon_0 \cdot \eta^*}$.*

*Then, there exists an $(\varepsilon_0, \delta_0)$-DP algorithm $\mathcal{M}$, that takes as input $\mathbf{X}$ and outputs either some $\theta \in \Theta$ or $\bot$, such that for any dataset $\mathbf{X}$, if $n \geq C \cdot \max_{\eta':\eta\leq\eta'\leq\eta^*} \frac{\log(V_{\eta'}(\mathbf{X})/V_\eta(\mathbf{X})) + \log(1/(\beta_0\cdot\eta))}{\varepsilon_0 \cdot \eta'}$, then $\mathcal{M}(\mathbf{X})$ outputs some $\theta \in \Theta$ of score at most $2\eta n$ with probability $1 - \beta_0$.*

*We remark that the algorithm $\mathcal{M}$ is allowed prior knowledge of $n, D, \eta, \eta^*, \varepsilon_0, \delta_0, \beta_0$, as well as the domain $\Theta$, function $p(\theta)$ that dictates the normalized volume, and score function $\mathcal{S}$.*

We remark that the original result in [HKMN23] assumes the volumes are unnormalized, but the result immediately generalizes to normalized volumes. To see why, consider a modified domain $\Theta' \in \mathbb{R}^{D+1}$, where $\theta' = (\theta, z) \in \Theta'$ if and only if $\theta \in \Theta$ and $0 \leq z \leq p(\theta)$. Also, consider a modified score function $\mathcal{S}'$ acting on $\Theta'$, where $\mathcal{S}'((\theta, z), \mathbf{X}) := \mathcal{S}(\theta, \mathbf{X})$ for for any $(\theta, z) \in \Theta'$.

Then, note that the unnormalized volume of $\Theta'$, by Fubini's theorem, is precisely $\iint_{(\theta,z)\in\Theta'} 1 dz d\theta = \int_{\theta\in\Theta} p(\theta)d\theta$, which is precisely the normalized volume of corresponding to the new $\Theta'$ precisely match the normalized volumes corresponding to the old $\Theta$. A similar calculation will give us that the unnormalized volume of points $(\theta, z)$ in $\Theta'$ with $\mathcal{S}'((\theta, z), \mathbf{X}) \leq t$ equals the normalized volume of points $\theta \in \Theta$ with $\mathcal{S}(\theta, \mathbf{X}) \leq t$, for any $t$.

Thus, to privately find a parameter $\theta \in \Theta$ of low score with respect to $\mathcal{S}$ (assuming the normalized volume constraints hold), can create $\Theta'$ and $\mathcal{S}'$, and apply the unnormalized version of Theorem C.3 to obtain some $(\theta, z)$. Also, if $\mathcal{S}'((\theta, z), \mathbf{X}) \leq 2\eta n$, then $\mathcal{S}(\theta) = \mathcal{S}'((\theta, z), \mathbf{X}) \leq 2\eta n$, and if outputting $(\theta, z)$ is $(\varepsilon, \delta)$-DP, then so is simply outputting $\theta$.

### C.3 Computing Normalized Volume Ratios

**Lemma C.4.** *Let $M \in \mathbb{R}^{d \times d}$ be symmetric with $\|M\|_{op} \leq 0.1$, and let $1 \leq \nu \leq 2$ be a scaling parameter. Then,*

$$\nu^{-d} \leq \frac{\det(I + \nu M)}{\det(I + M)} \leq \nu^d.$$

*Proof.* If the eigenvalues of $M$ are $\lambda_1, \ldots, \lambda_d \in [-0.1, 0.1]$, then

$$\frac{\det(I + \nu M)}{\det(I + M)} = \prod_{i=1}^{d} \frac{1 + \nu \cdot \lambda_i}{1 + \lambda_i}.$$

Note that for $x \in [-0.1, 0.1]$ and $\nu \geq 1$, $\frac{1 + \nu \cdot x}{1 + x}$ is an increasing function. Therefore,

$$\prod_{i=1}^{d} \frac{1 + \nu \cdot \lambda_i}{1 + \lambda_i} \leq \prod_{i=1}^{d} \frac{1 + 0.1\nu}{1 + 0.1} \leq \prod_{i=1}^{d} \nu = \nu^d$$

and

$$\prod_{i=1}^{d} \frac{1 + \nu \cdot \lambda_i}{1 + \lambda_i} \geq \prod_{i=1}^{d} \frac{1 - 0.1\nu}{1 - 0.1} \geq \prod_{i=1}^{d} \frac{1}{\nu} = \nu^{-d},$$

where we used the fact that $1 \leq \nu \leq 2$. $\qquad \square$

We note an important lemmas about the normalized volumes of certain sets.

**Lemma C.5.** *Fix some $\gamma, \tau \leq 0.1$ and some scaling parameters $1 \leq \nu_1 \leq 2$ and $1 \leq \nu_2$. Let $\mathcal{R}_1$ be the set of $(\mu, \Sigma) \approx_{\gamma, \tau} (\mathbf{0}, I)$ and $\mathcal{R}_2$ be the set of $(\mu, \Sigma) \approx_{\nu_1 \cdot \gamma, \nu_2 \cdot \tau} (\mathbf{0}, I)$. Then, the ratio of normalized volumes*

$$\frac{\mathrm{vol_n}(\mathcal{R}_2)}{\mathrm{vol_n}(\mathcal{R}_1)} \leq \nu_1^{2d^2} \cdot \nu_2^d$$

*Proof.* By definition of normalized volume, we have

$$\mathrm{vol_n}(\mathcal{R}_2) = \iint_{\|M\|_{op} \leq \nu_1 \cdot \gamma, \|\mu\|_2 \leq \nu_2 \cdot \tau} \frac{1}{\det(I + M)^{(d+2)/2}} d\mu dM,$$

where we are slightly abusing notation because we are truly integrating along the upper-triangular part of $M$. (Overall, the integral is $\frac{d(d+3)}{2}$-dimensional.) We have a similar expression for $\mathrm{vol_n}(\mathcal{R}_1)$.

Let us consider the map sending $(\mu, I + M) \mapsto (\nu_2 \cdot \mu, I + \nu_1 \cdot M)$. This is a linear map, and for $\|\mu\|_2 \leq \gamma$ and symmetric $\|M\|_{op} \leq \tau$, this is a bijective map from $\mathcal{R}_1$ to $\mathcal{R}_2$. Therefore, by an integration by substitution, we have

$$\begin{aligned}
\mathrm{vol_n}(\mathcal{R}_2) &= \iint_{\|M\|_{op} \leq \gamma, \|\mu\|_2 \leq \tau} \frac{1}{\det(I + \nu_1 \cdot M)^{(d+2)/2}} \cdot \nu_1^{d(d+1)/2} \nu_2^d d\mu dM \\
&\leq \nu_1^{d(d+1)/2} \nu_2^d \cdot (\nu^{-d})^{-(d+2)/2} \cdot \iint_{\|M\|_{op} \leq \gamma, \|\mu\|_2 \leq \tau} \frac{1}{\det(I + M)^{(d+2)/2}} d\mu dM \\
&= \nu_1^{2d^2} \nu_2^d \cdot \mathrm{vol_n}(\mathcal{R}_1).
\end{aligned}$$

Above, the first line is integration by substitution, and the second line uses Lemma C.4. $\qquad \square$

Next, we note the following bound on the size of a net of matrices with small Frobenius norm.

**Lemma C.6.** *Fix some $\gamma \leq 0.1$ and $\frac{1}{d} \leq \rho \leq 0.1\sqrt{d}$. Let $\mathcal{R}_3$ be the set of symmetric matrices $M \in \mathbb{R}^{d \times d}$ with $\|M\|_{op} \leq \gamma$ and $\|M\|_F \leq \rho$. Then, for any $1 \leq \ell \leq d$, there exists a net $\mathcal{B}$ of size at most $e^{O(\ell \cdot d \log d)}$ such that for any $M \in \mathcal{R}_3$, there exists $B \in \mathcal{B}$ such that $\|M - B\|_{op} \leq \frac{2\rho}{\sqrt{\ell}}$.*

*Proof.* First, note that the unit sphere in $\mathbb{R}^d$ has a $\frac{1}{d^{10}}$-net (in Euclidean distance) of size $e^{O(d \log d)}$. Let $\mathcal{B}_0$ be this net. The net $\mathcal{B}$ will be the set of matrices $\sum_{i=1}^{\ell} \kappa_i w_i w_i^\top$, where every $w_i \in \mathcal{B}_0$, every $\kappa_i$ is an integral multiple of $\frac{1}{d^{10}}$, and every $|\kappa_i| \leq \gamma$. The cardinality of $\mathcal{B}$ is at most $|\mathcal{B}_0|^\ell \cdot (d^{10})^\ell \leq e^{O(\ell \cdot d \log d)}$.

Now, we show that $\mathcal{B}$ is actually a net. For any matrix $M \in \mathcal{R}_3$, we can write $M = \sum_{i=1}^{d} \lambda_i v_i v_i^\top$, where $\lambda_i, v_i$ are the eigenvalues and eigenvectors, respectively, of $M$. Assume the eigenvalues are sorted so that $|\lambda_i|$ are in decreasing order. Now, since $\sum_{i=1}^{d} \lambda_i^2 \leq \rho^2$, which means that $|\lambda_i| \leq \frac{\rho}{\sqrt{\ell}}$ for all $i > \ell$.

Now, let $B_1 = \sum_{i>\ell} \lambda_i v_i v_i^\top$: since the $v_i$'s are orthogonal and $|\lambda_i| \leq \frac{\rho}{\sqrt{\ell}}$, this means that $\|B_1\|_{op} \leq \frac{\alpha}{\sqrt{\ell}}$. Also, note that $M = B_1 + \sum_{i=1}^{\ell} \lambda_i v_i v_i^\top$. Suppose we replace each $\lambda_i$ with $\kappa_i$ by rounding to the nearest multiple of $1/d^{10}$, and replace each $v_i$ with $w_i \in \mathcal{B}_0$ such that $\|v_i - w_i\|_2 \leq 1/d^{10}$. Then, $\sum_{i=1}^{\ell} \kappa_i w_i w_i^\top$ is in $\mathcal{B}$, and by Triangle inequality, we have

$$\left\| \sum_{i=1}^{\ell} \lambda_i v_i v_i^\top - \sum_{i=1}^{\ell} \kappa_i w_i w_i^\top \right\|_{op} \leq \ell \cdot (|\lambda_i - \kappa_i| + 2 \cdot \lambda_i \cdot \|v_i - w_i\|_2) \leq O\left(\frac{1}{d^9}\right) \leq \frac{\rho}{\sqrt{\ell}}.$$

Thus, $\|M - \sum_{i=1}^{\ell} \kappa_i w_i w_i^\top\|_{op} \leq 2 \cdot \frac{\rho}{\sqrt{\ell}}$. $\qquad\square$

We also note the following lemma. We defer the proof to Appendix H.

**Lemma C.7.** *Fix any $\mu$ and positive definite $\Sigma$. Then, for any $\mu_1, \mu_2, \Sigma_1, \Sigma_2$, we have that $(\mu_1, \Sigma_1) \approx_{\gamma, \rho, \tau} (\mu_2, \Sigma_2)$ if and only if $(\Sigma^{1/2}\mu_1 + \mu, \Sigma^{1/2}\Sigma_1\Sigma^{1/2}) \approx_{\gamma, \rho, \tau} (\Sigma^{1/2}\mu_2 + \mu, \Sigma^{1/2}\Sigma_2\Sigma^{1/2})$.*

Our main lemma in this subsection is the following, which roughly bounds the normalized volume of a Frobenius norm ball "fattened" by an operator norm ball.

**Lemma C.8.** *Let $c_1 \in (0, 0.01)$ be a sufficiently small universal constant. Fix any $\mu$ and positive definite $\Sigma$. Fix some parameters $\gamma_1, \gamma_2 \in (\frac{1}{d}, c_1)$ and $\rho_2 \in (\frac{1}{d}, \frac{\gamma_1^3}{1000} \cdot \sqrt{d})$. Let $\mathcal{T}_1(\mu, \Sigma)$ represent the set of $\{(\mu_1, \Sigma_1)\}$ such that $(\mu_1, \Sigma_1) \approx_{\gamma_1, \gamma_1} (\mu, \Sigma)$. Let $\mathcal{T}_2(\mu, \Sigma)$ represent the set of $\{(\mu_2, \Sigma_2)\}$ such that $(\mu_2, \Sigma_2) \approx_{\gamma_1, \gamma_1} (\mu_1, \Sigma_1)$ for some $(\mu_1, \Sigma_1) \approx_{\gamma_2, \rho_2, \gamma_2} (\mu, \Sigma)$. Then,*

$$\frac{\text{vol}_n(\mathcal{T}_2(\mu, \Sigma))}{\text{vol}_n(\mathcal{T}_1(\mu, \Sigma))} \leq e^{O(d^{5/3} \log d \cdot \rho_2^{2/3}/\gamma_1^2)}.$$

*Furthermore, both $\text{vol}_n(\mathcal{T}_1(\mu, \Sigma))$ and $\text{vol}_n(\mathcal{T}_2(\mu, \Sigma))$ do not change even if we change $\mu, \Sigma$.*

*Proof.* For now, we also assume that $\mu = \mathbf{0}$ and $\Sigma = I$.

Let $1 \leq \ell \leq d$ be decided later, and let $\mathcal{B}$ be the net from Lemma C.6, where we set $\gamma = \gamma_2$ and $\rho = \rho_2$. Let $\mathcal{B}_1$ be a $\frac{1}{d^{10}}$-net (in Euclidean distance) over the $d$-dimensional unit ball, i.e., over $\mu$ with $\|\mu\|_2 \leq 1$. Note that $|\mathcal{B}_1| \leq e^{O(d \log d)}$, and recall that $|\mathcal{B}| \leq e^{O(\ell \cdot d \log d)}$. Now, for any $(\mu_2, \Sigma_2) \in \mathcal{T}_2(\mathbf{0}, I)$, we can write $(\mu_2, \Sigma_2) \approx_{\gamma_1, \gamma_1} (\mu_1, \Sigma_1)$, where $\|\Sigma_1 - I\|_{op} \leq \gamma_2$, $\|\Sigma_1 - I\|_F \leq \rho_2$, and $\|\mu_1\|_2 \leq \gamma_2$. Now, we can choose some $\mu' \in \mathcal{B}_1$ with $\|\mu_1 - \mu'\|_2 \leq d^{-10}$ and $B \in \mathcal{B}$ such that $\|(\Sigma_1 - I) - B\|_{op} \leq \frac{2\rho_2}{\sqrt{\ell}}$.

What can we say about the relationship between $(\mu_2, \Sigma_2)$ and $(\mu', I + B)$? First, note that because $(\mu_2, \Sigma_2) \approx_{\gamma_1, \gamma_1} (\mu_1, \Sigma_1)$, we have $(1 - \gamma_1) \cdot \Sigma_1 \preccurlyeq \Sigma_2 \preccurlyeq (1 + \gamma_1) \cdot \Sigma_1$. Next, since $\|\Sigma_1 - (I + B)\|_{op} = \|(\Sigma_1 - I) - B\|_{op} \leq \frac{2\rho_2}{\sqrt{\ell}}$, and since $\Sigma_1$ has all eigenvalues between $1 - \gamma_2 \geq \frac{1}{2}$ and $1 + \gamma_2 \leq 2$, this means $\left(1 - \frac{4\rho_2}{\sqrt{\ell}}\right) \cdot \Sigma_1 \preccurlyeq (I + B) \preccurlyeq \left(1 + \frac{4\rho_2}{\sqrt{\ell}}\right) \cdot \Sigma_1$. Thus, if $4\rho < \sqrt{\ell}$, we have that

$$\frac{1 - \gamma_1}{1 + (4\rho_2/\sqrt{\ell})} \cdot (I + B) \preccurlyeq \Sigma_1 \preccurlyeq \frac{1 + \gamma_1}{1 - (4\rho_2/\sqrt{\ell})} \cdot (I + B).$$

Next, because $(\mu_2, \Sigma_2) \approx_{\gamma_1, \gamma_1} (\mu_1, \Sigma_1)$, we have $\|\Sigma_1^{-1/2}(\mu_2 - \mu_1)\|_2 \leq \gamma_1$, which means $\|\mu_2 - \mu_1\|_2 \leq 2\gamma_1$. Thus, because $\|\mu_1 - \mu'\|_2 \leq d^{-10} \leq \gamma_1$, this means that $\|\mu_2 - \mu'\|_2 \leq 3\gamma_1$. Moreover,

assuming $10\rho \leq \sqrt{\ell}$, the eigenvalues of $I + B$ are at least $(1 - \gamma_1) \cdot \left(1 - \frac{4\rho_2}{\sqrt{\ell}}\right) \geq \frac{1}{4}$, which means that $\|(I + B)^{-1/2}(\mu_2 - \mu')\|_2 \leq 6\gamma_1$.

In summary, if $10\rho \leq \sqrt{\ell}$, then

$$\frac{1 - \gamma_1}{1 + (4\rho_2/\sqrt{\ell})} \cdot (I + B) \preccurlyeq \Sigma_2 \preccurlyeq \frac{1 + \gamma_1}{1 - (4\rho_2/\sqrt{\ell})} \cdot (I + B), \quad \|(I + B)^{-1/2}(\mu_2 - \mu')\|_2 \leq 6\gamma_1.$$

Note that if $\gamma_1 \leq 0.1$ and $10\rho \leq \sqrt{\ell}$, then by a simple calculation, $\frac{1+\gamma_1}{1-(4\rho_2/\sqrt{\ell})} \leq 1 + \gamma_1 + \frac{10\rho_2}{\sqrt{\ell}}$ and $\frac{1-\gamma_1}{1+(4\rho_2/\sqrt{\ell})} \geq 1 - \gamma_1 - \frac{4\rho_2}{\sqrt{\ell}}$. This implies that $(\mu_2, \Sigma_2) \approx_{\gamma_1 + 10\rho_2/\sqrt{\ell}, 6\gamma_1} (\mu', I + B)$, for some $\mu' \in \mathcal{B}_1$ and $B \in \mathcal{B}$. Note that this holds for any $(\mu_2, \Sigma_2) \in \mathcal{T}_2(\mathbf{0}, I)$.

Therefore, recalling that $|\mathcal{B}_1| \leq e^{O(d \log d)}$, and $|\mathcal{B}| \leq e^{O(\ell \cdot d \log d)}$ we can cover $\mathcal{T}_2(\mathbf{0}, I)$ with at most $e^{O(\ell \cdot d \log d)}$ regions, each of which is the set of $(\mu_2, \Sigma_2) \approx_{\gamma_1 + 10\rho_2/\sqrt{\ell}, 6\gamma_1} (\mu', \Sigma')$. Now, by Lemma C.7, $(\mu_2, \Sigma_2) \approx_{\gamma_1 + 10\rho_2/\sqrt{\ell}, 6\gamma_1} (\mu', \Sigma')$ if and only if $\mu_2 = \Sigma'^{1/2}\mu_0 + \mu'$ and $\Sigma_2 = \Sigma'^{1/2}\Sigma_0\Sigma'^{1/2}$, where $(\mu_0, \Sigma_0) \approx_{\gamma_1 + 10\rho_2/\sqrt{\ell}, 6\gamma_1} (\mathbf{0}, I)$. By Lemma C.2, this means that the volume of $\{(\mu_2, \Sigma_2) : (\mu_2, \Sigma_2) \approx_{\gamma_1 + 10\rho_2/\sqrt{\ell}, 6\gamma_1} (\mu', \Sigma')\}$ equals the volume of $\{(\mu_0, \Sigma_0) : (\mu_0, \Sigma_0) \approx_{\gamma_1 + 10\rho_2/\sqrt{\ell}, 6\gamma_1} (\mathbf{0}, I)\}$.

Thus, if $10\rho \leq \sqrt{\ell}$, then $\mathcal{T}_2(\mathbf{0}, I)$ can be covered by $e^{O(\ell \cdot d \log d)}$ regions, each of which has the same normalized volume as the volume of $\{(\mu_0, \Sigma_0) : (\mu_0, \Sigma_0) \approx_{\gamma_1 + 10\rho_2/\sqrt{\ell}, 6\gamma_1} (\mathbf{0}, I)\}$. Moreover, if we further have $\frac{10\rho_2}{\sqrt{\ell}} \leq \gamma_1$, then by Lemma C.8, this is at most $\left(1 + \frac{10\rho_2}{\sqrt{\ell} \cdot \gamma_1}\right)^{2d^2} \cdot 6^d \cdot \mathrm{vol_n}(\mathcal{T}_1(\mathbf{0}, I))$.

We will set $\ell = \frac{100(d\rho_2)^{2/3}}{\gamma_1^2}$. Since $\rho_2 \geq \frac{1}{d}$, $\ell \geq 100$ as long as $\gamma_1 \leq c_1 \leq 1$. Also, $\ell \leq 100 \cdot \left(\frac{\gamma_1^3}{1000} \cdot d^{3/2}\right)^{2/3} \cdot \frac{1}{\gamma_1^2} = d$. Finally, $\frac{10\rho_2}{\sqrt{\ell}} = \frac{\gamma_1 \cdot \rho_2^{2/3}}{d^{1/3}} \leq \gamma_1$, since $\rho_2 \leq \sqrt{d}$. Overall, the ratio

$$\frac{\mathrm{vol_n}(\mathcal{T}_2(\mathbf{0}, I))}{\mathrm{vol_n}(\mathcal{T}_1(\mathbf{0}, I))} \leq e^{O(\ell \cdot d \log d)} \cdot \left(1 + \frac{10\rho_2}{\sqrt{\ell} \cdot \gamma_1}\right)^{2d^2} \cdot 6^d \leq e^{O(d^{5/3} \log d \cdot \rho_2^{2/3}/\gamma_1^2)}.$$

Finally, we show how to remove the assumption that $\mu = \mathbf{0}$ and $\Sigma = I$. First, note that for any general $\mu, \Sigma$, $(\mu_1, \Sigma_1) \in \mathcal{T}_1(\mathbf{0}, I)$ if and only if $(\Sigma^{1/2}\mu_1 + \mu, \Sigma^{1/2}\Sigma_1\Sigma^{1/2}) \in \mathcal{T}_1(\mu, \Sigma)$, by Lemma C.7. So, by Lemma C.2, $\mathrm{vol_n}(\mathcal{T}_1(\mu, \Sigma)) = \mathrm{vol_n}(\mathcal{T}_1(\mathbf{0}, I))$. Next, $(\mu_2, \Sigma_2) \approx_{\gamma_1, \gamma_1} (\mu_1, \Sigma_1)$ and $(\mu_1, \Sigma_1) \approx_{\gamma_2, \rho_2, \gamma_2} (\mathbf{0}, I)$ if and only if $(\Sigma^{1/2}\mu_2 + \mu, \Sigma^{1/2}\Sigma_2\Sigma^{1/2}) \approx_{\gamma_1, \gamma_1} (\Sigma^{1/2}\mu_1 + \mu, \Sigma^{1/2}\Sigma_1\Sigma^{1/2})$ and $(\Sigma^{1/2}\mu_1 + \mu, \Sigma^{1/2}\Sigma_1\Sigma^{1/2}) \approx_{\gamma_2, \rho_2, \gamma_2} (\mu, \Sigma)$, by two applications of Lemma C.7. So, by Lemma C.2, $\mathrm{vol_n}(\mathcal{T}_2(\mu, \Sigma)) = \mathrm{vol_n}(\mathcal{T}_2(\mathbf{0}, I))$. Thus, the volume ratios stay the same as well. $\qquad\square$

## D   Fine Approximation via Hypothesis Selection

In this section, we prove that if we know a very crude approximation to all of the Gaussian mixture components with sufficiently large weight, we can privately learn the full density of the Gaussian mixture model up to low total variation distance. This will be useful for proving both Theorem 1.4 and Theorem 1.5.

We start with the following auxiliary lemma.

**Lemma D.1.** *Let $G \geq 1$ and $\zeta \leq 1$ be some parameters. For some $d \geq 1$, let $\hat{\mu} \in \mathbb{R}^d$, and let $\hat{\Sigma} \in \mathbb{R}^{d \times d}$ be positive definite. Let $\mathcal{U}_{\hat{\mu}, \hat{\Sigma}}$ be the set of $(\mu, \Sigma)$ such that $\frac{1}{G} \cdot \hat{\Sigma} \preccurlyeq \Sigma \preccurlyeq G \cdot \hat{\Sigma}$ and $\|\hat{\Sigma}^{-1/2}(\mu - \hat{\mu})\|_2 \leq G$. Then, there exists a net $\mathcal{B}_{\hat{\mu}, \hat{\Sigma}}$ of size $O(G\sqrt{d}/\zeta)^{d(d+3)}$ such that for every $(\mu, \Sigma) \in \mathcal{U}_{\hat{\mu}, \hat{\Sigma}}$, there exists $(\mu', \Sigma') \in \mathcal{B}_{\hat{\mu}, \hat{\Sigma}}$ such that $\mathrm{d_{TV}}(\mathcal{N}(\mu, \Sigma), \mathcal{N}(\mu', \Sigma')) \leq \zeta$.*

*Proof.* First, assume that $\hat{\Sigma} = I$ and $\hat{\mu} = \mathbf{0}$. Now, let us consider the set $\mathcal{U} := \mathcal{U}_{\mathbf{0}, I} = \{(\mu, \Sigma) : \|\mu\|_2 \leq G, \frac{1}{G} \cdot I \preccurlyeq \Sigma \preccurlyeq G \cdot I\}$. Let $G' \geq G$ be a parameter that we will set later. Now, we can look

at the *un-normalized* volume of $\mathcal{U}$ viewed in $\mathbb{R}^{d(d+3)/2}$ (i.e., we are projecting to the upper-triangular part of $\Sigma$). First, we claim (using a standard volume argument) that there is a cover $\mathcal{B} \subset \mathcal{U}$ of size $(2G')^{d(d+3)}$, such that for every $(\mu, \Sigma) \in \mathcal{U}$, there is $(\mu', \Sigma') \in \mathcal{B}$ with $\|\mu - \mu'\|_2 \le \frac{1}{G'}$ and $\|\Sigma - \Sigma'\|_{op} \le \frac{1}{G'}$.

To see why, consider a maximal packing of $(\mu_i', \Sigma_i') \in \mathcal{U}$ such that for every $i \ne j$, either $\|\mu_i' - \mu_j'\|_2 > \frac{1}{G'}$ or $\|\Sigma_i' - \Sigma_j'\|_{op} > \frac{1}{G'}$. Then, this set is clearly a cover of $\mathcal{U}$ (or else we could have increased the size of the packing, breaking maximality). Then, the sets $S_i := \{(\mu, \Sigma) : \|\mu - \mu_i'\| \le \frac{1}{2G'}, \|\Sigma - \Sigma_i'\|_{op} \le \frac{1}{2G'}\}$ are disjoint, and are all contained in the set $S := \{(\mu, \Sigma) : \|\mu\|_2 \le 2G', \|\Sigma\|_{op} \le 2G'\}$. $S$ is just a shifting and scaling of $S_i$ by a factor of $4(G')^2$, and thus $\mathrm{vol}(S) = (4(G')^2)^{d(d+3)/2} \cdot \mathrm{vol}(S_i) = (2G')^{d(d+3)} \cdot \mathrm{vol}(S_i)$. Because every $S_i$ is disjoint and contained in $S$, the number of such indices $i$ is at most $(2G')^{d(d+3)}$.

Next, consider any $(\mu, \Sigma) \in \mathcal{U}$ and $(\mu', \Sigma') \in \mathcal{B}$ with $\|\mu - \mu'\|_2 \le \frac{1}{G'}, \|\Sigma - \Sigma'\|_{op} \le \frac{1}{G'}$. Then, $\|\Sigma^{-1/2}\Sigma'\Sigma^{-1/2} - I\|_{op} = \|\Sigma^{-1/2}(\Sigma' - \Sigma)\Sigma^{-1/2}\|_{op} \le \|\Sigma^{-1}\|_{op} \cdot \|\Sigma' - \Sigma\|_{op} \le \frac{G}{G'}$. Also, $\|\Sigma^{-1/2}(\mu - \mu')\|_2 \le \|\Sigma^{-1/2}\|_{op} \cdot \|\mu - \mu'\|_2 \le \frac{\sqrt{G}}{G'}$. In other words, we have found a net $\mathcal{B}$ of size $(2G')^{d(d+3)}$ such that for every $(\mu, \Sigma) \in \mathcal{U}$, there exists $(\mu', \Sigma') \in \mathcal{B}$ with $(\mu', \Sigma') \approx_{G/G', G/G'} (\mu, \Sigma)$.

Next, consider general $\hat{\mu}, \hat{\Sigma}$. Note that $(\mu, \Sigma) \in \mathcal{U}_{\hat{\mu}, \hat{\Sigma}}$ is equivalent to $\mu = \hat{\Sigma}^{1/2}\mu_0 + \hat{\mu}$, where $\|\mu_0\|_2 \le G$, and $\Sigma = \hat{\Sigma}^{1/2}\Sigma_0\hat{\Sigma}^{1/2}$, where $\frac{1}{G} \cdot I \preccurlyeq \Sigma_0 \preccurlyeq G \cdot I$. So, if we consider the map $f : (\mu_0, \Sigma_0) \mapsto (\hat{\Sigma}^{1/2}\mu_0 + \hat{\mu}, \hat{\Sigma}^{1/2}\Sigma_0\hat{\Sigma}^{1/2})$, this bijectively maps $\mathcal{U}$ to $\mathcal{U}_{\hat{\mu}, \hat{\Sigma}}$. We can also consider the cover $\mathcal{B}_{\hat{\mu}, \hat{\Sigma}} = f(\mathcal{B})$, where $\mathcal{B}$ is the cover constructed for $\mathcal{U}$. Then, by Lemma C.7, $(\mu_0', \Sigma_0') \approx_{G/G', G/G'} (\mu_0, \Sigma_0)$ if and only if $f(\mu_0', \Sigma_0') \approx_{G/G', G/G'} f(\mu_0, \Sigma_0)$.

Hence, regardless of the choice of $\hat{\mu}, \hat{\Sigma}$, for every $(\mu, \Sigma) \in \mathcal{U}_{\hat{\mu}, \hat{\Sigma}}$, there exists $(\mu', \Sigma') \in \mathcal{B}_{\hat{\mu}, \hat{\Sigma}}$ with $(\mu', \Sigma') \approx_{G/G', G/G'} (\mu, \Sigma)$. Moreover, $|\mathcal{B}_{\hat{\mu}, \hat{\Sigma}}| = |\mathcal{B}| \le (2G')^{d(d+3)}$. Moreover, if $(\mu', \Sigma') \approx_{G/G', G/G'} (\mu, \Sigma)$, then $\|\Sigma^{-1/2}(\mu' - \mu)\|_2 \le \frac{G}{G'}$ and $\|\Sigma^{-1/2}\Sigma\Sigma^{-1/2} - I\|_F \le \sqrt{d} \cdot \|\Sigma^{-1/2}\Sigma\Sigma^{-1/2} - I\|_{op} \le \frac{\sqrt{d} \cdot G}{G'}$. So, by Lemma A.8, $d_{\mathrm{TV}}(\mathcal{N}(\mu, \Sigma), \mathcal{N}(\mu', \Sigma')) \le \frac{\sqrt{d} \cdot G}{G'}$. Hence, if we set $G' = \frac{G\sqrt{d}}{\zeta}$, then the size of the net $\mathcal{B}_{\hat{\mu}, \hat{\Sigma}}$ is $O(G\sqrt{d}/\zeta)^{d(d+3)}$ and for every $(\mu, \Sigma) \in \mathcal{U}_{\hat{\mu}, \hat{\Sigma}}$, there exists $(\mu', \Sigma') \in \mathcal{B}_{\hat{\mu}, \hat{\Sigma}}$ such that $d_{\mathrm{TV}}(\mathcal{N}(\mu, \Sigma), \mathcal{N}(\mu', \Sigma')) \le \zeta$. $\square$

Next, we note the following result about differentially private hypothesis selection.

**Theorem D.2.** *[BSKW19] Let $\mathcal{H} = \{H_1, \ldots, H_M\}$ be a set of probability distributions over some domain $D$. There exists an $\varepsilon$-differentially private algorithm (with respect to a dataset $\mathbf{X} = \{X_1, \ldots, X_n\}$) which has following guarantees.*

*Let $\mathcal{D}$ be an unknown probability distribution over $D$, and suppose there exists a distribution $H^* \in \mathcal{H}$ such that $d_{\mathrm{TV}}(\mathcal{D}, H^*) \le \alpha$. If $n \ge O\left(\frac{\log M}{\alpha^2} + \frac{\log M}{\alpha\varepsilon}\right)$, and if $X_1, \ldots, X_n$ are samples drawn independently from $\mathcal{D}$, then the algorithm will output a distribution $\hat{H} \in \mathcal{H}$ such that $d_{\mathrm{TV}}(\mathcal{D}, \hat{H}) \le 4\alpha$ with probability at least $9/10$.*

Given Lemma D.1 and Theorem D.2, we are in a position to convert any crude approximation into a fine approximation. Namely, we prove the following result.

**Lemma D.3.** *Let $\mathcal{D}$ represent an unknown GMM with representation $\{(w_i, \mu_i, \Sigma_i)\}_{i=1}^k$. Let $G \ge 1$ be some fixed parameter. Then there exists an $\varepsilon$-differentially algorithm, on $n \ge O\left(\frac{d^2 k \cdot \log(G \cdot k \cdot k' \cdot \sqrt{d}/\alpha)}{\alpha^2} + \frac{d^2 k \cdot \log(G \cdot k \cdot k' \cdot \sqrt{d}/\alpha)}{\alpha\varepsilon}\right)$ samples, with the following property.*

*Suppose the algorithm is given as input a set $\{(\hat{\mu}_j, \hat{\Sigma}_j)\}_{j=1}^{k'}$ for some $k' \ge 1$, such that for every $(\mu_i, \Sigma_i)$ with weight $w_i \ge \frac{\alpha}{k}$, there exists $j \le k'$ such that $\frac{1}{G} \cdot \hat{\Sigma}_j \preccurlyeq \Sigma_i \preccurlyeq G \cdot \hat{\Sigma}_j$ and $\|\hat{\Sigma}_j^{-1/2}(\mu_i - \hat{\mu}_j)\|_2 \le G$. Then, if the samples are $X_1, \ldots, X_n \overset{i.i.d.}{\sim} \mathcal{D}$, then with probability at least $9/10$, the algorithm outputs a mixture of at most $k$ Gaussians $H$, with $d_{\mathrm{TV}}(\mathcal{D}, H) \le O(\alpha)$.*

*Proof.* Let $\zeta = \frac{\alpha}{k}$. For each $(\hat{\mu}_j, \hat{\Sigma}_j)$, let $\mathcal{B}_{\hat{\mu}_j, \hat{\Sigma}_j}$ be as in Lemma D.1. We define $\hat{\mathcal{B}} = \bigcup_{j \leq k'} \mathcal{B}_{\hat{\mu}_j, \hat{\Sigma}_j}$.

Let $\mathcal{H}$ be the set of hypotheses consisting of mixtures of up to $k$ Gaussians $\mathcal{N}(\mu_i', \Sigma_i')$ with weights $w_i'$, with every $(\mu_i', \Sigma_i') \in \hat{\mathcal{B}}$ and with every $w_i'$ an integral multiple of $\zeta$. Note that the number of hypothesis $M = |\mathcal{H}|$ is at most $|\hat{\mathcal{B}}|^{O(k)} \cdot (1/\zeta)^{O(k)} = (k' \cdot G \cdot \sqrt{d}/\zeta)^{O(d^2 \cdot k)}$. Since we set $\zeta = \alpha/k$, $M \leq (G \cdot \frac{kk'\sqrt{d}}{\alpha})^{O(d^2 \cdot k)}$.

Next, let us consider the true GMM $\mathcal{D}$, with representation $\{(w_i, \mu_i, \Sigma_i)\}_{i=1}^k$. Because every $(\mu_i, \Sigma_i)$ with $w_i \geq \frac{\alpha}{k}$ satisfies $\frac{1}{G} \cdot \hat{\Sigma}_j \preccurlyeq \Sigma_i \preccurlyeq G \cdot \hat{\Sigma}_j$ and $\|\hat{\Sigma}_j^{-1/2}(\mu_i - \hat{\mu}_j)\|_2 \leq G$, by Lemma D.1, there exists $(\mu_i', \Sigma_i') \in \hat{\mathcal{B}}$ such that $d_{\mathrm{TV}}(\mathcal{N}(\mu_i, \Sigma_i), (\mu_i', \Sigma_i')) \leq \zeta$. Moreover, we can round each $w_i \geq \frac{\alpha}{k}$ to some $w_i'$ which is an integral multiple of $\zeta$, such that $|w_i - w_i'| \leq \zeta$. Finally, for each $w_i < \frac{\alpha}{k}$, we can choose an arbitrary Gaussian in $\hat{\mathcal{B}}$ and round $w_i$ to some $w_i'$. Then, the total variation distance between $\mathcal{D}$ and the GMM with representation $\{(w_i', \mu_i', \Sigma_i')\}_{i=1}^k$ is at most

$$\sum_{i \leq k: w_i > \alpha/k} \left(\frac{\alpha}{k} + |w_i - w_i'|\right) + \sum_{i \leq k: w_i \geq \alpha/k} (\zeta + |w_i - w_i'|) \leq \alpha + O(k \cdot \zeta).$$

Hence, if we set $\zeta = \frac{\alpha}{k}$, there exists a distribution $H^* \in \mathcal{H}$ with $d_{\mathrm{TV}}(\mathcal{D}, H^*) \leq O(\alpha)$.

Therefore, the algorithm of Theorem D.2, using $n \geq O\left(\frac{d^2 k \cdot \log(G \cdot k \cdot k' \cdot \sqrt{d}/\alpha)}{\alpha^2} + \frac{d^2 k \cdot \log(G \cdot k \cdot k' \cdot \sqrt{d}/\alpha)}{\alpha \varepsilon}\right)$ samples, will find $H \in \mathcal{H}$ such that $d_{\mathrm{TV}}(\mathcal{D}, H) \leq O(\alpha)$, and is $\varepsilon$-differentially private. $\square$

**Pseudocode:** We give a simple pseudocode for the algorithm of Lemma D.3, in Algorithm 1.

---

**Algorithm 1:** FINE-ESTIMATE$(X_1, X_2, \ldots, X_n \in \mathbb{R}^d, d, k, k', \varepsilon, \alpha, G, \{(\hat{\mu}_j, \hat{\Sigma}_j)\}_{j \leq k'})$

---

**Input:** Samples $X_1, \ldots, X_n$, and crude predictions $(\hat{\mu}_j, \hat{\Sigma}_j)$, where $1 \leq j \leq k'$ for some $k'$.
**Output:** An $(\varepsilon, 0)$-DP prediction $H$, which is a mixture over at most $k$ Gaussians.
1 Set $\zeta = \alpha/k$.
2 **for** $j = 1$ *to* $k'$ **do**
3 $\quad$ Define the sets $\mathcal{B}_{\hat{\mu}_j, \hat{\Sigma}_j}$ as in Lemma D.1, for parameters $G, \zeta$.
4 **end**
5 $\hat{\mathcal{B}} \leftarrow \bigcup_{j \leq k'} \mathcal{B}_{\hat{\mu}_j, \hat{\Sigma}_j}$.
6 Let $\mathcal{H}$ be the set of mixtures $\{(w_i', \mu_i', \Sigma_i')\}$ of $k$ or fewer components, where every $(\mu_i', \Sigma_i') \in \hat{\mathcal{B}}$ and every $w_i'$ is an integral multiple of $\zeta$.
7 Run the $\varepsilon$-DP algorithm of Theorem D.2 on $X_1, \ldots, X_n$ with respect to $\mathcal{H}$.

---

# E  The High Dimensional Setting

In this section, we provide the algorithm and analysis for Theorem 1.4. We first describe and provide pseudocode for the algorithm, and then we prove that it can privately learn mixtures of $d$-dimensional Gaussians with low sample complexity.

## E.1  Algorithm

**High-Level Approach:** Suppose that the unknown distribution is a GMM with representation $\{(w_i, \mu_i, \Sigma_i)\}_{i=1}^k$.

Define $\Theta$ to be the set of all feasible mean-covariance pairs $(\mu, \Sigma)$, i.e., where $\mu \in \mathbb{R}^d$ and $\Sigma \in \mathbb{R}^{d \times d}$ is positive definite. We also start off with a set $\Omega = \Theta$, which will roughly characterize the region of "remaining" mean-covariance pairs.

At a high level, our algorithm proceeds as follows. We will first learn a very crude approximation of each $(\mu_i, \Sigma_i)$, one at a time. Namely, using roughly $\left(\frac{\varepsilon}{\sqrt{k \log(1/\delta)}}, \frac{\delta}{k}\right)$-DP, we will learn some

$(\hat{\mu}_1, \hat{\Sigma}_1) \in \Omega$ which is "vaguely close" to some $(\mu_i, \Sigma_i)$. We will then remove every $(\mu, \Sigma)$ that is vaguely close to this $(\hat{\mu}, \hat{\Sigma})$ from our $\Omega$, and then attempt to repeat this process. We will repeat it up to $k$ times, in hopes that every $(\mu_i, \Sigma_i)$ is vaguely close to some $(\hat{\mu}_j, \hat{\Sigma}_j)$. By advanced composition, the full set of $\{(\hat{\mu}_j, \hat{\Sigma}_j)\}$ is still $(\varepsilon, \delta)$-DP.

At this point, the remainder of the algorithm is quite simple. Namely, we have some crude estimate for every $(\mu_i, \Sigma_i)$ (it is "vaguely close" to some $(\hat{\mu}_j, \hat{\Sigma}_j)$). Moreover, one can create a fine net of roughly $e^{\tilde{O}(d^2)}$ $(\mu, \Sigma)$-pairs that cover all mean-covariance pairs vaguely close to each $(\hat{\mu}_j, \hat{\Sigma}_j)$, since the dimension of $(\mu, \Sigma)$ is $O(d^2)$. Moreover, the weight vector $(w_1, \ldots, w_k)$ has a fine net of size roughly $e^{\tilde{O}(k)}$. As a result, we can reduce the problem to a small set of hypotheses: there are roughly $e^{\tilde{O}(d^2)}$ choices for each $(\mu_i, \Sigma_i)$, and $e^{\tilde{O}(k)}$ choices for $w$, for a total of $e^{\tilde{O}(k \cdot d^2)}$ choices for the mixture of Gaussians. We can then apply known results on private hypothesis selection [BSKW19], which will suffice.

The main difficulty in the algorithm is in privately learning some $(\hat{\mu}, \hat{\Sigma})$ which is "vaguely close" to some $(\mu_i, \Sigma_i)$. We accomplish this task by applying the robustness-to-privacy conversion of [HKMN23], along with a carefully constructed score function, which we will describe later in this section.

**Algorithm Description:** Let $c_0 \leq 0.01$ and $C_0 > 1$ be the constants of Lemma B.6, and $c_1 \leq 0.01$ be the constant of Lemma C.8. Let $c_2 \leq 0.1 \cdot \min(c_0, c_1)$ be a sufficiently small constant. We set parameters $\eta^* = \frac{c_2 \cdot \alpha}{8k}, \eta = \frac{\eta^*}{10}, \varepsilon_0 = \frac{\varepsilon}{\sqrt{4k \log(1/\delta)}}, \delta_0 = \frac{\delta}{2k}, \beta_0 = e^{-d}$. We also set $m = \tilde{O}(d^{1.75}), N = \tilde{O}\left(m \cdot \frac{\sqrt{k \log(1/\delta)}}{\varepsilon} + \frac{\sqrt{k} \cdot \log^{3/2}(1/\delta)}{\varepsilon} + kd\right)$, and $n = N \cdot \frac{2k}{\alpha}$. Finally, we define parameters $\gamma = \tau = c_2$ and $\rho = c_2 \cdot d^{1/8}$. (See Lines 1–4 of Algorithm 2 for a more precise description of some of the parameters.)

We now define the main score function. To do so, we first set some auxiliary parameter $\eta' = \gamma' = \frac{c_2}{8C_0}$ and $\rho' = \frac{\rho}{8C_0}$. We will consider the (deterministic) algorithm $\mathcal{A}$ of Lemma B.6, where $\mathcal{A}$ is given parameters $\eta', \gamma', \rho', \beta_0$, that acts on a dataset $\mathbf{Z}$ and outputs either $\perp$ or some $(\hat{\mu}, \hat{\Sigma})$. Next, for some domain $\Omega \subset \Theta$ of "feasible" mean-covariance pairs $(\mu, \Sigma)$, we will define a function $f_\Omega$, which takes as input a dataset $\mathbf{Y}$ of size $N$ and a mean-covariance pair $(\mu, \Sigma) \in \Theta$, and outputs

$$f_\Omega(\mathbf{Y}, \mu, \Sigma) = \begin{cases} 1 & \text{if } (\mu, \Sigma) \in \Omega, \ \mathcal{A}(\mathbf{Y}) \approx_{\gamma, \rho, \tau} (\mu, \Sigma), \text{ and } \ \Pr_{\substack{\mathbf{Z} \subset \mathbf{Y} \\ |\mathbf{Z}| = m}} [\mathcal{A}(\mathbf{Z}) \approx_{\gamma, \rho, \tau} (\mu, \Sigma)] \geq \frac{2}{3} \\ 0 & \text{else}. \end{cases}$$

(2)

Finally, given a dataset $\mathbf{X}$ of size $n$ and $(\widetilde{\mu}, \widetilde{\Sigma}) \in \Theta$, we define the score function

$$\mathcal{S}_\Omega((\widetilde{\mu}, \widetilde{\Sigma}), \mathbf{X}) = \min \big\{ t : \ \exists \mathbf{X}', \mathbf{Y}', \mu, \Sigma \text{ such that}$$
$$d_H(\mathbf{X}, \mathbf{X}') = t, \mathbf{Y}' \subset \mathbf{X}', |\mathbf{Y}'| = N, (\widetilde{\mu}, \widetilde{\Sigma}) \approx_{\gamma, \tau} (\mu, \Sigma), f_\Omega(\mathbf{Y}', \mu, \Sigma) = 1 \big\}. \quad (3)$$

Note that $(\widetilde{\mu}, \widetilde{\Sigma})$ does not need to be in $\Omega$, but it must satisfy $(\widetilde{\mu}, \widetilde{\Sigma}) \approx_{\gamma, \tau} (\mu, \Sigma)$ for some $(\mu, \Sigma) \in \Omega$. Also, note that it is possible that no matter how one changes the data points, the conditions are never met (for instance if $(\widetilde{\mu}, \widetilde{\Sigma}) \not\approx_{\gamma, \tau} (\mu, \Sigma)$ for any $(\mu, \Sigma) \in \Omega$). This is not a problem: we will simply set the score to be $+\infty$ if this occurs.

With this definition of score function $\mathcal{S}_\Omega$, we can let $\mathcal{M}$ be an $(\varepsilon_0, \delta_0)$-DP algorithm based on Theorem C.3, with the settings of $n, \eta, \eta^*, \varepsilon_0, \delta_0, \beta_0$ as above, and with $D = \frac{n(n+3)}{2}$. Here, we recall that $(\widetilde{\mu}, \tilde{\Sigma})$ is viewed as $\frac{n(n+3)}{2}$-dimensional by only considering the upper-triangular part of $\tilde{\Sigma}$. We also assume the domain is $\tilde{\Theta}$ and the normalized volume is as in (1).

Given this algorithm $\mathcal{M}$ (which implicitly depends on $\Omega$), the algorithm works as follows. We will actually draw a total of $n + n'$ samples (for $n$ as above, and $n'$ to be defined later), though we start by just looking at the first $n$ samples $\mathbf{X} = \{X_1, \ldots, X_n\}$. We initially set $\Omega = \Theta$, and run the following procedure up to $k$ times, or until the algorithm outputs $\perp$. For the $j^{\text{th}}$ iteration, we use $\mathcal{M}(\mathbf{X})$ to

compute some prediction $(\hat{\mu}_j, \hat{\Sigma}_j)$ (or possibly we output $\perp$). Assuming we do not output $\perp$, we remove from $\Omega$ every $(\mu, \Sigma)$ such that $n^{-12} \cdot \hat{\Sigma}_j \preceq \Sigma \preceq n^{12} \cdot \hat{\Sigma}_j$ and $\|\hat{\Sigma}_j^{-1/2}(\mu - \hat{\mu}_j)\|_2 \leq n^{12}$.

At the end, we have computed some $\{(\hat{\mu}_j, \hat{\Sigma}_j)\}_{j=1}^{k'}$ where $0 \leq k' \leq k$. If $k' = 0$ we simply output $\perp$. Otherwise, we will draw $n' = \tilde{O}\left(\frac{d^2 k}{\alpha^2} + \frac{d^2 k}{\alpha \varepsilon}\right)$ fresh samples. We run the $\varepsilon$-DP hypothesis selection based algorithm, Algorithm 1 on the fresh samples, using parameters $G = n^{12}$ and $\{(\hat{\mu}_j, \hat{\Sigma}_j)\}_{j=1}^{k'}$.

**Pseudocode:**  We give pseudocode for the algorithm in Algorithm 2.

---

**Algorithm 2:** ESTIMATE$(X_1, X_2, \ldots, X_{n+n'} \in \mathbb{R}^d, k, \varepsilon, \delta, \alpha)$

**Input:** Samples $X_1, \ldots, X_n, X_{n+1}, \ldots, X_{n+n'}$.
**Output:** An $(\varepsilon, \delta)$-DP prediction $\{(\tilde{\mu}_i, \tilde{\Sigma}_i), \tilde{w}_i\}$.
/* Set parameters */
1 Let $c_2$ be a sufficiently small constant and $K = \text{poly} \log(d, k, \frac{1}{\alpha}, \frac{1}{\varepsilon}, \log \frac{1}{\delta})$ be sufficiently large.
2 $\eta^* \leftarrow \frac{c_2 \cdot \alpha}{8k}$, $\eta \leftarrow \frac{\eta^*}{10}$, $\varepsilon_0 \leftarrow \frac{\varepsilon}{\sqrt{4k \log(1/\delta)}}$, $\delta_0 \leftarrow \frac{\delta}{2k}$, $\beta_0 \leftarrow e^{-d}$.
3 $m = K \cdot d^{1.75}$, $N \leftarrow K \cdot \left(\frac{m + \log(1/\delta_0)}{\varepsilon_0} + kd\right)$, $n \leftarrow N \cdot \frac{2k}{\alpha}$.
4 $\gamma, \tau \leftarrow c_2, \rho \leftarrow 10 \cdot d^{1/8}$.
5 $\Theta, \Omega \leftarrow \{(\mu, \Sigma) : \mu \in \mathbb{R}^d, \Sigma \in \mathbb{R}^{d \times d}$ Symmetric, Positive Definite$\}$.
/* Get $\mathbf{X}$ */
6 Obtain samples $\mathbf{X} \leftarrow \{X_1, X_2, \ldots, X_n\}$.
/* Learn a crude approximation $(\hat{\mu}_j, \hat{\Sigma}_j)$ of the mean-covariance pairs, one at a time */
7 **for** $j = 1$ *to* $k$ **do**
8 $\quad$ Define $f_\Omega(\mathbf{Y}, \mu, \Sigma)$ and $\mathcal{S}_\Omega((\mu, \Sigma), \mathbf{X})$ as in (2) and (3), respectively.
9 $\quad$ Let $\mathcal{M}$ be the $(\varepsilon_0, \delta_0)$-DP algorithm obtained by Theorem C.3 with the score function $\mathcal{S}_\Omega$,
$\quad\quad$ with parameters $n, \eta, \eta^*, \varepsilon_0, \delta_0, \beta_0$ as defined above, $D = \frac{n(n+3)}{2}$, domain $\Theta$, and
$\quad\quad$ $p(\mu, \Sigma) = (\det \Sigma)^{-(d+2)/2}$.
10 $\quad$ $A \leftarrow \mathcal{M}(\mathbf{X})$
11 $\quad$ **if** $A \neq \perp$ **then**
12 $\quad\quad$ $\hat{\mu}_j, \hat{\Sigma}_j \leftarrow A$
13 $\quad\quad$ $\Omega \leftarrow \Omega \backslash \{(\mu, \Sigma) : n^{-12} \cdot \hat{\Sigma}_j \preceq \Sigma \preceq n^{12} \cdot \hat{\Sigma}_j$ and $\|\hat{\Sigma}_j^{-1/2}(\mu - \hat{\mu}_j)\|_2 \leq n^{12}\}$
14 $\quad$ **else**
15 $\quad\quad$ Break $\quad\quad\quad\quad\quad\quad$ // Break out of the for loop
16 $\quad$ **end**
17 **end**
/* Fine approximation, via private hypothesis selection */
18 Set $n' \leftarrow K \cdot \left(\frac{d^2 k}{\alpha^2} + \frac{d^2 k}{\alpha \varepsilon}\right)$.
19 Sample $X_{n+1}, \ldots, X_{n+n'}$, and redefine $\mathbf{X} \leftarrow \{X_{n+1}, \ldots, X_{n+n'}\}$.
20 Run Algorithm 1 on $\mathbf{X}$, $\{(\hat{\mu}_j, \hat{\Sigma}_j)\}$, with $k' = \#\{(\hat{\mu}_j, \hat{\Sigma}_j)\}$ and $G = n^{12}$.

---

## E.2  Analysis of Crude approximation

In this section, we analyze Lines 7–17 of Algorithm 2. The main goal is to show that the algorithm is private, and that for samples drawn from a mixture of Gaussians, every component $(\mu_i, \Sigma_i)$ with large enough weight $w_i$ is "vaguely close" to some $(\hat{\mu}_j, \hat{\Sigma}_j)$ computed by the algorithm.

First, we show that when the samples are actually drawn as a Gaussian mixture, then under some reasonable conditions, any $(\mu, \Sigma)$ close to a true mean-covariance pair has low score.

**Proposition E.1.** *Suppose* $\mathbf{X} = \{X_1, \ldots, X_n\}$ *is drawn from a Gaussian mixture model, with representation* $\{(w_i, \mu_i, \Sigma_i)\}_{i=1}^k$. *Then, with* $1 - O(k \cdot \beta_0)$ *probability over* $\mathbf{X}$*, for any set* $\Omega$

*of mean-covariance pairs, for all $i \in [k]$ such that $(\mu_i, \Sigma_i) \in \Omega$ and $w_i \geq \alpha/k$, and for all $(\widetilde{\mu}, \widetilde{\Sigma}) \approx_{\gamma,\tau} (\mu_i, \Sigma_i)$, $\mathcal{S}_\Omega((\widetilde{\mu}, \widetilde{\Sigma}), \mathbf{X}) = 0$.*

*Proof.* Fix any $i \in [k]$ with $w_i \geq \alpha/k$. Set $\mu = \mu_i$ and $\Sigma = \Sigma_i$. Also, let $\mathbf{X}' = \mathbf{X}$, and $\mathbf{Y}$ be a random subset of size $N$ among the samples in $\mathbf{X}$ actually drawn from $\mathcal{N}(\mu_i, \Sigma_i)$. Note that the number of such samples has distribution $\mathrm{Bin}(n, w_i)$, which for $w_i \geq \frac{\alpha}{k}$ and $n = \frac{2k}{\alpha} \cdot N$, is at least $N$ with $e^{-\Omega(N)} \leq \beta_0$ failure probability.

Then, $\mathbf{Y}$ is just $N$ i.i.d. samples from $\mathcal{N}(\mu_i, \Sigma_i)$. So, if we draw a random subset $\mathbf{Z}$ of size $m$ of $\mathbf{Y}$, it has the same distribution as $m$ i.i.d. samples from $\mathcal{N}(\mu_i, \Sigma_i)$. We apply Part 2 of Lemma B.6 (where we use parameters $\eta', \gamma', \rho'$ in the application). Note that $m \geq \widetilde{O}\left(\frac{d}{\gamma'^2} + \frac{d^2}{\rho'^2}\right)$, so with probability at least $1 - O(\beta_0)$ over $\mathbf{Z}$, $\mathcal{A}(\mathbf{Z}) \approx_{8C_0\gamma', 8C_0\rho', 8C_0\gamma'} (\mu_i, \Sigma_i)$. By our setting of $\gamma', \rho'$, this means that $\mathcal{A}(\mathbf{Z}) \approx_{\gamma,\rho,\gamma} (\mu_i, \Sigma_i)$.

In other words, for each index $i \in \binom{N}{m}$ and corresponding subset $\mathbf{Z}$ of $\mathbf{Y}$, if we let $W_i$ be the indicator that $\mathcal{A}(\mathbf{Z}) \not\approx_{\gamma,\rho,\gamma} (\mu_i, \Sigma_i)$, then $\mathbb{P}(W_i = 1) \leq O(\beta_0)$. While the values of $W_i$ are not necessarily independent, by linearity of expectation we have that $\mathbb{E}[\sum W_i] \leq \binom{N}{m} \cdot O(\beta_0)$, so the probability that $\mathbb{E}[\sum W_i] \geq \frac{1}{3} \cdot \binom{N}{m}$ is at most $O(\beta_0)$ by Markov's inequality. Moreover, because $N \geq m \geq \widetilde{O}\left(\frac{d}{\gamma'^2} + \frac{d^2}{\rho'^2}\right)$, we can apply $\mathcal{A}$ on $\mathbf{Y}$ and we again obtain that with probability $1 - O(\beta_0)$, $\mathcal{A}(\mathbf{Y}) \approx_{\gamma,\rho,\gamma} (\mu_i, \Sigma_i)$.

In summary, for any fixed $i \in [k]$ with $w_i \geq \frac{\alpha}{k}$, with probability at least $1 - O(\beta_0)$ over $\mathbf{X}$, there exists $\mathbf{Y} \subset \mathbf{X}$ of size $N$, such that $\mathcal{A}(\mathbf{Y}) \approx_{\gamma,\rho,\gamma} (\mu_i, \Sigma_i)$ and at least $2/3$ of the subsets $\mathbf{Z} \subset \mathbf{Y}$ of size $m$ have $\mathcal{A}(\mathbf{Z}) \approx_{\gamma,\rho,\gamma} (\mu_i, \Sigma_i)$. Thus, for any set $\Omega$, if $(\mu_i, \Sigma_i) \in \Omega$ then $\mathcal{S}_\Omega((\widetilde{\mu}, \widetilde{\Sigma}), \mathbf{X}) = 0$ for all $(\widetilde{\mu}, \widetilde{\Sigma}) \approx_{\gamma,\tau} (\mu_i, \Sigma_i)$. The proof follows by a union bound over all $i \in [k]$. $\qquad \square$

Next, we show that for any dataset $\mathbf{X}$, if there is even a single $(\mu^*, \Sigma^*)$ with low score, there must be a region of $(\widetilde{\mu}, \widetilde{\Sigma})$ which all has low score.

**Proposition E.2.** *Suppose that $t = \min_{\mu^*, \Sigma^*} \mathcal{S}_\Omega((\mu^*, \Sigma^*), \mathbf{X})$. Then, there exists some $\mu, \Sigma$ such that for all $(\widetilde{\mu}, \widetilde{\Sigma}) \approx_{\gamma,\tau} (\mu, \Sigma)$, we have that $\mathcal{S}_\Omega((\widetilde{\mu}, \widetilde{\Sigma}), \mathbf{X}) = t$.*

*Proof.* Fix $\mu^*, \Sigma^*$ so that $\mathcal{S}_\Omega((\mu^*, \Sigma^*), \mathbf{X}) = t$. Then, there is some $(\mu, \Sigma)$ and some $\mathbf{X}', \mathbf{Y}$ such that $\mathrm{d}_H(\mathbf{X}, \mathbf{X}') = t$, $\mathbf{Y} \subset \mathbf{X}'$, $|\mathbf{Y}| = N$, and $f_\Omega(\mathbf{Y}, \mu, \Sigma) = 1$. Thus, for any $(\widetilde{\mu}, \widetilde{\Sigma}) \approx_{\gamma,\tau} (\mu, \Sigma)$, by definition, we have that $\mathcal{S}_\Omega((\widetilde{\mu}, \widetilde{\Sigma}), \mathbf{X}) \leq t$. But since $t$ is the minimum possible score, we in fact have equality: $\mathcal{S}_\Omega((\widetilde{\mu}, \widetilde{\Sigma}), \mathbf{X}) = t$ for all such $(\widetilde{\mu}, \widetilde{\Sigma})$. $\qquad \square$

Next, we want to show that regardless of the dataset (even if not drawn from any Gaussian mixture distribution), the set of points of low score can't have that much volume.

**Proposition E.3.** *Fix a dataset $\mathbf{X}$ of size $n$. Then, the set of $\widetilde{\mu}, \widetilde{\Sigma}$ such that $\mathcal{S}_\Omega((\widetilde{\mu}, \widetilde{\Sigma}), \mathbf{X}) \leq \eta^* \cdot n$ can be partitioned into at most $\binom{n}{m}$ regions $S_i$, which is indexed by some $(\mu'_i, \Sigma'_i)$. Moreover, for all $(\widetilde{\mu}, \widetilde{\Sigma}) \in S_i$, there exists $(\mu, \Sigma)$ such that $(\widetilde{\mu}, \widetilde{\Sigma}) \approx_{\gamma,\tau} (\mu, \Sigma)$ and $(\mu, \Sigma) \approx_{8\gamma,8\rho,8\tau} (\mu'_i, \Sigma'_i)$.*

*Proof.* Pick any $(\widetilde{\mu}, \widetilde{\Sigma})$ with score at most $\eta^* \cdot n$. Let $\mathbf{X}', \mathbf{Y}', \mu, \Sigma$ be such that $\mathrm{d}_H(\mathbf{X}, \mathbf{X}') \leq \eta^* \cdot n$, $\mathbf{Y}' \subset \mathbf{X}'$, $|\mathbf{Y}'| = N$, $(\widetilde{\mu}, \widetilde{\Sigma}) \approx_{\gamma,\tau} (\mu, \Sigma)$, and $f_\Omega(\mathbf{Y}', \mu, \Sigma) = 1$. If we define $\mathbf{Y} \subset \mathbf{X}$ of size $N$ to be the corresponding subset as $\mathbf{Y}'$ is to $\mathbf{X}'$, then $\mathrm{d}_H(\mathbf{Y}, \mathbf{Y}') \leq \eta^* \cdot n = \frac{c_2}{4} \cdot N$.

Now, if we take a random subset $\mathbf{Z}'$ of size $m$ in $\mathbf{Y}'$, and look at the corresponding subset $\mathbf{Z}$ of size $m$ in $\mathbf{Y}$, by a Chernoff bound, with at least $0.99$ probability $\mathrm{d}_H(\mathbf{Z}, \mathbf{Z}') \leq \frac{c_2}{2} \cdot m$. Moreover, with at least $2/3$ probability over the random subset $\mathbf{Z}'$, $\mathcal{A}(\mathbf{Z}') \approx_{\gamma,\rho,\tau} (\mu, \Sigma)$. Therefore, there *always* exists a subset $\mathbf{Z} \subset \mathbf{X}$ of size $m$ and a set $\mathbf{Z}'$ of size $m$ such that $\mathrm{d}_H(\mathbf{Z}, \mathbf{Z}') \leq \frac{c_2}{2} \cdot m$ and $\mathcal{A}(\mathbf{Z}') \approx_{\gamma,\rho,\tau} (\mu, \Sigma)$.

Now, for any fixed $\mathbf{Z} \subset \mathbf{X}$ of size $m$, if we look at any sets $\mathbf{Z}', \mathbf{Z}''$ of size $m$ and Hamming distance at most $\frac{c_2}{2} \cdot m$ from $\mathbf{Z}$, then $\mathrm{d}_H(\mathbf{Z}', \mathbf{Z}'') \leq c_2 \cdot m$. So, by Property 1 of Lemma B.6, for every such $\mathbf{Z}'$ and $\mathbf{Z}''$ with $\mathcal{A}(\mathbf{Z}'), \mathcal{A}(\mathbf{Z}'') \neq \perp$, we must have $\mathcal{A}(\mathbf{Z}'') \approx_{\gamma,\rho,\tau} \mathcal{A}(\mathbf{Z}')$.

To complete the proof, we order the subsets $\mathbf{Z}_1, \mathbf{Z}_2, \ldots, \mathbf{Z}_{\binom{n}{m}}$ of size $m$ in $\mathbf{X}$. For each $i \leq \binom{n}{m}$, if there exists some $\mathbf{Z}'$ such that $\mathrm{d_H}(\mathbf{Z}_i, \mathbf{Z}') \leq \frac{c_2}{2} \cdot m$ and $\mathcal{A}(\mathbf{Z}') \neq \perp$, choose an arbitrary such $\mathbf{Z}'$ and let $(\mu_i', \Sigma_i') := \mathcal{A}(\mathbf{Z}')$. Otherwise, we do not define $(\mu_i', \Sigma_i')$. Then, for any $(\widetilde{\mu}, \widetilde{\Sigma})$ of score at most $\eta^* \cdot n$, there exists $(\mu, \Sigma)$, a subset $\mathbf{Z}_i \subset \mathbf{X}$ of size $m$, and a set $\mathbf{Z}_i'$ of size $m$ such that $(\widetilde{\mu}, \widetilde{\Sigma}) \approx_{\gamma, \tau} (\mu, \Sigma)$, $\mathrm{d_H}(\mathbf{Z}_i, \mathbf{Z}_i') \leq \frac{c_2}{2} \cdot m$, and $\mathcal{A}(\mathbf{Z}_i') \approx_{\gamma, \rho, \tau} (\mu, \Sigma)$. Because $\mathcal{A}(\mathbf{Z}_i') \neq \perp$ and $\mathbf{Z}_i'$ has Hamming distance at most $\frac{c_2}{2} \cdot m$ from $\mathbf{Z}_i$, this also implies that $\mathcal{A}(\mathbf{Z}_i') \approx_{\gamma, \rho, \tau} (\mu_i', \Sigma_i')$. By Proposition B.2, this means that $(\mu, \Sigma) \approx_{8\gamma, 8\rho, 8\tau} (\mu_i', \Sigma_i')$, which completes the proof. $\qquad\square$

**Proposition E.4.** *Suppose $\mathbf{X} = \{X_1, \ldots, X_n\}$ is drawn from a Gaussian mixture model, with representation $\{(w_i, \mu_i, \Sigma_i)\}_{i=1}^k$. Then, with probability at least $1 - O(\beta_0)$ over the randomness of $\mathbf{X}$, for any set $\Omega$ of mean-covariance pairs, and for every $(\widetilde{\mu}, \widetilde{\Sigma})$ with $\mathcal{S}_\Omega((\widetilde{\mu}, \widetilde{\Sigma}), \mathbf{X}) \leq \frac{N}{2}$, there exists an index $i \in [k]$ such that $\frac{1}{O(n^8)} \cdot \Sigma_i \preccurlyeq \widetilde{\Sigma} \preccurlyeq O(n^8) \cdot \Sigma_i$ and $\|\Sigma_i^{-1/2}(\widetilde{\mu} - \mu_i)\|_2 \leq O(n^6)$.*

*Proof.* We apply Part 3 of Lemma B.6, though we use $N$ to replace the value $m$ in Lemma B.6. We are assuming $N \geq 40d \cdot k$. So, by definition of score, $(\widetilde{\mu}, \widetilde{\Sigma}) \approx_{\gamma, \tau} (\mu, \Sigma)$ where $f_\Omega(\mathbf{Y}', \mu, \Sigma) = 1$, for some $\mathbf{Y}'$ which can be generated by taking a subset of $\mathbf{X}$ of size at most $N$ and altering at most $N/2$ elements. Since $f_\Omega(\mathbf{Y}', \mu, \Sigma) = 1$, this means that if $(\hat{\mu}, \hat{\Sigma}) = \mathcal{A}(\mathbf{Y}')$ then $(\hat{\mu}, \hat{\Sigma}) \approx_{\gamma, \rho, \tau} (\mu, \Sigma)$. So, by Proposition B.2, $(\widetilde{\mu}, \widetilde{\Sigma}) \approx_{8\gamma, 8\tau} (\hat{\mu}, \hat{\Sigma})$. Assuming $\gamma = \tau = c_2$ is sufficiently small, this means that $(\widetilde{\mu}, \widetilde{\Sigma}) \approx_{1/2, 1/2} (\hat{\mu}, \hat{\Sigma})$.

By Part 3 of Lemma B.6, with probability at least $1 - O(\beta_0)$, there exists $i \in [k]$ such that $\frac{1}{O(n^8)} \cdot \Sigma_i \preccurlyeq \hat{\Sigma} \preccurlyeq O(n^8) \cdot \Sigma_i$ and $\|\Sigma_i^{-1/2}(\hat{\mu} - \mu_i)\|_2 \leq O(n^6)$. However, we know that $\frac{1}{2}\hat{\Sigma} \preccurlyeq \widetilde{\Sigma} \preccurlyeq 2\hat{\Sigma}$, which means that $\frac{1}{O(n^8)} \cdot \Sigma_i \preccurlyeq \widetilde{\Sigma} \preccurlyeq O(n^8) \cdot \Sigma_i$. Moreover, $\|\hat{\Sigma}^{-1/2}(\hat{\mu} - \widetilde{\mu})\|_2 \leq 0.5$, so $\|\Sigma_i^{-1/2}(\hat{\mu} - \widetilde{\mu})\|_2 \leq O(n^4)$. Thus, by triangle inequality, we have that $\|\Sigma_i^{-1/2}(\widetilde{\mu} - \mu_i)\|_2 \leq O(n^6)$. $\qquad\square$

Now, given a set $\mathbf{X} = \{X_1, \ldots, X_n\}$ drawn from a GMM with representation $\{(w_i, \mu_i, \Sigma_i)\}_{i=1}^k$, we say $\mathbf{X}$ is *regular* if it satisfies Propositions E.1 and E.4. In other words:

1. for any set $\Omega$ of mean-covariance pairs, for all $i \in [k]$ such that $(\mu_i, \Sigma_i) \in \Omega$ and $w_i \geq \alpha/k$, and for all $(\widetilde{\mu}, \widetilde{\Sigma}) \approx_{\gamma, \tau} (\mu_i, \Sigma_i)$, $\mathcal{S}_\Omega((\widetilde{\mu}, \widetilde{\Sigma}), \mathbf{X}) = 0$.

2. for any set $\Omega$ of mean-covariance pairs, and for every $(\widetilde{\mu}, \widetilde{\Sigma})$ with $\mathcal{S}_\Omega((\widetilde{\mu}, \widetilde{\Sigma}), \mathbf{X}) \leq \frac{N}{2}$, there exists an index $i \in [k]$ such that $\frac{1}{O(n^8)} \cdot \Sigma_i \preccurlyeq \widetilde{\Sigma} \preccurlyeq O(n^8) \cdot \Sigma_i$ and $\|\Sigma_i^{-1/2}(\widetilde{\mu} - \mu_i)\|_2 \leq O(n^6)$.

When we say $\mathbf{X}$ is regular, the components $(\mu_i, \Sigma_i)$ and weights $w_i$ are implicit.

We now show that every step of the crude approximation (Lines 8–16 in Algorithm 2) will, with $1 - O(\beta_0)$ probability, find some crude approximation $(\hat{\mu}_j, \hat{\Sigma}_j)$ to some $(\mu_i, \Sigma_i)$, as long as $(\mu_i, \Sigma_i) \in \Omega$.

**Lemma E.5.** *For any $\Omega \subset \Theta$, the corresponding algorithm $\mathcal{M}$ (which depends on $\Omega$) is $(\varepsilon_0, \delta_0)$-DP on datasets $\mathbf{X}$.*

*Suppose additionally that $\mathbf{X}$ is regular, and that there exists $i \in [k]$ such that $w_i \geq \frac{\alpha}{k}$ and $(\mu_i, \Sigma_i) \in \Omega$. Then, with $1 - O(\beta_0)$ probability (just over the mechanism $\mathcal{M}$), $\mathcal{M}(\mathbf{X}) = (\widetilde{\mu}, \widetilde{\Sigma})$ and*

- *There exists $i \in [k]$ such that $\frac{1}{O(n^8)} \cdot \Sigma_i \preccurlyeq \widetilde{\Sigma} \preccurlyeq O(n^8) \cdot \Sigma_i$ and $\|\Sigma_i^{-1/2}(\widetilde{\mu} - \mu_i)\|_2 \leq O(n^6)$.*

- *There exists $(\mu, \Sigma) \in \Omega$ such that $(\widetilde{\mu}, \widetilde{\Sigma}) \approx_{\gamma, \tau} (\mu, \Sigma)$.*

*Proof.* We will apply Theorem C.3. We first need to check the necessary conditions. It follows almost immediately from the definition of $\mathcal{S}_\Omega$ that $\mathcal{S}_\Omega((\widetilde{\mu}, \widetilde{\Sigma}), \cdot)$ has the bounded sensitivity property with respect to neighboring datasets, for any fixed $\Omega, \widetilde{\mu}, \widetilde{\Sigma}$.

To check the volume condition, note that for any dataset $\mathbf{X}$ of size $n$, if $\min_{\mu,\Sigma} \mathcal{S}_\Omega((\mu,\Sigma), \mathbf{X}) \leq 0.7\eta^* n$, then by Proposition E.2, there exists $\mu, \Sigma$ such that $\mathcal{S}_\Omega((\widetilde{\mu}, \widetilde{\Sigma}), \mathbf{X}) \leq 0.8\eta^* n$ for all $(\widetilde{\mu}, \widetilde{\Sigma}) \approx_{\gamma,\tau} (\mu, \Sigma)$. Conversely, for any $\mathbf{X}$, by Proposition E.3, the set of $(\widetilde{\mu}, \widetilde{\Sigma})$ of score at most $\eta^* n$ can be partitioned into $\binom{n}{m}$ regions $S_i$, indexed by $(\mu_i', \Sigma_i')$. Moreover, each $S_i$ is contained in the set of $(\widetilde{\mu}, \widetilde{\Sigma})$ such that there exists $(\mu, \Sigma)$ such that $(\widetilde{\mu}, \widetilde{\Sigma}) \approx_{\gamma,\tau} (\mu, \Sigma) \approx_{8\gamma, 8\rho, 8\tau} (\mu_i', \Sigma_i')$.

Let us use the notation of Lemma C.8, where $\gamma_1 = \gamma$, $\gamma_2 = 8\gamma$, and $\rho_2 = 8\rho$. Then, we have that the set of points with score at most $0.8\eta^* n$ contains $\mathcal{T}_1(\mu, \Sigma)$ for some $(\mu, \Sigma)$, but the set of points with score at most $\eta^* n$ is contained in the union of $\mathcal{T}_2(\mu_i', \Sigma_i')$ for at most $\binom{n}{m}$ choices of $(\mu_i', \Sigma_i')$. So, as long as $\min_{\mu,\Sigma} \mathcal{S}_\Omega((\mu,\Sigma), \mathbf{X}) \leq 0.7\eta^* n$, we can apply Lemma C.8 to obtain

$$\frac{V_{\eta^*}(\mathbf{X})}{V_{0.8\eta^*}(\mathbf{X})} \leq \binom{n}{m} \cdot \exp\left(O\left(\frac{d^{5/3} \cdot \log d \cdot \rho_2^{2/3}}{\gamma_1^2}\right)\right) = \exp\left(O\left(d^{7/4}\log d + m\log n\right)\right),$$

since $\rho_2 = 8\rho = 8c_2 \cdot d^{1/8}$ and $\gamma_1 = \gamma = c_2$, and since $c_2$ is a constant. Thus, if

$$n \geq C \cdot \frac{O(d^{7/4}\log d + m\log n) + \log(1/\delta_0)}{\varepsilon_0 \cdot \eta^*},$$

we have that $\mathcal{M}$ is $(\varepsilon_0, \delta_0)$-DP, by Theorem C.3. This holds by our parameter settings, assuming $K$ is sufficiently large.

Next, we prove accuracy. Assume that $\mathbf{X}$ is regular. We again adopt the notation of Lemma C.8, (with $\gamma_1 = \gamma$, $\gamma_2 = 8\gamma$, and $\rho_2 = 8\rho$). By the first condition of regularity, for all $i \in [k]$ such that $(\mu_i, \Sigma_i) \in \Omega$ and $w_i \geq \alpha/k$, every $(\widetilde{\mu}, \widetilde{\Sigma}) \in \mathcal{T}_1(\mu_i, \Sigma_i)$ satisfies $\mathcal{S}_\Omega((\widetilde{\mu}, \widetilde{\Sigma}), \mathbf{X}) = 0$. We still have that the set of points of score at most $\eta^* n$ is contained in the union of at most $\binom{n}{m}$ sets $\mathcal{T}_2(\mu_i', \Sigma_i')$. Thus,

$$\frac{V_{\eta^*}(\mathbf{X})}{V_\eta(\mathbf{X})} \leq \exp\left(O\left(d^{7/4}\log d + m\log n\right)\right),$$

where we recall that we set $\eta = \frac{\eta^*}{10}$. So, by Theorem C.3, as long as

$$n \geq C \cdot \frac{O(d^{7/4}\log d + m\log n) + \log(1/(\beta_0 \cdot \eta))}{\varepsilon_0 \cdot \eta},$$

which indeed holds, $\mathcal{M}(\mathbf{X})$ outputs some $(\widetilde{\mu}, \widetilde{\Sigma})$ of score at most $2\eta n$, with $1 - \beta_0$ probability. By the second condition of regularity,, there exists an index $i \in [k]$ such that $\frac{1}{O(n^8)} \cdot \Sigma_i \preccurlyeq \widetilde{\Sigma} \preccurlyeq O(n^8) \cdot \Sigma_i$ and $\|\Sigma_i^{-1/2}(\widetilde{\mu} - \mu_i)\|_2 \leq O(n^6)$. Moreover, because $(\widetilde{\mu}, \widetilde{\Sigma})$ has finite score, $(\widetilde{\mu}, \widetilde{\Sigma}) \approx_{\gamma,\tau} (\mu, \Sigma)$ for some $(\mu, \Sigma) \in \Omega$. □

**Lemma E.6.** *Lines 7–17 of Algorithm 2 are $(\varepsilon, \delta)$-DP. Moreover, for every regular set $\mathbf{X}$, with probability at least $1 - O(k \cdot \beta_0)$ over the randomness of Algorithm 2, for every $i \in [k]$ such that $w_i \geq \frac{\alpha}{k}$, there exists $j$ such that $n^{-12} \cdot \hat{\Sigma}_j \preccurlyeq \Sigma_i \preccurlyeq n^{12} \cdot \hat{\Sigma}_j$ and $\|\hat{\Sigma}_j^{-1/2}(\mu - \hat{\mu}_j)\|_2 \leq n^{12}$.*

*Proof.* The privacy guarantee is immediate by (adaptive) advanced composition. Indeed, the algorithm $\mathcal{M}$ at each step is $(\varepsilon_0, \delta_0)$-DP, and only depends on $\mathbf{X}$ and $\Omega$, which is determined only by the output of all previous runs of $\mathcal{M}$.

Now, let's say that $\mathbf{X} = \{X_1, \ldots, X_n\}$ is regular, and we run Algorithm 2 on $\mathbf{X}$. Now, after $j \geq 0$ steps of the loop, define $P_j$ to be the set of indices $i \in [k]$ such that $w_i \geq \frac{\alpha}{k}$ and $(\mu_i, \Sigma_i) \in \Omega_j$ (where $\Omega_j$ refers to the set $\Omega$ after $j$ steps of the loop are completed). Likewise, define $Q_j$ to be the set of indices $i$ such that there exists $(\hat{\mu}, \hat{\Sigma}) \in \Omega_j$ with $\frac{1}{n^{10}} \cdot \Sigma_i \preccurlyeq \hat{\Sigma} \preccurlyeq n^{10} \cdot \Sigma_i$ and $\|\Sigma_i^{-1/2}(\hat{\mu} - \mu_i)\|_2 \leq n^{10}$. Note that $P_0 \subset Q_0 = [k]$, and that $P_j \subset Q_j$ always. Moreover, because $\Omega$ only shrinks, $P_{j+1} \subset P_j$ and $Q_{j+1} \subset Q_j$ always.

Now, after some step $j < k$, we claim that if $P_j \neq \emptyset$, then $|Q_{j+1}| \leq |Q_j| - 1$ with failure probability at most $O(\beta_0)$, i.e., $Q$ decreases in size for the next step if $P$ isn't currently empty. The failure probability is only over the randomness of $\mathcal{M}$ at each step, and holds for any regular dataset $\mathbf{X}$. A union bound says the total failure probability is $O(k \cdot \beta_0)$.

To see why this holds, note that if some index $i \in P_j$, then $(\mu_i, \Sigma_i) \in \Omega_j$. So, we can apply Lemma E.5. At step $j + 1$, we find some $(\hat{\mu}_{j+1}, \hat{\Sigma}_{j+1})$, with the following properties. First, there exists an index $i' \in [k]$ with $\frac{1}{O(n^8)} \cdot \Sigma_{i'} \preccurlyeq \hat{\Sigma}_{j+1} \preccurlyeq O(n^8) \cdot \Sigma_{i'}$ and $\|\Sigma_{i'}^{-1/2}(\hat{\mu}_{j+1} - \mu_{i'})\|_2 \leq O(n^6)$. Second, $(\hat{\mu}_{j+1}, \hat{\Sigma}_{j+1}) \approx_{\gamma, \tau} (\mu, \Sigma)$, for some $(\mu, \Sigma) \in \Omega_j$.

We claim this means $i' \in Q_j$. To see why, it suffices to show that $\frac{1}{n^{10}} \cdot \Sigma_{i'} \preccurlyeq \Sigma \preccurlyeq n^{10} \cdot \Sigma_{i'}$ and $\|\Sigma_{i'}^{-1/2}(\mu - \mu_{i'})\|_2 \leq n^{10}$, by definition of $Q_j$ and because $(\mu, \Sigma) \in \Omega_j$. However, we know that $\frac{1}{O(n^8)} \cdot \Sigma_{i'} \preccurlyeq \hat{\Sigma}_{j+1} \preccurlyeq O(n^8) \cdot \Sigma_{i'}$ and $\frac{1}{2}\hat{\Sigma}_{j+1} \preccurlyeq \Sigma \preccurlyeq 2\hat{\Sigma}_{j+1}$, which implies $\frac{1}{n^{10}} \cdot \Sigma_{i'} \preccurlyeq \Sigma \preccurlyeq n^{10} \cdot \Sigma_{i'}$. Also, we know that $\|\Sigma_{i'}^{-1/2}(\hat{\mu}_{j+1} - \mu_{i'})\|_2 \leq O(n^6)$ and $\|\Sigma^{-1/2}(\mu - \hat{\mu}_{j+1})\|_2 \leq \tau \leq 1$ The latter inequality along with the fact that $\Sigma \preccurlyeq n^{10} \cdot \Sigma_{i'}$ implies that $\|\Sigma_{i'}^{-1/2}(\mu - \hat{\mu}_{j+1})\|_2 \leq n^5$. So, by triangle inequality, $\|\Sigma_{i'}^{-1/2}(\mu - \mu_{i'})\|_2 \leq O(n^6) \leq n^{10}$.

Next, we claim that $i' \notin P_{j+1}$, so $i' \notin Q_{j+1}$. Indeed, we have that $\frac{1}{O(n^8)} \cdot \Sigma_{i'} \preccurlyeq \hat{\Sigma}_{j+1} \preccurlyeq O(n^8) \cdot \Sigma_{i'}$, which immediately means $\frac{1}{n^{10}} \cdot \hat{\Sigma}_{j+1} \preccurlyeq \frac{1}{O(n^8)} \cdot \hat{\Sigma}_{j+1} \preccurlyeq \Sigma_{i'} \preccurlyeq O(n^8) \cdot \hat{\Sigma}_{j+1} \preccurlyeq n^{10} \cdot \hat{\Sigma}_{j+1}$. Also, $\|\Sigma_{i'}^{-1/2}(\hat{\mu}_{j+1} - \mu_{i'})\|_2 \leq O(n^6)$, and since $\Sigma_{i'} \preccurlyeq n^{10} \cdot \hat{\Sigma}_{j+1}$, this means $\|\hat{\Sigma}_{j+1}^{-1/2}(\mu_{i'} - \hat{\mu}_{j+1})\|_2 \leq O(n^{11}) \leq n^{12}$. Thus, the algorithm will remove $(\mu_{i'}, \Sigma_{i'})$ from $\Omega$ at step $j + 1$, so $i' \notin Q_{j+1}$.

Thus, either $P_j$ is empty, or $Q_j$ decreases in size to $Q_{j+1}$ for each $0 \leq j \leq k + 1$. This implies that $P_k$ is empty, i.e., for every $i \in [k]$ such that $w_i \geq \frac{\alpha}{k}$, $(\mu_i, \Sigma_i) \notin \Omega_j$. This can only happen if each such $(\mu_i, \Sigma_i)$ was removed at some point. So, for every such $i$, there was some index $j$ such that $n^{-12} \cdot \hat{\Sigma}_j \preccurlyeq \Sigma_i \preccurlyeq n^{12} \cdot \hat{\Sigma}_j$ and $\|\hat{\Sigma}_j^{-1/2}(\mu - \hat{\mu}_j)\|_2 \leq n^{12}$. $\qquad\square$

### E.3 Summary and Completion of Analysis

We first quickly summarize how to put everything together to prove Theorem 1.4. To prove our main result, need to ensure that our algorithm is private, uses few enough samples, and accurately learns the mixture. Privacy will be simple, as we have shown in Lemma E.6 that the crude approximation is private, and the hypothesis selection is private as shown in Appendix D. The sample complexity will come from the settings of $n$ and $n'$ in lines 3 and 18 of the algorithm, and from our setting of parameters. Finally, we showed in Lemma E.6 that we have found a set of $(\hat{\mu}_j, \hat{\Sigma}_j)$, of at most $k$ mean-covariance pairs, such that every true $(\mu_i, \Sigma_i)$ of sufficiently large weight is crudely approximated by some $(\hat{\mu}_j, \hat{\Sigma}_j)$. Indeed, this is exactly the situation for which we can apply the result on private hypothesis selection (Lemma D.3, based on [BSKW19]).

We are now ready to complete the analysis and prove Theorem 1.4.

*Proof of Theorem 1.4.* By Lemma E.6, note that the algorithm up to Line 17 is $(\varepsilon, \delta)$-DP with respect to $X_1, \ldots, X_n$, and does not depend on $X_{n+1}, \ldots, X_{n+n'}$. Assuming $\{(\hat{\mu}_j, \hat{\Sigma}_j)\}$ from these lines is fixed, Lines 18–20 are $(\varepsilon, \delta)$-DP with respect to $X_{n+1}, \ldots, X_{n+n'}$, by Lemma D.3, and do not depend on $X_1, \ldots, X_n$. So, the overall algorithm is $(\varepsilon, \delta)$-DP.

Next, we verify accuracy. Note that by Lemma E.6, and for $G = n^{12}$, the sets $(\hat{\mu}_j, \hat{\Sigma}_j)$ that we find satisfy the conditions for Lemma D.3, with failure probability at most $O(k \cdot \beta_0)$. This probability is at most 0.1, assuming $e^d$ is significantly larger than $k$. So, it suffices for $n'$, the number of samples used in Line 18 of Algorithm 2, to satisfy $n' \geq O\left(\frac{d^2 k \cdot \log(G \cdot k \sqrt{d}/\alpha)}{\alpha^2} + \frac{d^2 k \cdot \log(G \cdot k \sqrt{d}/\alpha)}{\alpha \varepsilon}\right) = \widetilde{O}\left(\frac{d^2 k}{\alpha^2} + \frac{d^2 k}{\alpha \varepsilon}\right)$, where the last part of the bound holds by our assumptions on $G$ and $n$. Thus, the total sample complexity is

$$n + n' = \widetilde{O}\left(\frac{kd^2}{\alpha^2} + \frac{kd^2 + d^{1.75}k^{1.5}\log^{0.5}(1/\delta) + k^{1.5}\log^{1.5}(1/\delta)}{\alpha \varepsilon} + \frac{k^2 d}{\alpha}\right).$$

By Lemma D.3, with failure probability at most 0.1, Lines 18–20 of the algorithm will successfully output a hypothesis which is a mixture of $k$ Gaussians, with total variation distance at most $O(\alpha)$ from the right answer. So, the overall success probability is at least 0.8.

Finally, we note that the assumption that $e^d$ is much larger than $k$ can be made WLOG, by padding $d$ to be a sufficiently large multiple of $\log k$. Namely, if given samples $X_i = (X_{i,1}, \ldots, X_{i,d})$, we can add coordinates $X_{i,d+1}, \ldots, X_{O(\log k)} \sim \mathcal{N}(0,1)$. Then, if the original samples were a mixture of $k$ Gaussians $\mathcal{N}(\mu_i, \Sigma_i)$, the new distribution is a mixture of $k$ Gaussians $\mathcal{N}\left(\begin{pmatrix} \mu_i \\ 0 \end{pmatrix}, \begin{pmatrix} \Sigma_i & 0 \\ 0 & I \end{pmatrix}\right)$, with the same mixing weights. So, we can learn the mixture distribution up to total variation distance $\alpha$ in the larger space, and then remove all except the first $d$ coordinates, to output our final answer. This will not affect the sample complexity by more than a $\mathrm{poly}(\log k)$ factor. $\qquad\square$

# F  The Univariate Setting

In this section, we provide the algorithm and analysis for Theorem 1.5. We first describe and provide pseudocode for the algorithm, and then we prove that it can privately learn mixtures of Gaussians with low sample complexity.

We will use $\sigma_i := \sqrt{\Sigma_i}$ to denote the standard deviation of a univariate Gaussian $\mathcal{N}(\mu_i, \Sigma_i)$.

## F.1  Algorithm

As in the algorithm in Appendix E, we will actually draw a total of $n + n'$ samples. We start off by only considering the first $n$ data points $\mathbf{X} = \{X_1, \ldots, X_n\}$, where each $X_j \in \mathbb{R}$. Now, let $\mathbf{Y} = \{Y_1, \ldots, Y_n\}$ be the sorted version of $\mathbf{X}$, i.e., $Y_1 \leq Y_2 \leq \cdots \leq Y_n$, and $(X_1, \ldots, X_n)$ and $(Y_1, \ldots, Y_n)$ are the same up to permutation. Finally, for each $j \leq n-1$, define $Z_j$ to be the ordered pair $(Y_j, Y_{j+1} - Y_j)$, and define $\mathbf{Z} = \mathbf{Z}(\mathbf{X})$ to be the unordered multiset of $Z_j$'s. (Note that $\mathbf{Z}$ depends deterministically on $\mathbf{X}$.)

The algorithm now works as follows. Suppose we have data $\mathbf{X} = \{X_1, \ldots, X_n\}$. After sorting to obtain $Y_1, \ldots, Y_n$, let $r_j = Y_j$, $s_j = Y_{j+1} - Y_j$, and $Z_j = (r_j, s_j)$. So, $\mathbf{Z}(\mathbf{X}) = \{(r_j, s_j)\}_{1 \leq j \leq n-1}$, viewed as an unordered set. Note that every $s_j \geq 0$, and the $r_j$'s are in nondecreasing order. We will create a set of buckets that can be bijected onto $\mathbb{Z}^2$. For each $Z_j = (r_j, s_j)$, if $s_j > 0$ we assign $Z_j$ to the bucket labeled $(a,b) \in \mathbb{Z}^2$ if $2^a \leq s_j < 2^{a+1}$ and $b \cdot n^5 \cdot 2^a \leq r_j < (b+1) \cdot n^5 \cdot 2^a$. If $s_j = 0$, we do not assign $Z_j$ to any bucket.

For each element $e \in \mathbb{Z}^2$, we keep track of the number of indices $j \in [n-1]$ with $Z_j$ sent to $e$. In other words, for $e = (a,b)$, we define the count $c_e = \#\{j : 2^a \leq s_j < 2^{a+1}, b \cdot n^5 \cdot 2^a \leq r_j < (b+1) \cdot n^5 \cdot 2^a\}$. For each $c_e$, we sample an independent draw $g_e \sim \mathrm{TLap}(1, \varepsilon/10, \delta/10)$, and define $\tilde{c}_e = c_e + g_e$. Finally, we let $S = \{(\hat{\mu}_i, \hat{\Sigma}_i)\}$ be the set of pairs $(b \cdot n^5 \cdot 2^a, 2^{2a})$ where $e = (a,b)$ satisfies $\tilde{c}_e > \frac{100}{\varepsilon} \log \frac{1}{\delta}$.

Hence, we have computed some $\{(\hat{\mu}_i, \hat{\Sigma}_i)\}_{i=1}^{k'}$, for some $k' \geq 0$. (We will show in the analysis that $k' \leq n-1$ always, so $k'$ is finite.) If $k' = 0$ we simply output $\perp$. Otherwise, we will draw $\tilde{O}\left(\frac{k}{\alpha^2} + \frac{k}{\alpha\varepsilon}\right)$ fresh samples. We run the $\varepsilon$-DP hypothesis selection based algorithm, Algorithm 1 on the fresh samples, using parameters $G = n^{10}$ and $\{(\hat{\mu}_i, \hat{\Sigma}_i)\}_{i=1}^{k'}$.

**Pseudocode.**  We give pseudocode for the algorithm in Algorithm 3.

## F.2  Analysis

We start by analyzing Lines 1–19, which generates the set $S = \{(\hat{\mu}_i, \hat{\Sigma}_i)\}$ of candidate mean-covariance pairs.

First, we note a very basic proposition.

**Proposition F.1.**  *Let $\varepsilon, \delta \leq 1$. Then, with probability 1, $\mathrm{TLap}(1, \varepsilon/10, \delta/10)$ is at most $\frac{100}{\varepsilon} \cdot \log \frac{1}{\delta}$ in absolute value.*

*Proof.*  By definition of Truncated Laplace, $|\mathrm{TLap}(1, \varepsilon/10, \delta/10)| \leq \frac{1}{(\varepsilon/10)} \cdot \log\left(1 + \frac{e^{\varepsilon/10}-1}{2\delta/10}\right)$ with probability 1. For $\varepsilon, \delta \leq 1$, this is less than $\frac{100}{\varepsilon} \cdot \log \frac{1}{\delta}$. $\qquad\square$

---

**Algorithm 3:** ESTIMATE1D$(X_1, X_2, \ldots, X_{n+n'} \in \mathbb{R}, k, \varepsilon, \delta, \alpha)$

---

**Input:** Samples $X_1, \ldots, X_n, X_{n+1}, \ldots, X_{n+n'}$.
**Output:** An $(\varepsilon, \delta)$-DP prediction $\{(\widetilde{\mu}_i, \widetilde{\Sigma}_i), \widetilde{w}_i\}$.
/* Set parameters */

1 Let $K = \text{poly} \log \left(k, \frac{1}{\alpha}, \frac{1}{\varepsilon}, \log \frac{1}{\delta}\right)$ be sufficiently large.
2 Set $n \leftarrow K \cdot \frac{k \log(1/\delta)}{\alpha \varepsilon}$.
   /* Get **X** */
3 Obtain samples $\mathbf{X} \leftarrow \{X_1, X_2, \ldots, X_n\}$.
   /* Obtain some candidate mean-covariance pairs $(\hat{\mu}_i, \hat{\Sigma}_i)$ */
4 Let $\mathbf{Y}$ be $\mathbf{X}$ in sorted (nondecreasing) order.
5 Set $c_{(a,b)} \leftarrow 0$ for all $(a, b) \in \mathbb{Z}^2$.
6 **for** $j = 1$ *to* $n - 1$ **do**
7 $\quad$ Set $r_j \leftarrow Y_j, s_j \leftarrow Y_{j+1} - Y_j, Z_j \leftarrow (r_j, s_j)$.
8 $\quad$ **if** $s_j > 0$ **then**
9 $\quad\quad$ Let $a \leftarrow \lfloor \log_2(r_j) \rfloor, b \leftarrow \lfloor \frac{r_j}{n^5 \cdot 2^a} \rfloor$.
10 $\quad\quad$ $c_{a,b} \leftarrow c_{a,b} + 1$.
11 $\quad$ **end**
12 **end**
13 $S \leftarrow \emptyset$.
14 **for** $(a, b) \in \mathbb{Z}^2$ **do**
15 $\quad$ Sample $\tilde{c}_{a,b} \leftarrow c_{a,b} + \text{TLap}(1, \varepsilon/10, \delta/10)$.
16 $\quad$ **if** $\tilde{c}_{a,b} > \frac{100}{\varepsilon} \log \frac{1}{\delta}$ **then**
17 $\quad\quad$ $S \leftarrow S \cup \{(b \cdot n^5 \cdot 2^a, 2^{2a})\}$.
18 $\quad$ **end**
19 **end**
   /* Fine approximation, via private hypothesis selection */
20 Set $n' \leftarrow K \cdot \left(\frac{d^2 k}{\alpha^2} + \frac{d^2 k}{\alpha \varepsilon}\right)$.
21 Sample $X_{n+1}, \ldots, X_{n+n'}$, and redefine $\mathbf{X} \leftarrow \{X_{n+1}, \ldots, X_{n+n'}\}$.
22 Run Algorithm 1 on $\mathbf{X}$, $S = \{(\hat{\mu}_i, \hat{\Sigma}_i)\}$, with parameters $d = 1, k' = \#\{(\hat{\mu}_i, \hat{\Sigma}_i)\}$, and $G = n^3$.

---

Next, we show that $S$ is not too large.

**Lemma F.2.** *The number of $e = (a, b) \in \mathbb{Z}^2$ such that $\tilde{c}_e > \frac{100}{\varepsilon} \cdot \log \frac{1}{\delta}$ is at most $n - 1$. Thus, $|S| \leq n - 1$.*

*Proof.* Note that every $Z_j$ only increases the count of a single $c_e$, so it suffices to prove that $\tilde{c}_e > \frac{100}{\varepsilon} \cdot \log \frac{1}{\delta}$ only if $c_e \geq 1$.

To prove this, suppose that $c_e = 0$. Then, $\tilde{c}_e \sim \text{TLap}(1, \varepsilon/10, \delta/10) \leq \frac{100}{\varepsilon} \cdot \log \frac{1}{\delta}$, by Proposition F.1. Thus, $\tilde{c}_e > \frac{100}{\varepsilon} \cdot \log \frac{1}{\delta}$ can only happen if $c_e > 0$, meaning $c_e \geq 1$. $\qquad\square$

Next, we show that for every Gaussian component $(\mu_i, \sigma_i)$ of sufficiently large weight $w_i$, there are several pairs $Z_j = (Y_j, Y_{j+1} - Y_j)$ that crudely approximate $(\mu_i, \sigma_i)$.

**Lemma F.3.** *Let $n \geq K \cdot \frac{k \log(1/\delta)}{\alpha \varepsilon}$, where $K = \text{poly} \log(k, \frac{1}{\alpha}, \frac{1}{\varepsilon}, \log \frac{1}{\delta})$ is sufficiently large. Let $\mathbf{X}$ be $n$ i.i.d. samples from a GMM with representation $\{(w_i, \mu_i, \Sigma_i)\}_{i=1}^k$. Then, with failure probability at least $0.99$, for every $i \in [k]$ with $w_i \geq \frac{\alpha}{k}$, there are at least $\frac{\alpha}{8k} \cdot n$ indices $j$ such that $Y_j \in [\mu_i - \sigma_i, \mu_i + \sigma_i]$ and $\frac{\sigma_i}{10^4 \cdot n^4} \leq Y_{j+1} - Y_j \leq 2\sigma_i$.*

*Proof.* Fix some $i \in [k]$ such that $w_i \geq \frac{\alpha}{k}$. By a Chernoff bound, since $w_i \cdot n \geq \frac{\alpha}{k} \cdot n$ is a sufficiently large multiple of $\log k$, with failure probability at most $\frac{1}{1000k}$, the number of data points $X_j$ from component $i$ is at least $\frac{\alpha}{2k} \cdot n$. Let us condition on $T_i \subset [n]$, the set of indices coming from the $i^{\text{th}}$ mixture component, and condition on $|T_i| \geq \frac{\alpha}{2k} \cdot n$. Then, by a Chernoff bound, with failure

probability at most $\frac{1}{1000k}$, at least $\frac{\alpha}{4k} \cdot n$ points $X_r : r \in T_i$ are in the interval $[\mu_i - \sigma_i, \mu_i + \sigma_i]$. Next, note that for any $X_r, X_{r'} \sim \mathcal{N}(\mu_i, \sigma_i^2)$, $X_r - X_{r'} \sim \mathcal{N}(0, 2\sigma_i^2)$, and thus has magnitude at least $\frac{\sigma_i}{10^4 n^3}$ with at most $\frac{1}{1000n^3}$ failure probability. So, by a union bound over all $r \neq r' \in T_i$, with at most $\frac{1}{1000n} \leq \frac{1}{1000k}$ failure probability, all data points $X_r, X_{r'}$ for $r \neq r' \in T_i$ are separated by at least $\frac{\sigma_i}{10^4 n^3}$.

So, with at least $\frac{3}{1000k}$ failure probability, there is a subset $U_i \subset T_i$ of size at least $\frac{\alpha}{4k} \cdot n$, such that for every $r, r' \in U_i$, $|X_r - X_{r'}|$, and for every $r \in U_i$, $X_r \in [\mu_i - \sigma_i, \mu_i + \sigma_i]$. By a union bound, this holds simultaneously for all $i \in [k]$ with $w_j \geq \frac{\alpha}{k}$, with probability at least $0.997$.

Conditioned on this event, if we consider the elements $\mathbf{X}$ in sorted order (i.e., $\mathbf{Y}$), between every consecutive pair $X_r, X_{r'} : r, r' \in U_i$ (after sorting), we know $X_{r'} - X_r \geq \frac{\sigma_i}{10^4 \cdot n^3}$. So, there exist some $X_r \leq Y_j, Y_{j+1} \leq X_{r'}$ such that $Y_{j+1} - Y_j \geq \frac{\sigma_i}{10^4 \cdot n^4}$. Hence, for every $i \in [k]$ with $w_i \geq \frac{\alpha}{k}$, there exist at least $|U_i| - 1 \geq \frac{\alpha}{4k} - 1 \geq \frac{\alpha}{8k}$ indices $j$ such that $Y_{j+1} - Y_j \geq \frac{\sigma_i}{10^4 \cdot n^4}$. Moreover, note that $Y_j, Y_{j+1} \in [\mu_i - \sigma_i, \mu_i + \sigma_i]$, so $Y_{j+1} - Y_j \leq 2\sigma_i$. Hence, conditioned on the event from the previous paragraph, the lemma holds. $\square$

Given the above lemma, we can prove that for every $(\mu_i, \Sigma_i)$ in the Gaussian mixture with at least $\frac{\alpha}{k}$ weight, there is some corresponding crude approximation $(\hat{\mu}, \hat{\Sigma}) \in S$.

**Lemma F.4.** *Assume that the conditions and result of Lemma F.3 holds. Then, there exists $(\hat{\mu}, \hat{\Sigma}) \in S$ such that $n^{-10} \cdot \hat{\Sigma} \preccurlyeq \Sigma_i \preccurlyeq n^{10} \cdot \hat{\Sigma}$ and $\|\hat{\Sigma}^{-1/2}(\mu_i - \hat{\mu})\|_2 \leq n^6$.*

*Proof.* Fix any $i$ with $w_i \geq \frac{\alpha}{k}$. Then, for any $j$ with $Y_j, Y_{j+1} \in [\mu_i - \sigma_i, \mu_i + \sigma_i]$ and $\frac{\sigma_i}{10^4 \cdot n^4} \leq Y_{j+1} - Y_j \leq 2\sigma_i$, the index $i$ contributes to the count of some bucket $e = (a, b)$, where $\lfloor \log_2 \frac{\sigma_i}{10^5 \cdot n^4} \rfloor \leq a \leq \lfloor \log_2(2\sigma_i) \rfloor$. Moreover, $Y_j \in [b \cdot n^5 \cdot 2^a, (b+1) \cdot n^5 \cdot 2^a)$. Therefore, if we consider the set $V_i$ of $e = (a, b)$ such that $\lfloor \log_2 \frac{\sigma_i}{10^5 \cdot n^4} \rfloor \leq a \leq \lfloor \log_2(2\sigma_i) \rfloor$ and $[b \cdot n^5 \cdot 2^a, (b+1) \cdot n^5 \cdot 2^a) \cap [\mu_i - \sigma_i, \mu_i + \sigma_i]$ is nonempty, the sum of the counts $c_e$ across such $e \in V_i$ is at least $\frac{\alpha}{8k} \cdot n$. But there are at most $O(\log n)$ choices of $a$, and since $n^5 \cdot 2^a > 2\sigma$, there are at most two choices for $b$ for any fixed $a$. So, $|V_i| \leq O(\log n)$, and $\sum_{e \in V_i} c_e \geq \frac{\alpha}{8k} \cdot n$. Assuming that $\frac{\alpha}{8k} \cdot n \geq O(\log n \cdot \frac{1}{\varepsilon} \log \frac{1}{\delta})$, i.e., $n \geq K \cdot \frac{k \log(1/\delta)}{\alpha \varepsilon}$ for a sufficiently large polylogarithmic factor $K = \text{poly} \log \left(k, \frac{1}{\alpha}, \frac{1}{\varepsilon}, \log \frac{1}{\delta}\right)$, one of these buckets $e = (a, b) \in V_j$ must have $c_e > \frac{200}{\varepsilon} \log \frac{1}{\delta}$. In this case, $\tilde{c}_e = c_e + \text{TLap}(1, \varepsilon/10, \delta/10) > \frac{100}{\varepsilon} \log \frac{1}{\delta}$ by Proposition F.1, so we include the pair $(\hat{\mu}, \hat{\Sigma}) := (b \cdot n^5 \cdot 2^a, 2^{2a})$ in $S$.

Hence, there exists $(a, b) \in V_i$ such that $(\hat{\mu}, \hat{\Sigma}) := (b \cdot n^5 \cdot 2^a, 2^{2a})$ in $S$. By definition of $V_i$, $\frac{\sigma_i}{n^5} \leq 2^a \leq 2\sigma_i$, so $\frac{1}{n^{10}} \cdot \Sigma_i \leq 2^{2a} \leq 4\Sigma_i \leq n^{10} \cdot \Sigma_i$. Moreover, $|\hat{\Sigma}^{-1/2}(\mu_i - \hat{\mu})| = 2^{-a} \cdot |\mu_i - \hat{\mu}|$. But the definition of $V_i$ means $\hat{\mu} = b \cdot n^5 \cdot 2^a$ satisfies $\hat{\mu} \leq \mu_i + \sigma_i$ and $\hat{\mu} \geq \mu_i - \sigma_i - n^5 \cdot 2^a \geq \mu_i - (2n^5 + 1) \cdot \sigma_i$. So, $|\hat{\mu} - \mu_i| \leq (2n^5 + 1) \cdot \sigma_i \leq n^6 \cdot 2^a$. Thus, $|\hat{\Sigma}^{-1/2}(\hat{\mu} - \mu_i)| \leq n^6$. $\square$

We now prove that the mechanism (until Line 19) is differentially private. First, we note the following auxiliary claim.

**Lemma F.5.** *Let $\mathbf{X}, \mathbf{X}'$ be adjacent datasets of size $n$ (i.e., only differing on a single element). Then, the corresponding sets $\mathbf{Z} = \mathbf{Z}(\mathbf{X})$ and $\mathbf{Z}' = \mathbf{Z}(\mathbf{X}')$ differ in distance at most 3, where by distance we mean that there exists a permutation of the elements in $\mathbf{Z}$ and $\mathbf{Z}'$, respectively, such that at most three indices $i \leq n - 1$ satisfy $Z_i \neq Z_i'$.*

*Proof.* Note that for the sorted versions $\mathbf{Y}, \mathbf{Y}'$ of $\mathbf{X}, \mathbf{X}'$, respectively, we can convert from $\mathbf{Y}$ to $\mathbf{Y}'$ by removing one data point and adding one more data point, without affecting the order of any other data points.

Suppose we remove some $Y_j$ from $Y$. If $j = 1$, then this just removes $Z_1$, and if $j = n$, then this just removes $Z_{n-1}$. If $j \geq 2$, this modifies $Z_{j-1}$ and removes $Z_j$. Likewise, if we add some new $Y_{j'}$, this will either add one new ordered pair to $\mathbf{Z}$ (if $Y_{j'}$ is either the smallest or largest element), or replace one ordered pair in $\mathbf{Z}$ with two new pairs. Therefore, if we remove a $Y_j$ and then add a $Y_{j'}$, this will change at most 3 of the ordered pairs in $\mathbf{Z}$. $\square$

We now prove privacy.

**Lemma F.6.** *The set* $S = \{(\hat{\mu}_j, \hat{\Sigma}_j)\}$ *of candidate mean-covariance pairs is* $(\varepsilon, \delta)$-DP *with respect to* $\mathbf{X} = \{X_1, \ldots, X_n\}$.

*Proof.* Let $\mathbf{X}, \mathbf{X}'$ be adjacent datasets of size $n$. Then, the corresponding sets $\mathbf{Z}, \mathbf{Z}'$ (after a possible permutation) differ in 3 elements. Therefore, if we let $\{\tilde{c}_e\}_{e \in \mathbb{Z}^2}$ be the counts for $\mathbf{Z}$ and $\{\tilde{c}'_e\}_{e \in \mathbb{Z}^2}$ be the counts for $\mathbf{Z}'$, we have that $\|\tilde{c}_e - \tilde{c}'_e\|_1 \leq 6$. Because changing a single count $c_e$ by 1 leads to $(\varepsilon/10, \delta/10)$-DP, overall, the counts $\{\tilde{c}_e\}$ will satisfy $(\varepsilon, \delta)$-DP. Finally, $S$ is a deterministic function of the noisy counts $\tilde{c}_e$, and therefore must also be $(\varepsilon, \delta)$-DP. $\square$

Finally, we can incorporate the fine approximation (i.e., Lines 20–22 of Algorithm 3) and prove Theorem 1.5.

*Proof of Theorem 1.5.* By Lemma F.6, the algorithm up to Line 19 is $(\varepsilon, \delta)$-DP with respect to $X_1, \ldots, X_n$, and does not depend on $X_{n+1}, \ldots, X_{n+n'}$. Assuming $S = \{(\hat{\mu}_i, \hat{\Sigma}_i)\}$ from these lines is fixed, Lines 20–22 are $(\varepsilon, \delta)$-DP with respect to $X_{n+1}, \ldots, X_{n+n'}$, by Lemma D.3, and do not depend on $X_1, \ldots, X_n$. So, the overall algorithm is $(\varepsilon, \delta)$-DP.

Next, we verify accuracy. Note that by Lemma F.4, and for $G = n^{10}$, the sets $(\hat{\mu}_i, \hat{\Sigma}_i)$ that we find satisfy the conditions for Lemma D.3, with failure probability at most 0.01. Moreover, the size of $S = \{(\hat{\mu}_i, \hat{\Sigma}_i)\}$ is at most $n$, by Lemma F.2. So, because $d = 1$, it suffices for $n'$, the number of samples used in Line 20 of Algorithm 3, to satisfy $n' \geq O\left(\frac{k \cdot \log(G \cdot kn/\alpha)}{\alpha^2} + \frac{k \cdot \log(G \cdot kn/\alpha)}{\alpha \varepsilon}\right) = \widetilde{O}\left(\frac{k}{\alpha^2} + \frac{k}{\alpha \varepsilon}\right)$, where the last part of the bound holds by our assumptions on $G$ and $n$. Thus, the total sample complexity is

$$n + n' = \widetilde{O}\left(\frac{k \log(1/\delta)}{\alpha \varepsilon} + \frac{k}{\alpha^2}\right).$$

By Lemma D.3, with failure probability at most 0.1, Lines 20–22 of the algorithm will successfully output a hypothesis which is a mixture of $k$ Gaussians, with total variation distance at most $O(\alpha)$ from the right answer. So, the overall success probability is at least 0.8. $\square$

## G   Lower Bound

In this section, we prove Theorem 1.6. The proof of the lower bound will, at a high level, follow from known lower bounds for privately learning a single Gaussian [KV18, KMS22a], which we now state.

**Theorem G.1.** *[KV18] For some sufficiently small constant* $c^*$, $(\varepsilon, \delta)$-*privately learning an arbitrary univariate Gaussian* $\mathcal{N}(\mu, \sigma^2)$ *up to total variation distance* $c^*$ *requires* $\Omega\left(\frac{\log(1/\delta)}{\varepsilon}\right)$ *samples.*

*Moreover, this lower bound holds even if we are promised that* $|\mu| \leq (1/\delta)^C$ *for a sufficiently large constant* $C$, *and* $\sigma = 1$.

Note that this lower bound immediately implies the same lower bound for general dimension $d$.

**Theorem G.2.** *[KMS22a] Let* $\alpha$ *be at most a sufficiently small constant, and let* $\delta \leq \left(\frac{\alpha \varepsilon}{d}\right)^C$ *for a sufficiently large constant* $C$. *Then,* $(\varepsilon, \delta)$-*privately learning a* $d$-*dimensional Gaussian* $\mathcal{N}(\mu, \Sigma)$ *up to total variation distance* $\alpha$ *requires* $\tilde{\Omega}\left(\frac{d^2}{\alpha \varepsilon}\right)$ *samples.*

*Moreover, this lower bound holds even if we are promised that* $\mu = \mathbf{0}$, *and* $I \preccurlyeq \Sigma \preccurlyeq 2I$.

Note that a tighter version of the above theorem has been proved in [Nar23, PH24]; we also refer the reader to [BUV14].

*Proof Sketch of Theorem 1.6.* First, the lower bound of $\frac{kd^2}{\alpha^2}$ is already known – see [ABH$^+$18].

The lower bound of $\frac{kd^2}{\alpha \varepsilon}$ will follow from Theorem G.2. To explain how, we consider $k$ distinct Gaussians $\mathcal{N}(\mu_i, \Sigma_i)$, where the means $\mu_i$ are known and very far away from each other, and $I \preccurlyeq \Sigma_i \preccurlyeq 2I$ are unknown. The overall mixture that we will try to learn is the uniform mixture over $\mathcal{N}(\mu_i, \Sigma_i)$, i.e., every weight $w_i = 1/k$. By making them very far away from each other, we are

making learning the full mixture equivalent to learning each component (on average). Namely, even if we are given the information of which Gaussian each sample comes from, we will need to learn at least 2/3 of the Gaussians up to total variation distance $O(\alpha)$, to learn the full mixture up to total variation distance $\alpha$. Hence, we will need at least $k$ times as many samples as for learning a single Gaussian, which means we need $\tilde{\Omega}\left(\frac{kd^2}{\alpha\varepsilon}\right)$ total samples.

The lower bound of $\frac{k\log(1/\delta)}{\alpha\varepsilon}$ will follow from Theorem G.1. Note that it suffices to prove the lower bound in the univariate case. We plant $k$ distinct Gaussians $\mathcal{N}(\mu_i, 1)$, where the $\mu_i$ are very far away from each other, i.e., pairwise $|\mu_i - \mu_j| \gg (1/\delta)^{10C}$. We also assume that $\mu_1 = 0$ is known, and the remaining $\mu_i$ are unknown but we are promised the value of each $\mu_i$ up to error $(1/\delta)^C$. The overall mixture will have the first Gaussian $\mathcal{N}(0, 1)$ of weight $w_1 = 1 - \alpha/c^*$, and the remaining Gaussians $\mathcal{N}(\mu_i, 1)$ each have weight $w_i = \alpha/(c^* \cdot (k-1))$. Even if we are given the information of which Gaussian component each sample comes from, to learn the overall mixture up to error $\alpha$, we need to learn at least 2/3 of the small-weight components, each up to total variation distance $O(c^*)$. Hence, we will need $\frac{\log(1/\delta)}{\varepsilon}$ samples from most of the small-weight components. Since the small weight components have weight $\Theta(\alpha/k)$, we need $\Omega\left(\frac{k\log(1/\delta)}{\alpha\varepsilon}\right)$ total samples. $\square$

We now give a formal proof of Theorem 1.6.

## G.1 Formal Proof of Theorem 1.6

First, we note two lemmas that will be helpful in proving the Theorem.

**Lemma G.3.** *Let $\alpha \in [0, 1)$. Let $f$ be a probability density function over $\mathbb{R}^d$. Assume $g : \mathbb{R}^d \to \mathbb{R}^{\geq 0}$ exists such that*

$$\int_{\mathbb{R}^d} |f(x) - g(x)| \ \mathrm{d}x \leq \alpha.$$

*Then there exists $h$ such that $h$ is a probability density function over $\mathbb{R}^d$, and*

$$\int_{\mathbb{R}^d} |f(x) - h(x)| \ \mathrm{d}x \leq 2\alpha.$$

*Proof.* We know that

$$\left| \int_{\mathbb{R}^d} f(x) \ \mathrm{d}x - \int_{\mathbb{R}^d} g(x) \ \mathrm{d}x \right| \leq \int_{\mathbb{R}^d} |f(x) - g(x)| \ \mathrm{d}x \leq \alpha.$$

Therefore, $G := \int_{\mathbb{R}^d} g(x) \ \mathrm{d}x = 1 \pm \alpha$.

Now two cases are possible either $G < 1$, or $G > 1$, otherwise we are done. If $G < 1$, let $h(x)$ be the density function corresponding to the following distribution: with probability $G$ take a sample from $g(x)/G$, and with probability $1 - G$ select $\mathbf{0}$. Then

$$\int_{\mathbb{R}^d} |f(x) - h(x)| \ \mathrm{d}x \leq 1 - G + \int_{\mathbb{R}^d} |f(x) - g(x)| \ \mathrm{d}x \leq 2\alpha,$$

as desired. If $G > 1$, let $h(x)$ be a density function as follows: $\forall x : 0 \leq h(x) \leq g(x)$, and $\int_{\mathbb{R}^d} h(x) \ \mathrm{d}x = 1$. It is easy to see such an $h$ exists by greedily picking $h$. Then we have $\int_{\mathbb{R}^d} |g(x) - h(x)| \ \mathrm{d}x = \int_{\mathbb{R}^d} g(x) - h(x) \ \mathrm{d}x \leq G - 1 \leq \alpha$. Therefore, we can write

$$\int_{\mathbb{R}^d} |f(x) - h(x)| \ \mathrm{d}x \leq \int_{\mathbb{R}^d} |f(x) - g(x)| \ \mathrm{d}x + \int_{\mathbb{R}^d} |g(x) - h(x)| \ \mathrm{d}x \leq 2\alpha,$$

as desired.

$\square$

**Lemma G.4.** *Suppose $w \in (0, 1)$ is fixed. Suppose an $(\varepsilon, \delta)$ differentially private algorithm exists that takes $n$ samples from the mixture $D = wD_1 + (1 - w)D_2$, and learns $D_1$ in total variation distance up to error $\alpha$, with success probability $1 - \beta$. Moreover, assume while sampling it is known which component the sample is sampled from. Then there exists an $(\varepsilon, \delta)$ differentially private algorithm $\mathcal{A}$ that takes $nw/\gamma$ samples from $D_1$ and outputs an estimate of $D_1$ up to total variation distance $\alpha$, with success probability $1 - \beta - \gamma$.*

*Proof.* We can view the sampling procedure of the mixture as sampling a random variable $t \sim \text{Bin}(n, w)$, and taking $t$ samples from $D_1$ and $n - t$ samples from $D_2$. In order to make an algorithm using $nw/\gamma$ samples we can take that many samples from $D_1$, and then sample $t$ from $\text{Bin}(n, w)$, and run $\mathcal{A}$ on $n$ data points constructed as follows: If $t$ is smaller than $nw/\gamma$, use $t$ of the samples taken from $D_1$ and set the rest to 0, If $t$ is larger than $nw/\gamma$, just run $\mathcal{A}$ on all zeroes. Finally output the output of $\mathcal{A}$ on this input. Clearly, this would be $(\varepsilon, \delta)$ differentially private. We know the Algorithm succeeds with probability $1 - \beta$, over the random coins of the algorithm and the randomness of sampling, if the input is sampled from $wD_1 + (1 - w)D_2$. From Markov's inequality, we know $\mathbb{P}[t \leq nw/\gamma] \geq 1 - \gamma$. Therefore, our constructed sample is drawn i.i.d from $wD_1 + (1 - w)D_2$ with probability at least $1 - \gamma$, where $D_2$ is the fixed 0 distribution. Therefore, there exists an $(\varepsilon, \delta)$ differentially private algorithm that takes $nw/\gamma$ many samples from $D_1$ and outputs an estimate of $D_1$ up to total variation distance $\alpha$, with success probability $1 - \beta - \gamma$, as desired. $\qquad\square$

We now prove Theorem 1.6.

*Proof.* We prove the lower bound terms one by one. The first term $\frac{kd^2}{\alpha^2}$ (the non private term) is known by previous work [ABH+18]. We prove the lower bound for the second term and the last term here using Theorems G.1 and G.2.

Let's prove the second term $\frac{kd^2}{\alpha\varepsilon}$. We apply Theorem G.2, this theorem implies that for any $\alpha$ smaller than a sufficiently small constant, and $\delta \leq (\frac{\alpha\varepsilon}{d})^C$ for a sufficiently large $C$, any $(\varepsilon, \delta)$ differentially private algorithm $\mathcal{A}$ taking $n$ samples in $\mathbb{R}^d$, satisfying

$$\forall \Sigma \text{ such that } I \preccurlyeq \Sigma \preccurlyeq 2I : \mathbb{P}_{X \sim \mathcal{N}(0, \Sigma)^{\otimes n}, \, \mathcal{A}\text{'s internal random bits}}[\mathrm{d}_{\mathrm{TV}}(\mathcal{A}(X), \mathcal{N}(0, \Sigma)) \leq \alpha] > 0.6,$$

must also satisfy $n = \tilde{\Omega}(\frac{d^2}{\alpha\varepsilon})$.

Let $\mu_i's$ be $k$ distinct vectors in $\mathbb{R}^d$ each having $\ell_2$ distance $M$ from each other, for $M$ to be set later. To see why such a set exists, we can take $\mu_i = Mie_1$, where $e_1$ is the first unit vector. Now consider the following set of Gaussians: $D_i = \mathcal{N}(\mu_i, \Sigma_i)$, where $\mu_i$'s are known and constructed as above and $\Sigma_i$ unknown. Consider the uniform mixture $D$ over these Gaussians, with weights $w_i = 1/k$. We also assume that when sampling from this distribution we know that which component the samples came from. Consider an $(\varepsilon, \delta)$ differentially private algorithm $\mathcal{A}$ that takes $n$ samples from $D$ and outputs a distribution $\hat{D}$ such that $\mathrm{d}_{\mathrm{TV}}(D, \hat{D}) \leq \alpha$ with probability $2/3$.

Now consider a sample from $D_i$, from standard Gaussian tail bounds we know that at least $1 - \exp(-M^2/800)$ fraction of the mass of $D_i$ is contained within a ball of radius $M/10$, around $\mu_i$. Let $B_i$ denote this ball, and note that $B_i$'s are disjoint.

Let $f, f_i, \hat{f}$ be the probability density functions corresponding to $D, D_i, \hat{D}$ respectively. Assuming, we are in the success regime, we can write

$$2\alpha \geq 2\,\mathrm{d}_{\mathrm{TV}}(D, \hat{D}) = \int_{\mathbb{R}^d} \left| f(x) - \hat{f}(x) \right| \, \mathrm{d}x \geq \sum_{i=1}^{k} \int_{B_i} \left| f(x) - \hat{f}(x) \right| \, \mathrm{d}x.$$

Now let $\mathcal{B} \subseteq [k]$ be the set of indices $i$ such that $\int_{B_i} \left| f(x) - \hat{f}(x) \right| \, \mathrm{d}x \geq 200\alpha/k$, and $\mathcal{G}$ be its complement. Then we have that $|\mathcal{B}| \leq k/100$. Therefore, there exists $\mathcal{G}$ such that $|\mathcal{G}| \geq 0.99k$, and $\forall i \in \mathcal{G} : \int_{B_i} \left| f(x) - \hat{f}(x) \right| \, \mathrm{d}x \leq 200\alpha/k$. Assume $i \in \mathcal{G}$ is one such index. Note that $f(x) = \sum_{j=1}^{k} f_j(x)/k$. Therefore, we can write

$$
\begin{aligned}
\int_{B_i} \left| f_i(x)/k - \hat{f}(x) \right| \, \mathrm{d}x &\leq \int_{B_i} \left| f(x) - \hat{f}(x) \right| \, \mathrm{d}x + \int_{B_i} |f_i(x) - f(x)| \, \mathrm{d}x \\
&\leq 200\alpha/k + \max_{j \neq i} \mathbb{P}_{X \sim D_j}[X \notin B_j] \\
&\leq 200\alpha/k + \exp(-M^2/800) \\
&\leq 300\alpha/k,
\end{aligned}
$$

where the last inequality comes from taking $M \geq 30\sqrt{\log(k/100\alpha)}$. We show that using the distribution $\hat{D}$, we can construct an answer to the problem of learning the Gaussian $D_i$. To do so first take $g_i$ to be equal to $k\hat{f}(x)$ over $B_i$ and 0 everywhere else. We have

$$\int_{\mathbb{R}^d} |f_i(x) - g_i(x)| \, \mathrm{d}x = \int_{B_i} \left| f_i(x) - k\hat{f}(x) \right| \, \mathrm{d}x + \int_{\mathbb{R}^d \setminus B_i} f_i(x) \, \mathrm{d}x \leq 300\alpha + 100\alpha/k \leq 400\alpha.$$

Now we may apply Lemma G.3, and deduce that given $\hat{D}$ we can construct probability density functions and distributions $\hat{D}_i$'s such that $\mathrm{d}_{\mathrm{TV}}(\hat{D}_i, D_i) \leq 800\alpha$, for all $i \in \mathcal{G}$. To recap, so far we have shown that given an $(\varepsilon, \delta)$ differentially private algorithm that takes as inputs samples from our constructed mixture of Gaussians and outputs a density $\hat{D}$ that has total variation distance at most $\alpha$, from the ground truth distribution with success probability $2/3$, we can use $\hat{D}$ to construct densities $\hat{D}_i$ such that $\mathrm{d}_{\mathrm{TV}}(\hat{D}_i, D_i) \leq 800\alpha$, for $0.99k$ of the indices $i$. This implies that there exists a fixed index $i$ for which the component $D_i$ is learned up to error $800\alpha$ with success probability $2/3 - 0.01 \geq 0.65$. Applying Lemma G.4, implies that there must exist an $(\varepsilon, \delta)$ differentially private algorithm that takes $100n/k$ samples from $D_i$ and estimates its density up to total variation distance $800\alpha$, with success probability at least $0.6$. Therefore, applying Theorem G.2, we conclude that $n = \tilde{\Omega}(\frac{kd^2}{\alpha\varepsilon})$.

Now let's prove the last term $\frac{k\log(1/\delta)}{\alpha\varepsilon}$. We aim to apply Theorem G.1. Let $\mu_i$'s be $k$ distinct values in $\mathbb{R}$, each having distance $M$ from each other, for $M \gg (1/\delta)^{10C}$ to be set later, where $\mu_1 = 0$. It is easy to see such a set exists. Now consider the following set of Gaussians: $D_i = \mathcal{N}(\mu_i, 1)$, where $\mu_i$'s are known up to $\log(1/\delta)^C$, and $\mu_1 = 0$ is also known. Consider the mixture $D$ over these Gaussians, with weights $w_1 = 1 - \alpha/c^*$, and $w_i = \alpha/(c^*(k-1))$. We also assume that when sampling from the mixture we know which component each sample comes from. Consider an $(\varepsilon, \delta)$ differentially private algorithm $\mathcal{A}$ that takes $n$ samples from $D$ and outputs a distribution $\hat{D}$ such that $\mathrm{d}_{\mathrm{TV}}(D, \hat{D}) \leq \alpha$ with probability $2/3$.

Now consider a sample from $D_i$, from standard Gaussian tail bounds we know that at least $1 - \exp(-M^2/200)$ fraction of the mass of $D_i$ is contained within a ball of radius $M/10$, around $\mu_i$. Let $B_i$ denote this ball, and note that $B_i$'s are disjoint.

Let $f, f_i, \hat{f}$ be the probability density functions corresponding to $D, D_i, \hat{D}$ respectively. Assuming, we are in the success regime, similar to the proof of the previous term, we can show that there exists a set $\mathcal{G}$ of indices such that $|\mathcal{G}| \geq 0.99k$, and $\forall i \in \mathcal{G} : \int_{B_i} \left| f(x) - \hat{f}(x) \right| \, \mathrm{d}x \leq 200\alpha/k$. Moreover, with a similar argument as the previous term for $i \in \mathcal{G}$ we can say as long as $M \geq 20\sqrt{\log(k/100\alpha)}$, $\int_{B_i} \left| w_i f_i(x) - \hat{f}(x) \right| \, \mathrm{d}x \leq 300\alpha/k$. We show that using the distribution $\hat{D}$, we can construct an answer to the problem of learning the Gaussian $D_i$, for $i \neq 1$. To do so take $g_i$ to be equal to $\hat{f}(x)/w_i$, over $B_i$ and 0 everywhere else. We have

$$\int_{\mathbb{R}^d} |f_i(x) - g_i(x)| \, \mathrm{d}x = \int_{B_i} \left| f_i(x) - \hat{f}(x)/w_i \right| \, \mathrm{d}x + \int_{\mathbb{R}^d \setminus B_i} f_i(x) \, \mathrm{d}x \leq \frac{300\alpha}{kw_i} + \frac{100\alpha}{k} \leq 400c^*.$$

Now we may apply Lemma G.3, and deduce that given $\hat{D}$, we can construct probability density functions and distributions $\hat{D}_i$'s such that $\mathrm{d}_{\mathrm{TV}}(\hat{D}_i, D_i) \leq 800c^*$, for all $i \in \mathcal{G}$. To recap, so far we have shown that given an $(\varepsilon, \delta)$ differentially private algorithm that takes as inputs samples from our constructed mixture of Gaussians and outputs density $\hat{D}$ that has total variation distance at most $\alpha$ from the ground truth distribution with success probability $2/3$, we can use $\hat{D}$ to construct densities $\hat{D}_i$ such that $\mathrm{d}_{\mathrm{TV}}(\hat{D}_i, D_i) \leq 800c^*$, for $0.99k$ of the indices $i$. This implies that there exists a fixed index $i \neq 1$, for which the component $D_i$ is learned up to error $800c^*$, with success probability $2/3 - 0.01 \geq 0.65$. Applying Lemma G.4, implies that there must exist an $(\varepsilon, \delta)$ differentially private algorithm that takes $100nw_i$ samples from $D_i$ and estimates its density up to total variation distance $800c^*$, with success probability $0.6$. Therefore, applying Theorem G.1, and noting that $w_i = \alpha/(c^*(k-1))$ we conclude that $n = \Omega(\frac{k\log(1/\delta)}{\alpha\varepsilon})$. $\qquad\square$

# H  Omitted Proofs

In this section, we prove Proposition B.2, Theorem B.3, and Lemma C.7.

## H.1  Proof of Proposition B.2

First, we note a basic fact.

**Fact H.1.** *For any $x, y \in [0.9, 1.1]$, we have that $(xy - 1)^2 \leq 4((x - 1)^2 + (y - 1)^2)$.*

*Proof.* Note that $|xy - 1| = |x - 1 + x(y - 1)| \leq |x - 1| + |x| \cdot |y - 1| \leq \sqrt{2} \cdot (|x - 1| + |y - 1|)$. Thus, $(xy - 1)^2 \leq 2(|x - 1| + |y - 1|)^2 \leq 4((x - 1)^2 + (y - 1)^2)$. □

We now prove Proposition B.2.

*Proof.* Let $J_1 := \Sigma_2^{-1/2} \Sigma_1^{1/2}$ and $J_2 := \Sigma_3^{-1/2} \Sigma_2^{1/2}$.

First, assume $(\mu_1, \Sigma_1) \approx_{\gamma, \rho, \tau} (\mu_2, \Sigma_2)$. This means $\|J_1 J_1^\top - I\|_{op} \leq \gamma$ and $\|J_1 J_1^\top - I\|_F \leq \rho$. We then have that $\Sigma_1^{-1/2} \Sigma_2 \Sigma_1^{-1/2} = J_1^{-1}(J_1^{-1})^\top = (J_1^\top J_1)^{-1}$. Now, note that $J_1 J_1^\top$ and $J_1^\top J_1$ are both symmetric and have the same eigenvalues. If we call these eigenvalues $\lambda_1, \ldots, \lambda_d$, then the eigenvalues of $\Sigma_1^{-1/2} \Sigma_2 \Sigma_1^{-1/2}$ are $\lambda_1^{-1}, \ldots, \lambda_d^{-1}$. Now, our assumption $(\mu_1, \Sigma_1) \approx_{\gamma, \rho, \tau} (\mu_2, \Sigma_2)$ implies that $1 - \gamma \leq \lambda_i \leq 1 + \gamma$ and $\sum(1 - \lambda_i)^2 \leq \rho^2$. This means that, assuming $\gamma \leq 0.1$, $1 - 2\gamma \leq \frac{1}{1+\gamma} \leq \lambda_i^{-1} \leq \frac{1}{1-\gamma} \leq 1 + 2\gamma$, and $\sum \left(1 - \lambda_i^{-1}\right)^2 \leq \sum(1 - \lambda_i)^2 \cdot \lambda_i^{-2} \leq 2 \cdot \sum(1 - \lambda_i)^2 = 2\rho^2$. This means that $\|\Sigma_1^{-1/2} \Sigma_2 \Sigma_1^{-1/2}\|_{op} \leq 2\gamma$ and $\|\Sigma_1^{-1/2} \Sigma_2 \Sigma_1^{-1/2}\|_F \leq 2\rho$.

Finally, $\Sigma_1^{-1/2}(\mu_1 - \mu_2) = J_1^{-1} \Sigma_2^{-1/2}(\mu_1 - \mu_2)$. Because $\Sigma_2^{-1/2}(\mu_1 - \mu_2)$ has magnitude at most $\tau$ by our assumption, $J_1^{-1} \Sigma_2^{-1/2}(\mu_1 - \mu_2)$ has magnitude at most the maximum singular value of $J_1^{-1}$ times $\tau$. But every singular value of $J_1^{-1}$ is some $\lambda_i^{-1/2}$ which is at most 2, so $\|\Sigma_1^{-1/2}(\mu_2 - \mu_1)\|_2 = \|J_1^{-1} \Sigma_2^{-1/2}(\mu_1 - \mu_2)\|_2 \leq 2\tau$.

Next, assume $(\mu_1, \Sigma_1) \approx_{\gamma, \rho, \tau} (\mu_2, \Sigma_2)$ and $(\mu_2, \Sigma_2) \approx_{\gamma, \rho, \tau} (\mu_3, \Sigma_3)$. First, note that $\Sigma_3^{-1/2} \Sigma_1 \Sigma_3^{-1/2} = J_2 J_1 J_1^\top J_2^\top = (J_2 J_1)(J_2 J_1)^\top$. If the eigenvalues of $J_1 J_1^\top$ are $\{\lambda_i\}$ and the eigenvalues of $J_2 J_2^\top$ are $\{\lambda_i'\}$, then the singular values of $J_1$ and $J_2$ are $\{\sqrt{\lambda_i}\}$ and $\{\sqrt{\lambda_i'}\}$, respectively. By our assumption, $1 - \gamma \leq \lambda_i, \lambda_i' \leq 1 + \gamma$, which means that $\sqrt{1 - \gamma} \leq \sqrt{\lambda_i}, \sqrt{\lambda_i'} \leq \sqrt{1 + \gamma}$. Thus, the singular values of $J_2 J_1$ are between $1 - \gamma$ and $1 + \gamma$, which means that the eigenvalues of $(J_2 J_1)(J_2 J_1)^\top$ are between $(1 - \gamma)^2$ and $(1 + \gamma)^2$. Hence, for $\gamma \leq 0.1$, $\|J_2 J_1 J_1^\top J_2^\top - I\|_{op} \leq 4\gamma$.

Assume that $\lambda_i$, $\lambda_i'$ are in decreasing order. We now consider the $k^{\text{th}}$ largest singular value of $J_2 J_1$. If $\sigma_k := \sqrt{\lambda_k}$ is the $k^{\text{th}}$ largest singular value of $J_1$ and $\sigma_k' := \sqrt{\lambda_k'}$ is the $k^{\text{th}}$ largest singular value of $J_2$, by Corollary A.5 there exist subspaces $V_k, V_k'$ of dimension $d - k + 1$ such that $\|J_1 v\|_2 \leq \sigma_k \|v\|_2$ for all $v \in V_k$ and $\|J_2 v\|_2 \leq \sigma_k' \|v\|_2$ for all $v \in V_k'$. Therefore, for every $v \in V_k \cap J_1^{-1} V_k'$ (note that $J_1$ is invertible since $J_1 J_1^\top$ has all eigenvalues between $1 - \gamma$ and $1 + \gamma$) we have that $\|J_2 J_1 v\|_2 \leq \sigma_k' \|J_1 v\|_2 \leq \sigma_k \sigma_k' \|v\|_2$. Because $V_k$ and $J_1^{-1} V_k'$ both have dimension $d - k + 1$, their intersection has dimension at least $d - 2k + 2$. So, there is a subspace of dimension at least $d - 2k + 2$ such that every $v$ in the subspace has $\|J_2 J_1 v\|_2 \leq \sigma_k \sigma_k' \cdot \|v\|_2$.

Thus, the $(2k - 1)^{\text{th}}$ largest singular value of $J_2 J_1$ is at most $\sigma_k \sigma_k'$, so the $(2k - 1)^{\text{th}}$ largest eigenvalue of $J_2 J_1 J_1^\top J_2^\top$ is at most $\lambda_k \lambda_k'$. The same argument, looking at the smallest singular values, tells us that the $(2k - 1)^{\text{th}}$ smallest eigenvalue of $J_2 J_1 J_1^\top J_2^\top$ is at least $\lambda_{d-k+1} \lambda_{d-k+1}'$. Thus, for any $t$, the $t^{\text{th}}$ largest eigenvalue of $J_2 J_1 J_1^\top J_2^\top$ is at most $\lambda_{\lfloor(t+1)/2\rfloor} \lambda_{\lfloor(t+1)/2\rfloor}'$ and at least $\lambda_{\lceil(d+t)/2\rceil} \lambda_{\lceil(d+t)/2\rceil}'$.

Overall, this means that

$$
\begin{aligned}
\|J_2 J_1 J_1^\top J_2^\top - I\|_F^2 &\leq \sum_{t=1}^{d} \max\left( (\lambda_{\lfloor (t+1)/2 \rfloor} \lambda'_{\lfloor (t+1)/2 \rfloor} - 1)^2, (\lambda_{\lceil (d+t)/2 \rceil} \lambda'_{\lceil (d+t)/2 \rceil} - 1)^2 \right) \\
&\leq \sum_{t=1}^{d} \left( (\lambda_{\lfloor (t+1)/2 \rfloor} \lambda'_{\lfloor (t+1)/2 \rfloor} - 1)^2 + (\lambda_{\lceil (d+t)/2 \rceil} \lambda'_{\lceil (d+t)/2 \rceil} - 1)^2 \right) \\
&= 2 \cdot \sum_{i=1}^{d} (\lambda_i \lambda'_i - 1)^2 \\
&\leq 8 \cdot \sum_{i=1}^{d} (\lambda_i - 1)^2 + 8 \cdot \sum_{i=1}^{d} (\lambda'_i - 1)^2 \\
&\leq 8 \cdot (\|J_1 J_1^\top - I\|_F^2 + \|J_2 J_2^\top - I\|_F^2) \leq 16\rho^2,
\end{aligned}
$$

where the fourth line uses Fact H.1. Thus, $\|\Sigma_3^{-1/2} \Sigma_1 \Sigma_3^{-1/2} - I\|_F \leq 4\rho$.

Finally, note that $\|\Sigma_3^{-1/2}(\mu_1 - \mu_3)\|_2 \leq \|\Sigma_3^{-1/2}(\mu_1 - \mu_2)\|_2 + \|\Sigma_3^{-1/2}(\mu_2 - \mu_3)\|_2 = \|J_2 \Sigma_2^{-1/2}(\mu_1 - \mu_2)\|_2 + \|\Sigma_3^{-1/2}(\mu_2 - \mu_3)\|_2$. By our assumptions, both $\|\Sigma_2^{-1/2}(\mu_1 - \mu_2)\|_2$ and $\|\Sigma_3^{-1/2}(\mu_2 - \mu_3)\|_2$ are at most $\tau$, and $J_2$ has operator norm at most $1.1 \leq 2$, which means that $\|\Sigma_3^{-1/2}(\mu_1 - \mu_3)\|_2 \leq 3\tau$. $\qquad\square$

## H.2 Proof of Theorem B.3

First, we note a series of known results that will be key to proving the theorem. We start with the bound for robust covariance estimation in spectral error.

**Lemma H.2** (e.g., [DK22, Exercise 4.3]). *Fix any $\eta \in (0, \eta_0)$, where $\eta_0 < 0.01$ is a small universal constant, and fix any $\beta \in (0,1)$. There is a (deterministic, inefficient) algorithm $\mathcal{A}_1$ with the following property. Let $\Sigma \in \mathbb{R}^{d \times d}$ be any covariance matrix, and let $\mathbf{X} = \{X_1, \ldots, X_n\} \sim \mathcal{N}(0, \Sigma)$, where $n \geq O((d + \log(1/\beta))/\eta^2)$. Then, with probability at least $1 - \beta$ over the randomness of $\mathbf{X}$, for any $\eta$-corruption $\mathbf{X}' = \{X'_1, \ldots, X'_n\}$ of $\mathbf{X}$, $\mathcal{A}_1(\mathbf{X}')$ outputs $\hat{\Sigma}_1$ such that $\|\Sigma^{-1/2} \hat{\Sigma}_1 \Sigma^{-1/2} - I\|_{op} \leq O(\eta)$. Importantly, $\mathcal{A}_1$ may have knowledge of $\eta$ and $\beta$, but does not have knowledge of $\mathbf{X}$ or $\Sigma$.*

Next, we prove how to robustly estimate the covariance up to Frobenius error. We start with the following structural lemma.

**Lemma H.3.** *There exists a universal constant $c \in (0, 0.1)$ with the following property. Fix any $1 \leq k \leq d$, and let $n \geq \widetilde{O}(d \cdot k)$ be a sufficiently large (i.e., $n \geq dk \cdot (C_3 \log(dk))^{C_4}$ for some absolute constants $C_3, C_4$). If we sample i.i.d. $\mathbf{X} = \{X_1, \ldots, X_n\} \sim \mathcal{N}(0, I)$, then with probability at least $1 - e^{-cn}$ over $\mathbf{X}$, for all symmetric matrices $P \subset \mathbb{R}^{d \times d}$ of rank at most $k$ and Frobenius norm 1, and for all subsets $S \subset [n]$ of size at least $(1 - c) \cdot n$, $\left| \frac{1}{n} \sum_{i \in S} \langle X_i X_i^\top - I, P \rangle \right| \leq 0.1$.*

*Proof.* Consider any fixed $P \subset \mathbb{R}^{d \times d}$ of rank $k$ and Frobenius norm 1, and fix any integer $m$. For any data points $X_1, \ldots, X_m \overset{i.i.d.}{\sim} \mathcal{N}(0, I)$, let $X = (X_1, \ldots, X_m) \in \mathbb{R}^{m \cdot d}$ be the concatenation of $X_1, \ldots, X_m$, and let $Q \in \mathbb{R}^{(md) \times (md)}$ be the block matrix

$$
\begin{pmatrix}
P & \mathbf{0} & \cdots & \mathbf{0} \\
\mathbf{0} & P & \cdots & \mathbf{0} \\
\vdots & \vdots & \ddots & \vdots \\
\mathbf{0} & \mathbf{0} & \cdots & P
\end{pmatrix}.
$$

Then, $\frac{1}{m}\sum_{i=1}^{m}\langle X_iX_i^\top - I, P\rangle = \frac{1}{m}\sum_{i=1}^{m}\left(X_i^\top P X_i - \text{Tr}(P)\right) = \frac{1}{m}\left(X^\top Q X - \text{Tr}(Q)\right)$. By the Hanson-Wright inequality, we have that for any fixed $\|P\|_F \leq 1$, and for any $t \leq 1$,

$$\mathbb{P}\left(\left|\frac{1}{m}\left(X^\top Q X - \text{Tr}(Q)\right)\right| > t\right) = \mathbb{P}\left(\left|X^\top Q X - \text{Tr}(Q)\right| > m \cdot t\right)$$

$$\leq 2 \cdot \exp\left(-\min\left(\frac{\Omega(mt)^2}{m \cdot \|P\|_F^2}, \frac{\Omega(mt)}{\|P\|_{op}}\right)\right)$$

$$\leq 2 \cdot \exp\left(-\Omega(m \cdot t^2)\right).$$

By setting $t = 0.01$, we have that for some universal constant $c_1$,

$$\mathbb{P}\left(\left|\frac{1}{m}\sum_{i=1}^{m}\langle X_iX_i^\top - I, P\rangle\right| > 0.01\right) \leq 2e^{-c_1 m}.$$

Now, we draw $X_1, \ldots, X_n \sim \mathcal{N}(0, I)$, and take a union bound over all subsets $S \subset [n]$ of size at least $(1-c)n$ (with $m = |S|$) and a union bound over a net of possible matrices $P$. The number of options for $S$ is at most $\sum_{i \leq cn}\binom{n}{i} \leq (e/c)^{cn}$. For $P$, we can choose a $1/n^{10}$-sized net over the Frobenius norm metric (i.e., the distance between two matrices $P_1, P_2$ is $\|P_1 - P_2\|_F$) for each of the $k$ nonzero eigenvalues and eigenvectors in the unit $d$-dimensional sphere, which has size at most $n^{100d \cdot k}$. Therefore, by a union bound, the probability that $\left|\frac{1}{n}\sum_{i \in S}\langle X_iX_i^\top - I, P'\rangle\right| \leq 0.01$ for every $|S| \geq (1-c)n$ and every $P'$ in the net is at least $1 - (2e/c)^{cn} \cdot n^{100d \cdot k} \cdot e^{-c_1 n/2}$.

Finally, we consider $P$ outside of the net. For any symmetric $P$ of rank $k$ and Frobenius norm 1, it has Frobenius distance at most $1/n^8$ from some $P'$ in the net. Let us consider the event that every $\|X_i\|_2^2 \leq 10n$, which for $n \geq d$ occurs with failure probability at most $2n \cdot e^{-c_1 n}$ by Hanson-Wright. Under this event, $\langle X_iX_i^\top - I, P - P'\rangle \leq \|X_iX_i^\top - I\|_F \cdot \|P - P'\|_F \leq (10n + \sqrt{d})/n^8 \leq 0.01$.

As a result, with failure probability at most $(2e/c)^{cn} \cdot n^{100d \cdot k} \cdot e^{-c_1 n/2} + 2n \cdot e^{-c_1 n} \leq e^{-cn}$ (assuming $c$ is sufficiently small), we have both properties. Namely, $\left|\frac{1}{n}\sum_{i \in S}\langle X_iX_i^\top - I, P'\rangle\right| \leq 0.01$ for every $|S| \geq (1-c)n$ and every $P'$ in the net, and for any $P$, $\langle X_iX_i^\top - I, P - P'\rangle 0.01$ for the closest $P'$ in the net to $P$. Overall, this means that for all such $P$ and $S$, $\left|\frac{1}{n}\sum_{i \in S}\langle X_iX_i^\top - I, P\rangle\right| \leq 0.1$. $\square$

We prove another lemma which contrasts with Lemma H.3.

**Lemma H.4.** *Fix any $2 \leq \rho \leq \sqrt{d}$. Let $k = 4d/\rho^2$, and let $n \geq \widetilde{O}(d \cdot k)$ and $c \in (0, 0.1)$ be as in Lemma H.3. Fix any covariance matrix $\Sigma$ and let $\mathbf{X} = \{X_1 \ldots, X_n\} \sim \mathcal{N}(0, \Sigma)$. Then, with probability at least $1 - e^{-cn}$, for every $\widetilde{\Sigma}$ such that $0.95 \cdot \Sigma \preccurlyeq \widetilde{\Sigma} \preccurlyeq 1.05 \cdot \Sigma$ and $\|\Sigma^{-1/2}\widetilde{\Sigma}\Sigma^{-1/2} - I\|_F \geq \rho$, for every symmetric matrix $P \in \mathbb{R}^{d \times d}$ of rank at most $k$ and Frobenius norm 1, and for every $S \subset [n]$ of size at least $(1-c)n$,*

$$\left|\frac{1}{n}\sum_{i \in S}\langle \widetilde{\Sigma}^{-1/2}X_iX_i^\top\widetilde{\Sigma}^{-1/2} - I, P\rangle\right| \geq 0.7.$$

*Proof.* Let $J = \widetilde{\Sigma}^{-1/2}\Sigma^{1/2}$, and let $Y_i = \Sigma^{-1/2}X_i$. Then, $\widetilde{\Sigma}^{-1/2}X_i = JY_i$, which means that
$$\widetilde{\Sigma}^{-1/2}X_iX_i^\top\widetilde{\Sigma}^{-1/2} - I = JY_iY_i^\top J^\top - I = J(Y_iY_i^\top - I)J^\top + (JJ^\top - I).$$

From now on, we assume that for all $S \subset [n]$ of size at least $(1-c) \cdot n$, and for all symmetric matrices $P$ of rank $k$ and Frobenius norm 1, $\left|\frac{1}{n}\sum_{i \in S}\langle Y_iY_i^\top - I, P\rangle\right| \leq 0.1$. This happens with at least $e^{-cn}$ probability, by Lemma H.3.

Now, for any subset $S \subset [n]$,

$$\frac{1}{n}\sum_{i \in S}\langle \widetilde{\Sigma}^{-1/2}X_iX_i^\top\widetilde{\Sigma}^{-1/2} - I, P\rangle = \frac{1}{n}\sum_{i \in S}\left(\langle J(Y_iY_i^\top - I)J^\top, P\rangle + \langle JJ^\top - I, P\rangle\right)$$

$$= \frac{1}{n}\left(\sum_{i \in S}\text{Tr}(J(Y_iY_i^\top - I)J^\top P)\right) + \frac{|S|}{n} \cdot \langle JJ^\top - I, P\rangle$$

$$= \frac{1}{n}\left(\sum_{i \in S}\langle Y_iY_i^\top - I, J^\top PJ\rangle\right) + \frac{|S|}{n} \cdot \langle JJ^\top - I, P\rangle.$$

Now, note that by our assumptions, $\|\Sigma^{-1/2}\widetilde{\Sigma}\Sigma^{-1/2} - I\|_{op} \leq 0.05$ and $\|\Sigma^{-1/2}\widetilde{\Sigma}\Sigma^{-1/2} - I\|_F \geq \rho$. Thus, by Proposition B.2, $\|\widetilde{\Sigma}^{-1/2}\Sigma\widetilde{\Sigma}^{-1/2} - I\|_{op} \leq 0.1$, and by Proposition B.2 again, applied the reverse direction this time, $\|\widetilde{\Sigma}^{-1/2}\Sigma\widetilde{\Sigma}^{-1/2} - I\|_F \geq \rho/2$. We just showed $\|JJ^\top - I\|_{op} \leq 0.1$, so by Proposition A.6, $\|J^\top PJ\|_F \leq 2\|P\|_F \leq 2$. Therefore, $\left|\frac{1}{n}\sum_{i \in S}\langle Y_iY_i^\top - I, J^\top PJ\rangle\right| \leq 0.2$.

Conversely, we just showed $\|JJ^\top - I\|_F \geq \rho/2$. So, if we order the eigenvalues of $JJ^\top$ as $\lambda_1, \lambda_2, \ldots, \lambda_d$ (and the corresponding unit eigenvectors $v_1, \ldots, v_d$) such that $|\lambda_i - 1|$ are in decreasing order, then $\sum_{i=1}^d (\lambda_i - 1)^2 \geq \rho^2/4$, which means that $\sum_{i=1}^{4d/\rho^2}(\lambda_i - 1)^2 \geq 1$. So, if we choose $P$ to be $\left(\sum_{i=1}^{4d/\rho^2}(\lambda_i - 1)^2\right)^{-1/2}\sum_{i=1}^{4d/\rho^2}(\lambda_i - 1)v_iv_i^\top$, we have that $\|P\|_F = 1$ and

$$\langle JJ^\top - I, P\rangle = \left(\sum_{i=1}^{4d/\rho^2}(\lambda_i - 1)^2\right)^{-1/2} \cdot \sum_{i=1}^{4d/\rho^2}(\lambda_i - 1)^2 = \left(\sum_{i=1}^{4d/\rho^2}(\lambda_i - 1)^2\right)^{1/2} \geq 1.$$

Therefore,

$$\frac{1}{n}\sum_{i \in S}\langle\widetilde{\Sigma}^{-1/2}X_iX_i^\top\widetilde{\Sigma}^{-1/2} - I, P\rangle \geq \frac{|S|}{n} - 0.2 \geq 1 - c - 0.2 \geq 0.7.$$

$\square$

Now, we can show how to learn the covariance of $\Sigma$ up to Frobenius error.

**Lemma H.5.** *Fix any* $\eta \in (0, \eta_0)$, *where* $\eta_0 < 0.01$ *is a small universal constant, any* $\beta \in (0, 1)$, *and any* $\widetilde{O}(\eta) \leq \rho \leq \sqrt{d}$. *There is a (deterministic, possibly inefficient) algorithm* $\mathcal{A}_2$ *with the following property. Let* $\Sigma \in \mathbb{R}^{d \times d}$ *be any covariance matrix, and let* $\mathbf{X} = \{X_1, \ldots, X_n\} \sim \mathcal{N}(0, \Sigma)$, *where* $n \geq O\left(\frac{d^2 + \log^2(1/\beta)}{\rho^2} + \log(\frac{1}{\beta})\right)$. *Then, with probability at least* $1 - \beta$ *over the randomness of* $\mathbf{X}$, *for any* $\eta$-corruption $\mathbf{X}' = \{X_1', \ldots, X_n'\}$ *of* $\mathbf{X}$, $\mathcal{A}_2(\mathbf{X}')$ *outputs* $\hat{\Sigma}_2$ *such that* $\|\Sigma^{-1/2}\hat{\Sigma}_2\Sigma^{-1/2} - I\|_F \leq O(\rho)$. *Importantly,* $\mathcal{A}_2$ *may have knowledge of* $\eta$, $\rho$, *and* $\beta$, *but does not have knowledge of* $\mathbf{X}$ *or* $\Sigma$.

*Proof.* In the case that $\rho \leq 2$, the claim follows immediately from known results (for instance, it is implicit from [HKMN23]).

Alternatively, assume that $\rho \geq 2$. In this case, the algorithm works as follows. Assume $\eta \leq \eta_0 \leq c/2$, where $c$ is the constant in Lemma H.3. First, compute $\hat{\Sigma}_1$ based on Lemma H.2. Note that $(1 - O(\eta)) \cdot \Sigma \preccurlyeq \hat{\Sigma}_1 \preccurlyeq (1 + O(\eta)) \cdot \Sigma$ with $1 - \beta$ probability, since the number of samples is sufficiently large. Now, find any positive definite $\widetilde{\Sigma}$ and a set $T \subset [n]$ of size at least $(1 - \eta)n$, such that:

- $(1 - O(\eta)) \cdot \hat{\Sigma}_1 \preccurlyeq \widetilde{\Sigma} \preccurlyeq (1 + O(\eta)) \cdot \hat{\Sigma}_1$.

- for any $S \subset T$ of size at least $(1 - 2\eta)n$, $\left|\frac{1}{n}\sum_{i \in S}\langle\widetilde{\Sigma}^{-1/2}X_iX_i^\top\widetilde{\Sigma}^{-1/2} - I, P\rangle\right| \leq 0.2$.

First, we note that $\widetilde{\Sigma} = \Sigma$ is a feasible choice of $\widetilde{\Sigma}$. Indeed, the first condition trivially holds. For the second condition, let $T$ be the subset of uncorrupted data points (i.e., $X_i' = X_i$). Then, for any $S \subset T$, the data points $X_i'$ for $i \in S$ are the same as $X_i$, so by Lemma H.3, with $1 - \beta$ probability, for every such $S$, $\left|\frac{1}{n}\sum_{i \in S}\langle\Sigma^{-1/2}X_iX_i^\top\Sigma^{-1/2} - I, P\rangle\right| \leq 0.1$.

Next, we show that every $\widetilde{\Sigma}$ with $\|\Sigma^{-1/2}\widetilde{\Sigma}\Sigma^{-1/2} - I\|_F \geq \rho$ is infeasible. First, we may assume that $0.95\Sigma \preccurlyeq \widetilde{\Sigma} \preccurlyeq 1.05\Sigma$, as otherwise, we cannot simultaneously satisfy $(1 - O(\eta)) \cdot \hat{\Sigma}_1 \preccurlyeq \widetilde{\Sigma} \preccurlyeq (1 + O(\eta)) \cdot \hat{\Sigma}_1$ and $(1 - O(\eta)) \cdot \Sigma \preccurlyeq \hat{\Sigma}_1 \preccurlyeq (1 + O(\eta)) \cdot \Sigma$, assuming $\eta \leq c/2$ is sufficiently small.

Hence, we just have to verify the infeasibility for every $\widetilde{\Sigma}$ such that $\|\Sigma^{-1/2}\widetilde{\Sigma}\Sigma^{-1/2} - I\|_F \geq \rho$ and $0.95\Sigma \preccurlyeq \widetilde{\Sigma} \preccurlyeq 1.05\Sigma$. Indeed, for any subset $T$ of size at least $(1 - \eta)n$, let $S$ be the uncorrupted points in $T$. Because there are at most $\eta n$ uncorrupted points, $|S| \geq (1 - 2\eta)n$. So by Lemma H.4, with $1 - \beta$ probability, for every such $S$, $\left|\frac{1}{n}\sum_{i \in S}\langle\widetilde{\Sigma}^{-1/2}X_iX_i^\top\widetilde{\Sigma}^{-1/2} - I, P\rangle\right| \geq 0.8$.

Therefore, with at most $O(\beta)$ failure probability, some $\hat{\Sigma}_2 := \widetilde{\Sigma}$ is returned, and it satisfies $\|\Sigma^{-1/2}\widetilde{\Sigma}\Sigma^{-1/2} - I\|_F \leq \rho$. $\qquad\square$

Next, we note some results on robust mean estimation. We first note the definition of stability, and some properties.

**Lemma H.6** ([DK22, Proposition 3.3]). *Let $n \geq O((d + \log(1/\beta))/\alpha^2)$, for some $\alpha \geq O(\eta\sqrt{\log 1/\eta})$. Let $\mathbf{X} = \{X_i\}_{i=1}^n \overset{i.i.d.}{\sim} \mathcal{D}$, where $\mathcal{D}$ is a subgaussian random variable with mean $\mu \in \mathbb{R}^d$ and covariance $I$. Then, with probability $1 - \beta$, for all vectors $b \in [0,1]^n$ such that $\mathbb{E}_i b_i \geq 1 - \eta$ and all unit vectors $v \in \mathbb{R}^d$, we have:*

1. *$|\mathbb{E}_i b_i \langle v, X_i - \mu\rangle| \leq \alpha$.*

2. *$\left|\mathbb{E}_i b_i \langle v, X_i - \mu\rangle^2 - 1\right| \leq \alpha$.*

*Given a dataset $\mathbf{X}$ with these properties, call it $(\eta, \alpha)$-stable with respect to $\mu$.*

**Lemma H.7** (implicit from [DK22, Section 2]). *Fix $\eta$ sufficiently small and $\alpha = \widetilde{O}(\eta)$. There is a deterministic algorithm $\mathcal{A}_3$ that, on a dataset $\mathbf{X}'$, outputs $\hat{\mu}$ such that $\|\hat{\mu} - \mu\|_2 \leq O(\alpha)$, for any $\eta$-corruption $\mathbf{X}'$ of any $\mathbf{X}$ that is $(\eta, \alpha)$-mean stable with respect to any $\mu \in \mathbb{R}^d$. Importantly, $\mathcal{A}_3$ does not require knowledge of $\mathbf{X}$ or $\mu$.*

We now prove Theorem B.3.

*Proof.* We first show how to estimate the covariance $\Sigma$. First, we apply a "sample pairing" trick (e.g., see [DK22, Section 4.4]). Namely, assume WLOG that $n$ is even, and define $\tilde{\mathbf{X}}$ to be the set $\{(X_{2i-1} - X_{2i})/\sqrt{2}\}_{i=1}^{n/2}$, and $\tilde{\mathbf{X}}' = \{(X'_{2i-1} - X'_{2i})/\sqrt{2}\}_{i=1}^{n/2}$. Note that $\tilde{\mathbf{X}}$ are i.i.d. samples from $\mathcal{N}(0, \Sigma)$, and $\tilde{\mathbf{X}}'$ is at most $2\eta$-corrupted.

Now, because $\eta \leq \gamma$, $\tilde{\mathbf{X}}'$ is at most $2\gamma$ corrupted, so Lemma H.2 on $\tilde{\mathbf{X}}'$ (replacing $\eta$ with $2\gamma$) gives us some $\hat{\Sigma}_1$ such that $\|\Sigma^{-1/2}\hat{\Sigma}_1\Sigma^{-1/2} - I\|_{op} \leq O(\gamma)$, by our assumed bound on the number of samples. Next, Lemma H.5 on $\tilde{\mathbf{X}}'$ gives us some $\hat{\Sigma}_2$ such that $\|\Sigma^{-1/2}\hat{\Sigma}_2\Sigma^{-1/2} - I\|_F \leq O(\rho)$, by our assumed bound on the number of samples. So, we can set $\hat{\Sigma}$ to be any covariance such that $\|\hat{\Sigma}^{-1/2}\hat{\Sigma}_1\hat{\Sigma}^{-1/2} - I\|_{op} \leq O(\gamma)$ and $\|\hat{\Sigma}^{-1/2}\hat{\Sigma}_2\hat{\Sigma}^{-1/2} - I\|_F \leq O(\rho)$. Note that $\Sigma$ satisfies these properties, and any $\hat{\Sigma}$ that satisfies these properties must satisfy $\|\Sigma^{-1/2}\hat{\Sigma}\Sigma^{-1/2} - I\|_F \leq O(\gamma)$ and $\|\Sigma^{-1/2}\hat{\Sigma}\Sigma^{-1/2} - I\|_F \leq O(\rho)$, by the approximate symmetry and transitivity properties (Proposition B.2).

Now, we estimate the mean $\mu$. Taking the original data $\mathbf{X}'$, we compute $\{\hat{\Sigma}^{-1/2}X_i'\}$. By stability (Lemma H.7), we know that with $1 - \beta$ probability, $\{\Sigma^{-1/2}X_i\}$ is $(\eta, \gamma)$-stable with respect to $\mu$ (where we are using the uncorrupted data and the true covariance $\Sigma$). Letting $J = \hat{\Sigma}^{-1/2}\Sigma^{1/2}$, we know that $J$ has all singular values between $1 - O(\gamma)$ and $1 + O(\gamma)$, and that $\{J^{-1} \cdot \hat{\Sigma}^{-1/2}X_i\}$ is $(\eta, \gamma)$-stable with respect to $\mu$. Moreover, note that we can write $\langle v, \hat{\Sigma}^{-1/2}(X_i - \mu)\rangle = \langle Jv, J^{-1} \cdot \hat{\Sigma}^{-1/2}(X_i - \mu)\rangle$, and that $1 - O(\gamma) \leq \|Jv\|_2 \leq 1 + O(\gamma)$. Therefore, $\{\hat{\Sigma}^{-1/2}X_i\}$ is $(\eta, O(\gamma))$-stable with respect to $\hat{\Sigma}^{-1/2}\mu$, which means that by Lemma H.7, $\mathcal{A}_3$ on $\{\hat{\Sigma}^{-1/2}X_i'\}$ outputs some value $\hat{\nu}$ such that $\|\hat{\nu} - \hat{\Sigma}^{-1/2}\mu\|_2 \leq O(\gamma)$. Thus, by setting $\hat{\mu} := \hat{\Sigma}^{1/2} \cdot \hat{\nu}$, we have that $\|\hat{\Sigma}^{-1/2}(\hat{\mu} - \nu)\|_2 \leq O(\gamma)$, which means that $\|\Sigma^{-1/2}(\hat{\mu} - \mu)\|_2 = \|J^{-1}\hat{\Sigma}^{-1/2}(\hat{\mu} - \mu)\|_2 \leq (1 + O(\gamma)) \cdot \|\hat{\Sigma}^{-1/2}(\hat{\mu} - \mu)\|_2 \leq O(\gamma)$. $\qquad\square$

### H.3 Proof of Lemma C.7

In this subsection, we prove Lemma C.7.

*Proof.* First, note that for any positive definite matrices $\Sigma_1, \Sigma_2$, $\|\Sigma_2^{-1/2}\Sigma_1\Sigma_2^{-1/2} - I\|_{op} \leq \gamma$ is equivalent to $(1 - \gamma)I \preccurlyeq \Sigma_2^{-1/2}\Sigma_1\Sigma_2^{-1/2} \preccurlyeq (1 + \gamma)I$. Since $M$ being PSD implies $AMA^\top$ is PSD

(and vice versa if $A$ is invertible), the previous statement is thus equivalent to $(1 - \gamma) \cdot \Sigma_2 \preccurlyeq \Sigma_1 \preccurlyeq (1 + \gamma)\Sigma_2$. Next, note that

$$
\begin{aligned}
\|\Sigma_2^{-1/2}\Sigma_1\Sigma_2^{-1/2} - I\|_F &= \sqrt{\mathrm{Tr}\left((\Sigma_2^{-1/2}\Sigma_1\Sigma_2^{-1/2} - I)^2\right)} \\
&= \sqrt{\mathrm{Tr}\left(\Sigma_2^{-1/2}\Sigma_1\Sigma_2^{-1}\Sigma_1\Sigma_2^{-1/2} - 2\cdot\Sigma_2^{-1/2}\Sigma_1\Sigma_2^{-1/2} + I\right)} \\
&= \sqrt{\mathrm{Tr}\left(\Sigma_1\Sigma_2^{-1}\Sigma_1\Sigma_2^{-1} - 2\cdot\Sigma_1\Sigma_2^{-1} + I\right)},
\end{aligned}
\tag{4}
$$

where the first two lines are a straightforward expansion, and the final line simply uses the fact that $\mathrm{Tr}(AB) = \mathrm{Tr}(BA)$ for any matrices $A, B$. Finally, note that

$$
\|\Sigma_2^{-1/2}(\mu_1 - \mu_2)\|_2 = \sqrt{(\mu_1 - \mu_2)^\top \Sigma_2^{-1}(\mu_1 - \mu_2)}.
\tag{5}
$$

Now, consider replacing $\Sigma_1$ with $\Sigma_3 := \Sigma^{1/2}\Sigma_1\Sigma^{1/2}$, $\Sigma_2$ with $\Sigma_4 := \Sigma^{1/2}\Sigma_2\Sigma^{1/2}$, $\mu_1$ with $\mu_3 := \Sigma^{1/2}\mu_1 + \mu$, and $\mu_2$ with $\mu_4 := \Sigma^{1/2}\mu_2 + \mu$. Again, since $M$ being PSD implies $AMA^\top$ is PSD (and vice versa), we have that $(1-\gamma)\cdot\Sigma_2 \preccurlyeq \Sigma_1 \preccurlyeq (1+\gamma)\cdot\Sigma_2$ if and only if $(1-\gamma)\cdot\Sigma_4 \preccurlyeq \Sigma_3 \preccurlyeq (1+\gamma)\cdot\Sigma_4$. Moreover, note that

$$
\mathrm{Tr}(\Sigma_3\Sigma_4^{-1}\Sigma_3\Sigma_4^{-1}) = \mathrm{Tr}(\Sigma^{1/2}\Sigma_1\Sigma_2^{-1}\Sigma_1\Sigma_2^{-1}\Sigma^{-1/2}) = \mathrm{Tr}(\Sigma_1\Sigma_2^{-1}\Sigma_1\Sigma_2^{-1})
$$

and

$$
\mathrm{Tr}(\Sigma_3\Sigma_4^{-1}) = \mathrm{Tr}(\Sigma^{1/2}\Sigma_1\Sigma_2^{-1}\Sigma^{-1/2}) = \mathrm{Tr}(\Sigma_1\Sigma_2^{-1}),
$$

which means that (4) would be the same if we replaced $\Sigma_1$ with $\Sigma_3$ and $\Sigma_2$ with $\Sigma_4$. Finally,

$$
(\mu_3-\mu_4)^\top\Sigma_4^{-1}(\mu_3-\mu_4) = (\mu_1-\mu_2)^\top\Sigma^{1/2}(\Sigma^{-1/2}\Sigma_2^{-1}\Sigma^{-1/2})\Sigma^{1/2}(\mu_1-\mu_2) = (\mu_1-\mu_2)^\top\Sigma_2^{-1}(\mu_1-\mu_2),
$$

which means that (4) would be the same if we replaced $\mu_1$ with $\mu_3$, $\mu_2$ with $\mu_4$, $\Sigma_1$ with $\Sigma_3$, and $\Sigma_2$ with $\Sigma_4$.

Overall, by the definition of $\approx_{\gamma,\rho,\tau}$, we have that $(\mu_1, \Sigma_1) \approx_{\gamma,\rho,\tau} (\mu_2, \Sigma_2)$ if and only if $(\mu_3, \Sigma_3) \approx_{\gamma,\rho,\tau} (\mu_4, \Sigma_4)$. $\qquad\square$

