# OpenReview forum: "Sample-Efficient Private Learning of Mixtures of Gaussians"
_NeurIPS.cc/2024/Conference — NeurIPS 2024 spotlight_

### Official Review · Reviewer_cTn5 · 2024-07-08

**Soundness:** 3
**Presentation:** 3
**Contribution:** 3
**Rating:** 6
**Confidence:** 3

**Summary:**

The paper studies the important problem of private learning of the Gaussian mixture model to estimate the underlying distribution within a desired total variation distance. By combining different techniques, the authors succeed in deriving bounds that are of quadratic dimension, thus significantly improving the existing results.

**Strengths:**

The paper contains several new results that significantly improve the state of the art. These include i) Theorem 1.3, which establishes a bound with quadratic complexity for any dimension, ii) Theorem 1.4, which proposes an improved bound for $d=1$, showing that the sample complexity can be linear in the number of components, iii) Theorem 1.5, which proposes a lower bound on the sample complexity. The latter, combined with Theorem 1.4, shows the optimality of the established bounds for the univariate Gaussian distributions.  In addition, the paper is generally well and smoothly written.

**Weaknesses:**

- The paper could benefit from some numerical verifications of the results/algorithms used.

- The appendix section is poorly organized. Without an outline and proof, it is difficult to follow such a long appendix.

**Questions:**

1. It would be better to explain more the difference between density estimation and parameter estimation.

2. How do the bounds of Theorems 1.3, 1.4., and 1.5 behave with respect to the failure probability $\beta$ order-wise?

3. Can the authors discuss the assumption $R=n^{100}$ (line 160) a bit more? Why does it depend on $n$ (and not $k$ or $d$?)? Also, is the polynomial dependence restrictive?

4. The first sentence of the informal statement of Theorem 2.1. does not seem rigorous enough. Is such a $\tilde{\Sigma}$ unique? I don't think so. Then, if not, which choice of $\tilde{\Sigma}$ is taken into account when computing $V_{\eta}(\mathbf{X})$? Probably maximum volume?

5. The appendix sections are very difficult to follow. There should be a detailed outline at the beginning to guide the reader as to what each appendix deals with. Also, there should be sufficient references in the main text. For example, after the informal statement of Theorem 2.1., it should be clearly stated where the full statement and proof can be found. Similarly, after each paragraph (e.g., Section 2.1) or after each mentioned previously established result (e.g., advanced composition theorem), there should be an exact pointer to where the complete version can be found in the appendices.

6. Since the submission was allowed for 9 pages, the authors had 1.5 more pages. I think some material from the appendices, for example the algorithms used to privately learn GMM, could be moved to the main text.

7. Finally, what is missing the most is the experimental section. If I'm not mistaken, this should be easy to do. Although not all theoretical work requires experimental verification, I believe that if the experiments verify the theoretical finding, this would greatly increase the importance of the results.

Minor comments:

- Line 52: it is better to add "The parameters $\alpha$ and $\beta$...".

- The constant $c^*$ in Theorem 1.5. does not appear in the bound.

- In the first paragraph of Section 2.1, it is mentioned that "... as we can finish the procedure with hypothesis selection, as discussed above". However, hypothesis selection has not been sufficiently discussed.

- Is it true that the intuition given in Section 2.1. holds if $ n \gg k^2$?

- Typo line 178 (a a)

- Is $k$ in lines 203-204 the number of components in the Gaussian mixture? Why does this statement ("if we altered $k$ data points...") only hold for $k$ changes?

---

> ### Author Rebuttal · Authors · 2024-08-07
>
> We would like to thank the reviewer for their detailed feedback. In the following we answer the raised questions.
>
> 1. To clarify this further, the goal of density estimation is to learn the overall distribution’s PDF (up to low total variation distance), whereas in parameter estimation we want to learn every Gaussian component’s mean and covariance. One can construct two different Gaussian mixtures with very similar PDFs, but with somewhat different components (in which case identifying the wrong Gaussian mixture is OK for density estimation but not for parameter estimation). We will add this discussion to the paper.
>
> 2. For the sake of clarity of the presentation we tried to avoid writing down the exact dependency on beta in our bounds. However, note that the sample complexity scales at most logarithmically with 1/beta. The reason is that any method that achieves a failure probability of < 0.50 can be boosted to have failure probability of beta with a mild (logarithmically in 1/beta) increase in the sample complexity. This can be done by running the estimator on log(1/beta) data sets, and then simply running private hypothesis selection on the outcomes. We will add this discussion to the paper.
>
> 3. The choice of $n^{100}$ is just for convenience, and the effect of the constant 100 on the sample complexity is negligible (i.e., the sample complexity does not change in terms of the $\tilde{O}(.)$ notation). In Line 157, we mention that the sample complexity (after the crude approximation is obtained) depends on $\log R$, so even if $R = n^{100}$ this only blows up the sample complexity by $\log(n^{100}) = O(\log n)$.
>
> 4. Indeed, the $\tilde{\Sigma}$ is not unique, and the volume in Theorem 2.1 refers to the volume (i.e., Lebesgue measure) of the set of all possible $\tilde{\Sigma}$, for a fixed choice of dataset $X$. As an example, if $d = 1$ (in which case $\tilde{\Sigma}$ is just a variance) and every $\tilde{\Sigma}$ between $1$ and $4$ satisfies the score function, then the volume is 3. In general, we use higher dimensional Lebesgue measure (see Appendix C.1) to formally define the volume.
>
> 5 and 6. We thank the reviewer for their suggestions about improving the presentation of the paper and the appendices. We will use the remaining space to give more details in the main paper (and will also improve the structure of the appendix to make it easier to navigate)
>
> 7. We would like to emphasize that our work focuses on the fundamental problem of determining the number of samples for learning a GMM privately. However, our algorithm is not computationally efficient. Note that even without privacy, it is not known whether GMMs can be learned efficiently in high dimensions. This remains an intriguing open problem in the field of computational statistics.
>
> Minor comments:
>
> Regarding Theorem 1.5, we note that $c^*$ will be some universal constant, so it can be hidden in the $\Omega$ notation. The intuition in Section 2.1 holds as long as $n \gg k$, so it also holds if $n \gg k^2$. In lines 203-204, the use of $k$ is a mistake - we will use $t$ (or another variable) to avoid confusing it with the number of components $k$. For all other comments, we agree with you and we will incorporate your feedback.

---

> > ### Comment · Reviewer_cTn5 · 2024-08-09
> >
> > I thank the authors for the clarifications. I believe the paper merits the publication, and I maintain my initial rating.

---

### Official Review · Reviewer_sFBV · 2024-07-09

**Soundness:** 3
**Presentation:** 2
**Contribution:** 3
**Rating:** 6
**Confidence:** 3

**Summary:**

This paper investigates the sample complexity of privately learning mixtures of Gaussians. The authors achieve a sample complexity of approximately $O(kd^2+k^{1.5}d^{1.75}+k^2d)\log R$ where $R$ is an upper bound on the condition number of the covariance matrix and the norm of the mean. This result improves upon the previous best bound of $O(k^2d^4)$. Additionally, they constructed a lower bound of $\Omega(kd^2+k\log(1/\delta)/\epsilon)$, which refutes a conjecture from prior work, and they achieve optimal sample complexity when $d=1$.

**Strengths:**

The improvement in sample complexity is significant.
The paper provides a thorough technical overview of the upper bound, detailing the incorporation of sample compression and the methods used to enhance dimension independence within the robustness-to-privacy conversion technique.

**Weaknesses:**

The paper is technically dense, making it challenging to understand and verify all the details.
The authors do not fully utilize the page limit, which could have been used to explain the techniques more clearly.
The technique for establishing the lower bound is not discussed in the main body of the paper.
Including formal definitions and crucial lemmas in the main text could help readers better understand the key ideas. For instance, the definition of volume was unclear until I consulted the appendix.

**Questions:**

1. Where does the $k^2$ term come from in the upper bound?
2. Are there alternative methods to address this problem that do not rely on the robustness-to-privacy conversion technique?

**Limitations:**

Yes

---

> ### Author Rebuttal · Authors · 2024-08-07
>
> We would like to thank the reviewer for their detailed feedback. We would like to emphasize that our bound does not depend on the condition number of the covariance matrix nor on the magnitude of the mean, i.e., our sample complexity has no dependence on $\log R$. (Otherwise, proving an upper bound would have been easy, e.g., by using private hypothesis selection on a finite cover for the set of possible mixtures.) We agree with the comments about the presentation of the paper, and will improve it in the next version (including adding more details about the definitions,lemmas, or proof sketches in the main paper). In the following we answer the raised questions.
>
> 1. To explain the $\frac{k^2 d}{\alpha}$ term, the point is that for the private algorithm to work, we need two things. First, the private algorithm finds a covariance matrix (or more precisely, a mean-covariance pair $(\tilde{\mu}, \tilde{\Sigma})$) with low score (see Line 932 and the above few lines for the definition). Second, if a mean-covariance pair has a low score, then it is actually similar to a true mixture component $(\mu_i, \Sigma_i)$.  The term $\frac{d^{1.75} k^{1.5} \sqrt{\log (1/\delta)}}{\alpha \epsilon}$ is needed for the first part. However, the $\frac{k^2 d}{\alpha}$ term is needed for the second part: that any mean-covariance pair of low score is similar to some true mixture component (this is the goal of Proposition E.4 in our paper).
>
> The reason for the $k^2$ dependence in this second term is nontrivial so here’s a high-level intuition. Given a dataset $X$ and mean-covariance pair $(\tilde{\mu}, \tilde{\Sigma})$, the score $S((\tilde{\mu}, \tilde{\Sigma}), X)$ is small if there exists roughly $\alpha/k$ fraction of data points that “look like” they came from $\mathcal{N}(\tilde{\mu}, \tilde{\Sigma})$. The point is that one can have one point come from each of $k$ different mixture components which, together, look like $k$ samples from a totally different Gaussian from any of the $k$ mixture components. As a result, we need to make sure we have more than $k^2/\alpha$ total points, because then an $\alpha/k$ fraction of the data points is more than $k$ total samples. We actually need an additional factor of $d$, because it turns out that we can even have $\Omega(d)$ points from each of the mixture components which look like $\Omega(kd)$ samples from a totally different Gaussian.
>
> 2. The only other known approach for privately learning unbounded and high-dimensional GMMs (density estimation) is [AAL24] that uses sample-and-aggregate. For the univariate setting (density estimation for GMMs with unbounded parameters), there is another approach that uses stability-based histograms [AAL21].

---

> > ### Comment · Reviewer_sFBV · 2024-08-12
> >
> > Thanks for the response. I maintain my score.

---

### Official Review · Reviewer_9BbN · 2024-07-11

**Soundness:** 3
**Presentation:** 3
**Contribution:** 3
**Rating:** 6
**Confidence:** 3

**Summary:**

This works focuses on the task of density estimation of a mixture of Gaussians under the restriction of differential privacy. Unlike parameter estimation, density estimation does not require accurately estimating the mixture's parameters, but instead bounds the total variation distance between estimated and true distributions. This task can be achieved even without any boundedness or separation assumptions on the parameters of the components.

This task is known to require sample size of $O(k d^{2}/\alpha^{2})$ even in the non-private setting, where $k$ is the number of components, $d$ is the dimension, $\alpha$ is the bound on the TV distance, and the $O$ notation ignores logarithmic factors. In the private setting, previous work [1] achieved $\alpha$ accuracy guarantee under $(\epsilon, \delta)$-DP with $O(k^{2} d^{4}/\alpha^{2} \epsilon)$ (the exact bound includes several other terms that depend on $\log(1/\delta)$ as well).

This work provides an improved bound, reducing the dependence on the parameters to $O \left(\frac{k d^{2}}{\alpha^{2}} + \frac{k d^{2}}{\alpha \epsilon} \right)$ in the high dimensional regime, $O \left(\frac{k}{\alpha^{2}} + \frac{k \cdot \log(1/\delta)}{\alpha \epsilon} \right)$ in the univariate setting, and lower bounds which asymptotically match the upper bound in the univariate case and nearly match it in the multivariate one.

The proposed algorithm relays on first achieving crude estimation of the parameters and then using hypothesis selection to improve the estimation. The crude estimation is achieved using robust GMM estimation techniques and robusteness-to-privacy conversion, based on an inverse sensitivity-like instantiation of the exponential mechanism. This method is computationally inefficient.

[1] Mohammad Afzali, Hassan Ashtiani, and Christopher Liaw. Mixtures of gaussians are privately learnable with a polynomial number of samples. In Algorithmic Learning Theory, pages 1–27. PMLR, 2024.

**Strengths:**

The results presented in this work provide a significant improvement over the previously known ones. Though the proof technique is highly involved, the authors presented the proof outline in a relatively intuitive way, and provided motivation for the various steps it includes.

**Weaknesses:**

Despite the great work done by the authors and my best efforts, I was not able to follow up all the proof structure. In particular, I could not find the justification for some of terms that appeared in the bound presented on Theorem 1.3. It seems to me like a section that "puts everything together" will be of great benefit. I will try to describe my current understanding and existing gaps.

To the best of my understanding:
* The $\frac{k d^{2}}{\alpha^{2}}$ and $\frac{k d^{2}}{\alpha \epsilon}$ were explained at the "Reducing to “crude approximation”" section, and they represent the sample size required to accurately and privately learn the GMM given some crude estimation of its components, using hypothesis selection method.
* The $\frac{d^{1.75} k^{1.5} \sqrt{\log(1/\delta)}}{\alpha \epsilon}$ was explained at the "Improving Dimension Dependence" section, where $\frac{d^{1.75} k}{\alpha \epsilon}$ results from the fact $O\left(\frac{d^{1.75} k}{c \epsilon} \right)$ points are required to get the crude estimation of the parameters of a single component under $c$-robustness, which then can be translated to a DP estimation with $\alpha$ accuracy using rubousteness-to-privacy conversion techniques (Theorem 2.1), and the additional $\sqrt{k \cdot \log(1/\delta)}$ term results from advanced composition over $k$ components.
* I failed to understand where the additional two terms ($\frac{(k \cdot \log(1/\delta))^{1.5}}{\alpha \epsilon}$ and $\frac{k^{2} d}{\alpha}$) come from, and I suspect they were accumulated at some point during the rubousteness-to-privacy transformation.

**Questions:**

As I mentioned before, I will find an additional section that brings all the components together to outline the final proof method, focusing on the final achieved bound, to be very useful.

---

> ### Author Rebuttal · Authors · 2024-08-07
>
> We would like to thank the reviewer for their detailed feedback. We are happy to add some description that puts everything together and explain where each of the terms come from, as suggested by the reviewer. In the following, we explain the terms in the sample complexity that the reviewer asked about.
>
> To explain the $\frac{(k \log (1/\delta))^{1.5}}{\alpha \epsilon}$ term, we note that in the crude estimation part, the sample complexity (from applying Theorem C.3, the formal version of Theorem 2.1) also has an assumption (see the “Volume” bullet) that $n \ge \frac{\log (1/\delta_0)}{\epsilon_0 \cdot \eta^*}$. Here, $\epsilon_0, \delta_0$ represent the privacy terms for a single iteration of the crude estimation (to learn one parameter), and $\eta^*$ will represent the robustness threshold, and ends up being roughly $\alpha/k$. The reason the robustness threshold depends on $k$ like this is that each component on average has weight $1/k$, so you can corrupt $1/k$ fraction points and completely alter a component. Also, since we run the crude estimation $k$ times, we can use advanced composition to say that if each iteration was $(\epsilon_0, \delta_0) = (\frac{\epsilon}{\sqrt{k \log (1/\delta)}}, \frac{\delta}{k})$-DP, the overall algorithm is $(\epsilon, \delta)$-DP. With all of these parameters set, we precisely get $ \frac{\log (1/\delta_0)}{\epsilon_0 \cdot \eta^*} = \Theta\left(\frac{(k \log (1/\delta))^{1.5}}{\alpha \epsilon}\right)$.
>
> To explain the $\frac{k^2 d}{\alpha}$ term, the point is that for the private algorithm to work, we need two things. First, the private algorithm finds a covariance matrix (or more precisely, a mean-covariance pair $(\tilde{\mu}, \tilde{\Sigma})$) with low score (see Line 932 and the above few lines for the definition). Second, if a mean-covariance pair has a low score, then it is actually similar to a true mixture component $(\mu_i, \Sigma_i)$.  The previous terms of $\frac{d^{1.75} k^{1.5} \sqrt{\log (1/\delta)}}{\alpha \epsilon}$ and $\frac{(k \log (1/\delta))^{1.5}}{\alpha \epsilon}$ are needed for the first part. However, the $\frac{k^2 d}{\alpha}$ term is needed for the second part: that any mean-covariance pair of low score is similar to some true mixture component (this is the goal of Proposition E.4 in our paper).
>
> The reason for the $k^2$ dependence is nontrivial so here’s a high-level intuition. Given a dataset $X$ and mean-covariance pair $(\tilde{\mu}, \tilde{\Sigma})$, the score $S((\tilde{\mu}, \tilde{\Sigma}), X)$ is small if there exists roughly $\alpha/k$ fraction of data points that “look like” they came from $\mathcal{N}(\tilde{\mu}, \tilde{\Sigma})$. The point is that one can have one point come from each of $k$ different mixture components which, together, look like $k$ samples from a totally different Gaussian from any of the $k$ mixture components. As a result, we need to make sure we have more than $k^2/\alpha$ total points, because then an $\alpha/k$ fraction of the data points is more than $k$ total samples. We actually need an additional factor of $d$, because it turns out that we can even have $\Omega(d)$ points from each of the mixture components which look like $\Omega(kd)$ samples from a totally different Gaussian.

---

> > ### Comment · Reviewer_9BbN · 2024-08-09
> >
> > I thank the authors for their explanation, and hope it will be reflected in the final version, so that all readers will have the opportunity to fully comprehend this important result.

---

### Official Review · Reviewer_u11i · 2024-07-17

**Soundness:** 3
**Presentation:** 4
**Contribution:** 3
**Rating:** 7
**Confidence:** 3

**Summary:**

The paper studies the problem of learning a mixture of $k$ $d$-dimensional Gaussians using a differentially private mechanism with respect to the samples. It provides an improved sample complexity which has asymptotically optimal dependence on the dimension $d$ for small $k$, the lower bound is also given by the paper. Additionally, for $d=1$ the paper provides optimal sample complexity.
The paper follows a high level approach of obtaining crude approximations of the mean and covariances of the component Gaussians, which can be used to reduce – using a net based argument – the problem to private hypothesis selection for which existing algorithm (needing only log the size of the net number of samples) can be used. To obtain the crude approximations respecting the DP guarantee, the paper uses a variant of the exponential mechanism with a carefully constructed scoring function measuring the distance between the a candidate Gaussian and any “heavy” component of Gaussian mixture. This uses a number of samples depending on the size of the sample set $n$ and the dimensionality the hypotheses $d^2$. A naive approach fails due to the blowup incurred, and the paper utilizes sample compression to reduce the dependence to $O(d\\log n)$ and improves the dimensionality dependence to $O(d)$ via an estimation argument.
The above is outlined in more detail in the main paper and proved formally in the appendices. While all the details have not been verified by the reviewer, the approach taken by the paper seems correct.

**Strengths:**

1. The paper proves improved (and in some cases optimal) sample complexity for privately learning Gaussian mixtures.
2. The paper uses a novel crude approximation based approach paired with existing algorithm for private hypothesis selection.
3. The paper leverages inverse sensitivity mechanisms for  decoding the crude approximations, techniques for sample compression, and combines them with a way to improve the dimensionality dependence.
4. The main result as well the one for univariate case and the lower bound together constitute notable progress on a well studied problem.
5. The paper is well written and the provided roadmap greatly aids the understanding.

**Weaknesses:**

1. The sample complexity does not match the lower bound in the dependence on $k$.
2. The degradation on the dependence on $k$ from the univariate to the multi-variate case is not explained in the main paper.
3. The details of the various technical parts are tedious and could be alleviated by better presentation, e.g. by listing all the parameters and their dependencies in a table.

**Questions:**

It will be useful to combine definitions 1.1 and 1.2 into a precise definition of privately learning GMMs.

**Limitations:**

Yes, addressed.

---

> ### Author Rebuttal · Authors · 2024-08-07
>
> We would like to thank the reviewer for their detailed feedback. We now address some of the issues/questions raised by the reviewer.
>
> Note that the algorithm for the univariate case (Section F) is completely different from the multivariate case (Section E). The algorithm in the univariate case requires us to order the data points from smallest to largest, which doesn’t make sense in high dimensions. So, it only works for $d = 1$. We could use the multivariate algorithm when $d = 1$ as well, but we would get a worse dependence on $k$ (more specifically, we would get $k^2$ in the bound rather than linear dependence on $k$).
>
> We are happy to list our results, along with previous results, in a table so that one can easily compare our work with previous work (as well as compare upper/lower bounds). We also agree that a final definition combining Definitions 1.1 and 1.2 would be useful, we will add that immediately after these two definitions.

---

> > ### Comment · Reviewer_u11i · 2024-08-11
> >
> > I acknowledge the rebuttal by the authors. Just to clarify, in my weakness no. 3 comment, I had suggested listing the parameters used in the different proofs and their dependencies in table(s). This would make the proofs a bit more accessible in my opinion and I hope the authors will look into it.
> > Overall my rating of Accept remains unchanged.

---

### Author Rebuttal · Authors · 2024-08-07

We thank all of the reviewers for their helpful feedback.

We apologize for the difficulty in reading the paper. We will make changes as suggested by the reviewers to improve the readability of the paper, such as adding a table of results, adding some additional technical description to the main body (up to space limitations), and adding some outline and summary sections in the appendix so that it will be more structured and understandable.

---

### Decision · Program_Chairs · 2024-09-25

**Decision:**

Accept (spotlight)

**Comment:**

The paper considers the problem of estimating k-component and d-dimensional Gaussian mixtures with central differential privacy. Previous algorithms required k^2 d^4 samples and the proposed work shows that kd^2 + k^1.5 d^1.75 + k^2 d samples suffice. In the high-dimensional setting, when d >>k^2, they further show that this bound is optimal. They also show that sample complexity of learning one dimensional Gaussian mixtures is linear in k. The results are interesting and I recommend acceptance.

As reviewers mention, the writing is dense and based on my own read of the paper, I completely agree.  I strongly urge authors to utilize the full page limit and improve the writing in the final version of the paper by addressing reviewer concerns.  It would also be great if authors can add  additional intuition on how each of the terms in the sample complexity arise and also highlight the technical novelty that enabled this result.